# Transparent integrated pyroelectric-photovoltaic structure for photo-thermo hybrid power generation

Malkeshkumar Patel[1,2], Hyeong-Ho Park [3], Priyanka Bhatnagar[1,2], Naveen Kumar[1,2], Junsik Lee[1,2] & Joondong Kim [1,2] ✉

Thermal losses in photoelectric devices limit their energy conversion efficiency, and cyclic input of energy coupled with pyroelectricity can overcome this limit. Here, incorporating a pyroelectric absorber into a photovoltaic heterostructure device enables efficient electricity generation by leveraging spontaneous polarization based on pulsed light-induced thermal changes. The proposed pyroelectric-photovoltaic device outperforms traditional photovoltaic devices by 2.5 times due to the long-range electric field that occurs under pulse illumination. Optimization of parameters such as pulse frequency, scan speed, and illumination wavelength enhances power harvesting, as demonstrated by a power conversion efficiency of 11.9% and an incident-photon-to-current conversion efficiency of 200% under optimized conditions. This breakthrough enables reconfigurable electrostatic devices and presents an opportunity to accelerate technology that surpasses conventional limits in energy generation.

Compound capabilities of energy harvesting using semiconductor technology can be realized through the complementary utilization of photovoltaics and pyroelectrics. Due to the accompanying light and thermal features in such systems, there is a greater chance of realizing multi-energy harvesting schemes. The light-responsive physics of piezoelectricity, ferroelectricity, flexoelectricity, and pyroelectricity offer a device design methodology for integrated systems that comprise low-cost materials and offer efficiencies and functionalities above the thermodynamic limits of the individual technologies[1–6]. Therefore, such platforms afford opportunities for interesting and practical combinations. Light-responsive entities present opportunities for a highly practical combination that makes use of photonic and thermal energy. Notably, while photons are widely recognized for their ability to convert energy through photovoltaic effects, the thermal energy that comes with photon absorption has yet to be utilized in mature systems.

Pyroelectric phenomena occur due to the spontaneous polarization ($P_S$) of certain materials as a function of temperature. The polarity of these materials is a prerequisite for pyroelectric behavior. When a polar material is subjected to heat disturbance, it results in internal spontaneous polarization as cation/anion delocalization occurs[7–9]. Primary pyroelectricity (clamped lattice) refers to the polarization that develops due to an internal process in a dielectric, whereas secondary pyroelectricity (electron-phonon) occurs due to external influences such as thermal expansion[2,10–13]. The main pyroelectricity-dominated $P_S$ is characterized by Born effective charges, which are controlled by the ionized cation/anion charges[14]. Polar pyroelectric materials such as ZnO, GaN, CdS, $In_2S_3$, $BiFeO_3$, $PbTiO_3$, and $SrTiO_3$[10–15] have been applied in thermal sensors[16], photodetectors[17], nanogenerators[18], energy harvesters[3], hydrogen generation[19], solar cells[20], nuclear fusion[21], and water purification[22]. From the mechanism perspective, it is well known that the pyro-phototronic effect enhances photovoltaic energy harvesting by enhancing the pyro-photocurrent signal; its quantum efficiency and power conversion efficiency (PCE) are currently unknown[17,23–25].

[1]Photoelectric and Energy Device Application Lab (PEDAL), Multidisciplinary Core Institute for Future Energies (MCIFE), 119 Academy Rd. Yeonsu, Incheon 22012, Republic of South Korea. [2]Department of Electrical Engineering, Incheon National University, 119 Academy Rd., Yeonsu, Incheon 22012, Republic of South Korea. [3]Optical Device Lab., Device Technology Division, Korea Advanced Nanofab Center (KANC), 109 Gwanggyo-ro, Yeongtong-gu, Suwon 16229, Republic of South Korea. ✉e-mail: joonkim@incheon.ac.kr

**Table 1 | Summary of the application evaluation based on the various pyroelectric phenomena for energy conversion, where $P_i$ is the pyroelectric coefficient, FOM is the figure of merit, τ is the response speed, PCE is the power conversion efficiency, and IPCE is the incident-photon-to-current conversion efficiency**

| Application | Device | Pyroelectric phenomena | Input energy | $P_i$ (μC m$^{-2}$ K$^{-1}$) | FOM (cm$^2$ μC$^{-1}$) | τ (μs) | PCE (%) | IPCE (%) | Year ref. |
|---|---|---|---|---|---|---|---|---|---|
| Thermal sensors | Triglycine sulfate | Pyroelectric | IR radiation | 270 | - | - | - | - | 2005[16] |
| Nuclear fusion | LiTaO$_3$ crystal | Pyroelectric | Neutron flux | - | - | - | - | - | 2005[21] |
| UV Photodetector | FTO/ZnO/MAPbI$_3$/HTM/Cu | Light-induced pyroelectric | Pulsed light 325 nm | - | - | 53 | - | - | 2015[17] |
| Hybrid energy cell | PEDOTs | Photo-thermal-pyroelectric | NIR sunlight | - | - | - | 11.7 | - | 2015[20] |
| Bolometer Mid-infrared photodetector | LiNbO$_3$/graphene | Pyroelectric | Polarized quantum cascade LASER (6.2–10 μm) | 780 | - | - | - | - | 2017[26] |
| Energy harvesters | Pb(Mg$_{1/3}$Nb$_{2/3}$)O$_3$–PbTiO$_3$ | Pyroelectric | Joule heating (25–115 °C) | 550 | - | - | - | - | 2018[3] |
| Hydrogen generation Water purification | Few layer black phosphorene | Pyro-catalytic | IR lamp thermal cycles (15–65 °C) | 5,287 | - | - | - | - | 2018[19] |
| Nanogenerator | PVDF film PZT ceramic | Pyroelectric | Thermal cycle using Peltier module (26–55 °C) | 27,000 | - | - | - | - | 2019[18] |
| Water purification | Pb(Zr$_{0.52}$Ti$_{0.48}$)O$_3$ | Pyro-catalytic | Water-bath (15–70 °C) | 605 | - | - | - | - | 2019[30,22] |
| CO$_2$ to methanol reduction | Bi$_2$WO$_6$ | Pyro-catalytic | Water-bath (15–70 °C) | - | - | - | - | - | 2021[31] |
| Hydrogen production | BaTiO$_3$-Au | Pyro-catalytic | Nanosecond LASER 532 nm, 786 mW/cm$^2$ | 300 | - | - | - | - | 2022[27] |
| Tooth whitening | BaTiO$_3$ nanowire | Pyro-catalysis | Oral temperature fluctuation | 210 | - | - | - | - | 2022[32] |
| Wide spectrum photodetector | Au/Molecular N-IBATFA/Cu | Photo-pyroelectric | Photon 266–1950 nm | 69,000 | 188× 10$^{-2}$ | - | - | | 2023[28] |
| Slippery surfaces microrobots | SiO$_2$/LiNbO$_3$ | Photo-pyroelectric | LASER 808 nm, 100–1000 mW | 83 | - | - | - | - | 2023[29] |
| Pyro-photovoltaic | FTO/ZnO/NiO/AgNW/ZnO | Pyro-photovoltaic | Pulsed illumination 340–520 nm | 25000–35000 | 58–121 | <10 | 11 | 200 | This work |

Table 1 summarizes the application evolution of pyroelectric materials, including ZnO, BaTiO₂, LiNbO₃, and PZT. The input energy of photon radiation from UV–visible-NIR[16,17,20,26–29], neutron flux[21], Joule heating[3,18,20], water baths[22,30,31], and oral temperature fluctuation[32] has been utilized by pyroelectric materials due to changes in the thermal state for thermal energy conversion. The pyroelectric coefficient ($P_i$) resulting from the thermal change of $P_S(dP_S/dT)$ is μC m$^{-2}$ K$^{-1}$. Various pyroelectric materials are available in the range of $P_i$ values. Recently, Pandya et al. demonstrated that the electric field-driven enhancement of $P_i$ yielded a pyroelectric energy conversion efficiency of 19% for Carnot, where low-grade heat is converted into electrical energy[3].

Pyroelectric energy conversion is a process that includes using varying temperature profiles to extract electrical energy from the pyroelectric, in contrast to the steady-state operations used in traditional energy harvesting. Pyroelectric devices have shown improved photoresponses and self-powering capabilities in response to the application of pulsed light, making them great platforms for realizing broadband sensing applications[13,25,33,34]. When combined with photovoltaic structures, these devices are expected to generate significant amounts of power by introducing heat loss into the electric cycle. ZnO is one of the most promising materials for pyroelectric energy conversion because it has demonstrated extraordinary performance compared to other pyroelectric candidates[10,11,35,36]. However, whether a power generation device incorporating a pyroelectric film can achieve a power conversion efficiency above the thermodynamic limit is currently unknown.

Figure 1a shows the Carnot energy diagram, which represents the work done by a material while transferring heat from a hot temperature source ($T_H$ = 5800 K for the sun) to a cold sink ($T_C$ = 300 K for the device). When the conversion process is reversible, the energy conversion efficiency of the Carnot engine is $\eta = 1 - \frac{T_C}{T_H}$, which is 94.82%[37]. In practice, energy conversion devices can consume energy in steady or pulsed modes, as shown in Fig. 1b. Single-junction photovoltaic cells, which are usually used in steady-state mode, have a limited power conversion efficiency (PCE) due to the bandgap of the absorbing material and the thermalization process. However, pyroelectric materials can overcome this thermodynamic limit by producing work under pulsed illumination. Figure 1c shows that the pyroelectric material does cyclic work under pulsed illumination, which exceeds the power conversion efficiency limit[2,6,12]. The reversible polarization current produced by the electric field modulation is responsible for this. The cyclic process comprises four steps: C→D and A→B are isoelectric, coincide with no differences in the electric field, and are used for heat addition and rejection. B→C and D→A are adiabatic processes, i.e., thermodynamic processes that occur without heat transfer for charging and discharging[38]. Pyroelectric materials can convert energy cyclically through switchable spontaneous polarization, producing work beyond the thermodynamic limit under pulsed illumination[39].

The discovery of light-emitting diodes (LEDs) based on group III/nitride semiconductors has transformed energy-efficient lighting[40]. These LEDs emit light in a narrow spectral range of 400–750 nm and operate in pulsed mode, typically between 50–200 Hz[41–43]. Importantly, a pyroelectric-photovoltaic device is required to capture the energy from LED illumination, which operates under pulsed illumination with frequency. Such a device is necessary for creating a widespread energy system for the Internet of Things, providing energy harvesting and exceptional photosensing capabilities. Additionally, for underwater exploration, a device must operate within the illumination wavelength range of 300–500 nm because longer visible and infrared wavelengths significantly attenuate in water. This device must be able

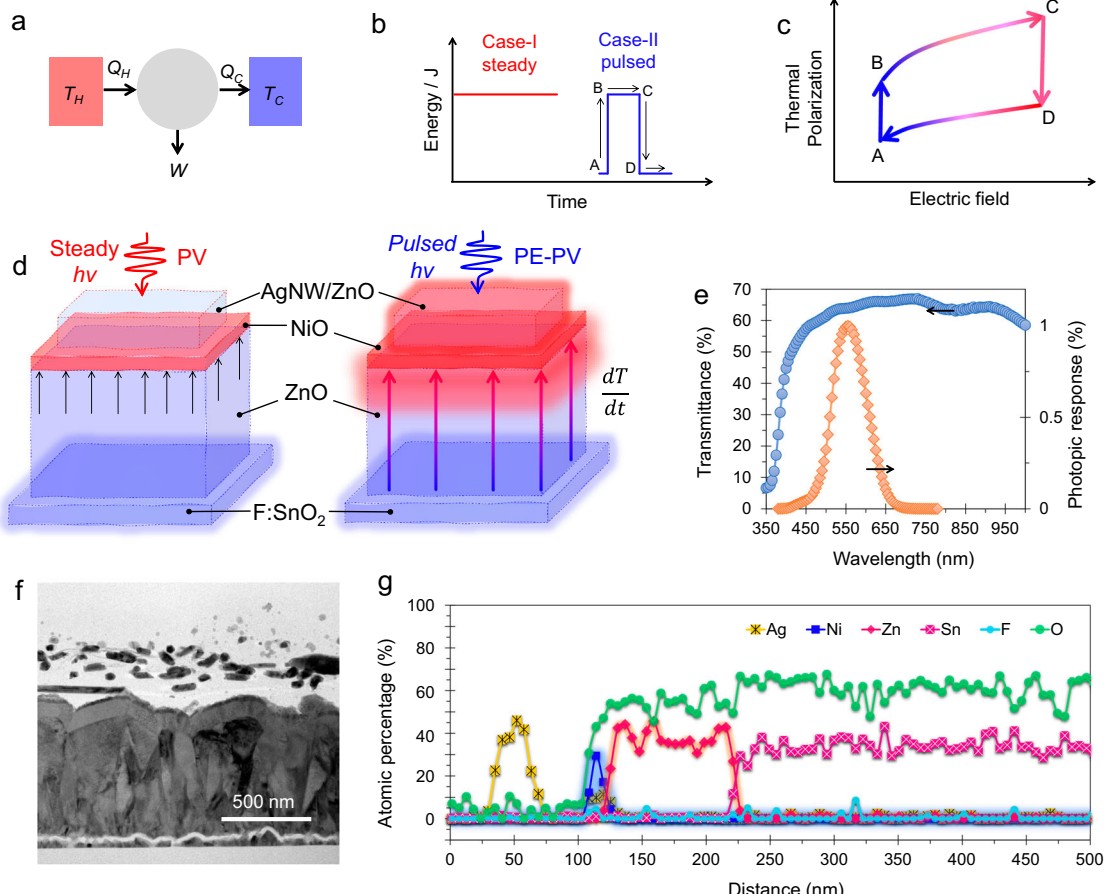

**Fig. 1 | Transparent pyroelectric heterojunction device (TPHD). a** Carnot engine diagram, where $Q_H$ (amount of heat) flows from a $T_H$ (higher temperature) of the working substance and the $Q_C$ (remaining heat) flows into the cold sink ($T_C$) to do work (W). **b** Steady (case I) and pulsed (case II) input energy for energy conversion. **c** Polarization versus electric field for transient signals illustrating the Brayton pyroelectric cycle. **d** Schematic depiction of TPHD, the built-in electric field at the ZnO/NiO heterojunction (photovoltaic-PV operation under steady illumination), and the long-range electric field function of dT/dt (pyroelectric-photovoltaic PE-PV operation under pulsed illumination). **e** Transmittance profile of the device and the photopic response. **f** Cross-section of the device, and **g** elemental line profile for Ag, Ni, Zn, Sn, F, and O.

to produce powerful and ultrasensitive photodetection to meet underwater needs of onsite power, UV and short visible wavelength photodetection, and photo-communication[44].

A photovoltaic heterostructure comprising a pyroelectric absorber can harness pyroelectricity, wherein energy is generated due to effects of the chemical bonds along the out-of-plane direction. Such a device can produce power above the thermodynamic limit and use pyroelectric polarization to regulate the electrostatic charge. In this study, photonic ionized defects were analyzed for pyroelectric-photovoltaic power production under pulsed light illumination in a ZnO/NiO heterostructure device. The proposed device displayed an exceptional pyroelectric coefficient, enabling a PCE and incident-photon-to-current conversion efficiency (IPCE) above the fundamental limit. The convergence of hybrid energy production from light for photovoltaic and photoelectric applications indicates the potential for the efficient use and integration of multiple energy sources (i.e., photonic and thermal energy).

## Results

### Pyroelectric-embedded heterojunction device and polarization *c*-axis orientation

ZnO is a pyroelectric material, and its n-type nature is suitable for forming a heterojunction with p-type NiO[11,15]. The pyroelectric-embedded ZnO/NiO device with F:SnO$_2$ (FTO) and silver nanowire (AgNW)/ZnO transparent electrodes is shown in Fig. 1d. The schematic

depicts a built-in electric field and temporal electric field caused by a temperature rate change. The device structure is transparent, as shown in Fig. 1e, due to the wide bandgaps and nanostructure, resulting in broadband transmittance suitable for photopic response. A transparent pyroelectric heterojunction device (TPHD) is created at room temperature using magnetron sputtering to form a pyroelectric ZnO layer and its heterojunction with NiO. We produced a ZnO film by RF sputtering of the ZnO target and NiO films by reactive DC sputtering of the Ni target. By controlling the O$_2$ gas flow rate to 5 sccm, we achieved a highly doped p-type and nanocrystalline NiO film suitable for forming a heterostructure with intrinsically n-doped ZnO[45]. The sputtering power played a crucial role in the process, with lower power leading to the formation of an amorphous film and higher power inducing lattice strain and atomic defects due to the higher kinetic energy of the sputtering materials[46,47]. We opted for 150 W of RF power to enable the formation of a dense film, which resulted in a better pyroelectric ZnO film. The transmission electron microscopy image of the device structure is shown in Fig. 1f, while Fig. 1g confirms the presence of Sn, Zn, Ni, O, Ag, and F in the device structure corresponding to SnO$_2$, ZnO, NiO, AgNWs, and ZnO. This result also confirms the thicknesses of the ZnO and NiO films, which are 100 ± 5 nm and 12 ± 5 nm, respectively. We observed that the amount of C and Si required for the elemental ratios increased to 100% (Supplementary Table 1). Because the thickness of the top ZnO layer is only approximately 12 nm, the EDS signal step resolution of >6 nm does not provide

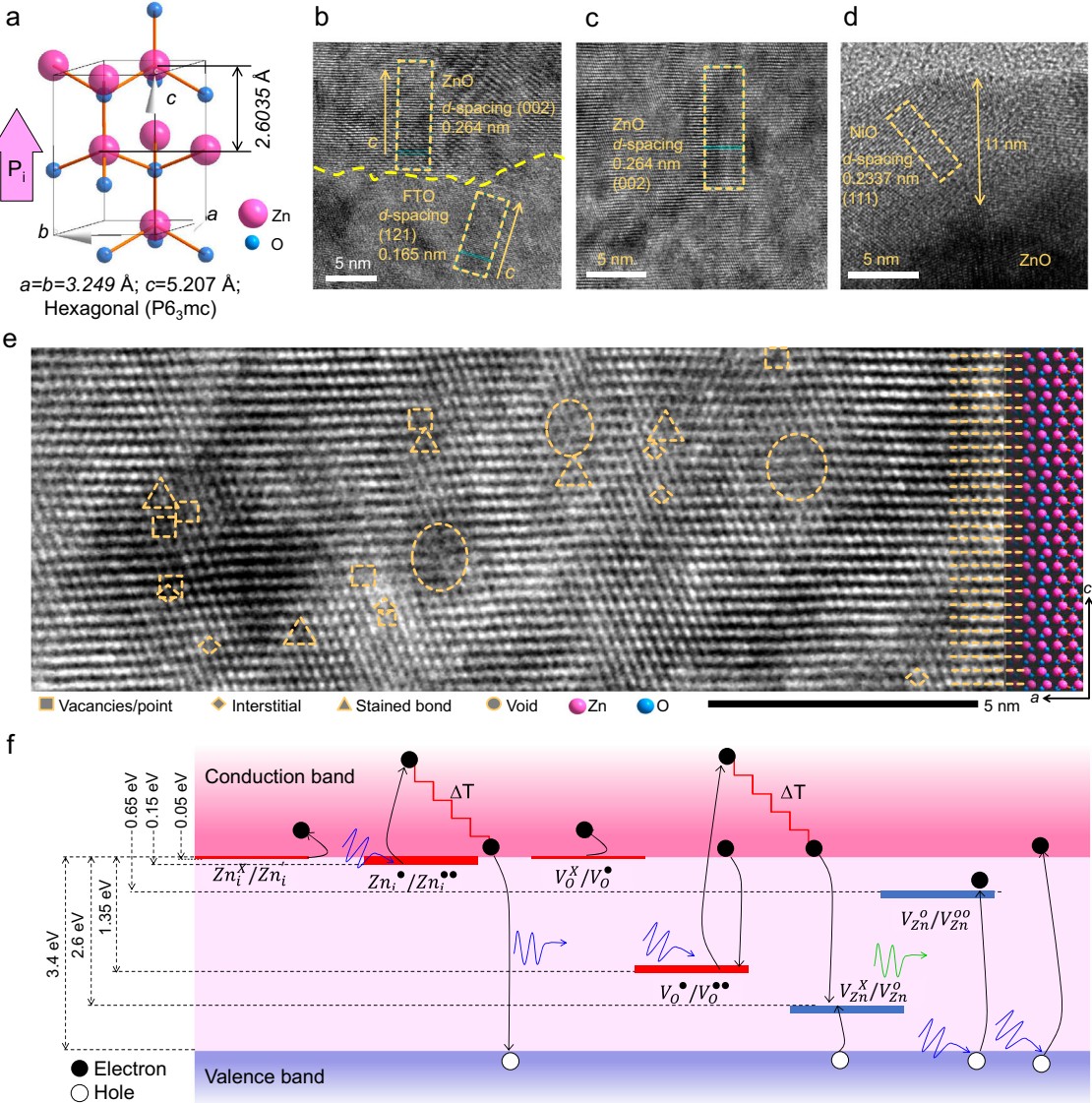

**Fig. 2 | Polarization axis and photonic defects. a** Crystal structure of hexagonal ZnO, its lattice parameters (**a**–**c**), and polarization along the *c*-axis, in which pink and blue spheres represent Zn and O atoms, respectively, and orange sticks represent covalent bonds. High-resolution image of the device showing the **b** FTO/ZnO, **c** ZnO film, and **d** ZnO/NiO interfaces. **e** High-angle annular dark-field scanning transmission electron microscope image of the ZnO, middle of the device, overlain with the ZnO crystal cells, showing atomic defects of vacancies/points, interstitials, stained bonds, and voids. **f** The energy band of ZnO depicts optical excitation, defect modulation, carrier generation, and recombination. The Kroger Vink notation is used for interstitial (i), zinc (Zn), oxygen (O), and vacancy (V) states.

a clear picture of the Zn concentration for the AgNW/ZnO region. Instead, the evidence confirms that the ZnO top layer thickness of 12 nm in Supplementary Fig. 1 depicts the distribution of Zn, O, and Ag.

Pyroelectric materials exhibit internal spontaneous polarization that causes the pyroelectric effect. This effect is temporary and occurs when the pyroelectric field undergoes spontaneous polarization due to a temperature gradient. The unit cell of wurtzite ZnO has a hexagonal structure that is highly anisotropic along and perpendicular to the *c*-axis[36], as shown in Fig. 2a. The structure, with hexagonal crystal symmetry P6₃mc, consists of a regular tetrahedron with a $Zn^{2+}$ cation at the center and adjacent $O^{2-}$ anions. When there is a thermal perturbation along the *c*-axis of the tetrahedron, the cation and anions are repelled from each other, creating a dipole moment. This dipole moment is the source of the large pyroelectric potential that results from the superposition of all the cells in the crystal[11–13,36].

The *c*-axis orientation of ZnO is presented in Fig. 2b, which shows a device interface of FTO/ZnO and *d*-spacings of 0.165 and 0.264 nm corresponding to the (121) and (002) planes of tetragonal-SnO₂ and

hexagonal-ZnO, respectively. The growth rates of the ZnO films are highly anisotropic, with the fastest growth occurring on the Zn-terminated polar surface[48–50]. Fig. 2c confirms that the ZnO film grows along the pyroelectric polarization *c*-axis. Additionally, Fig. 2d shows the ZnO/NiO interface, which has a *d*-spacing of 0.2337 nm, confirming the (111) plane of the cubic structure of the NiO film (with a thickness of 11 nm). The crystal structures of SnO₂, ZnO, and NiO are summarized in Supplementary Fig. 2. Additionally, the line profiles of the marked region in Fig. 2b–d are summarized in Supplementary Fig. 3, which depicts the estimated *d*-spacing over a length of 5 nm.

## Defects in ZnO and photon-induced ionization
In Fig. 2e, a high-resolution transmission electron microscope (HRTEM) image of the device is depicted using annular dark-field imaging. This technique enables clear resolution of the Zn and O atomic columns, with minimal delocalization (as indicated by color contrast) at the surface. By analyzing the ZnO unit cell array (which is available in the crystallographic open database, COD-9011662), the *c*-

axis orientation of the wurtzite structure was confirmed to have a linear relationship between the atomic column and the raw material. This analysis identified various types of defects, including vacancies, interstitials, strained bonds, and voids. Among these defects, vacancy defects are dominant.

The Kroger Vink notation is used for interstitial (i), zinc (Zn), oxygen (O), and vacancy (V) states. According to defect chemistry[51,52], the Frenkel reaction forms Zn interstitials:

$$Zn_{Zn} \overset{h\upsilon}{\Longleftrightarrow} Zn_i^X + V_{Zn}^X \tag{1}$$

The Schottky reactions form O vacancies:

$$O \overset{h\upsilon}{\Longleftrightarrow} V_{Zn}^X + V_O^X \tag{2}$$

Further ionization by photoexcitation led to four donor-type reactions:

$$Zn_i^X \overset{0.05\,eV}{\Longleftrightarrow} Zn_i^+ + e^{\cdot}; \tag{3}$$

$$Zn_i^+ \overset{0.15\,eV}{\Longleftrightarrow} Zn_i^{++} + e^{\cdot}; \tag{4}$$

$$V_O^X \overset{0.05\,eV}{\Longleftrightarrow} V_O^+ + e^{\cdot}; \tag{5}$$

$$V_O^+ \overset{0.15\,eV}{\Longleftrightarrow} V_O^{++} + e^{\cdot}; \tag{6}$$

In addition, a two acceptor-type reaction occurs:

$$V_{Zn}^X \overset{0.8\,eV}{\Longleftrightarrow} V_{Zn}^- + h^{\circ}; \tag{7}$$

$$V_{Zn}^- \overset{2.8\,eV}{\Longleftrightarrow} V_{Zn}^{--} + h^{\circ}; \tag{8}$$

Among these materials, $Zn_i^{++}$, $Zn_i^+$, $Zn_i^X$, $V_O^{++}$, $V_O^+$ and $V_O$ are donor-type defects, while $V_{Zn}^{\circ\circ}$ and $V_{Zn}^{\circ}$ are acceptor-type defects. The ionization energy of Zn interstitials and oxygen vacancies varies between 0.05 and 2.8 eV[51–53]. These are the predominant ionic defects in the ZnO film, as confirmed by the photoluminescence spectra, as shown in Supplementary Fig. 4. This result indicates the presence of a higher order of defect ionized by a photon energy ($h\upsilon$) of 3.493 eV, which also appears in the ZnO/NiO heterostructure. Figure 2f shows the respective locations of the photonic ionization defects and illustrates the thermal energy induction processes. Notably, a photon-triggered ionized defect induces a larger amount of thermal energy in the lattice through thermalization in the corresponding band, which can be estimated from the excess energy in the conduction band. The energy band diagram shows that an enormous amount of energy in the range of 0.1–0.5 eV thermally dissipates under steady-state conditions[6,37].

## Heterojunction array and its pyroelectric-photovoltaic performance

Figure 3a shows a device array design comprising sixteen ZnO/NiO devices. The unit cell represents the cathode (FTO), while the circular anode of AgNW/ZnO (top electrode) defines the working area of the device. The large-area and room-temperature fabrication process enables a state-of-the-art heterojunction device array and prototype unit, as shown in Fig. 3b.

Pulsed illumination induces spontaneous polarization and is used to investigate the pyroelectric-photovoltaic performances of all the array devices. Thus, pulsed illumination at a frequency of 60 Hz, a wavelength of 365 nm, and an intensity ($P_{h\upsilon}$) of 500 μW cm$^{-2}$ was initially applied, and the resulting current density-voltage (J-V) plot was recorded with a voltage scan speed of 0.35 V s$^{-1}$ and a sample interval of 3 μV. Figure 3c shows this characteristic J-V plot of the unit cell device, in which the spikes correspond to spontaneous polarization in the illuminated and dark states. This result shows the diode nature of the heterojunction for the steady dark state, photovoltaic (PV) nature for the steady illumination state, and pyroelectric-photovoltaic (PE-PV) nature for the upon-light illumination state; therefore, the PV and PE-PV segments are marked. Furthermore, the power density-voltage (P-V) characteristic plot shown in Fig. 3d confirms that the maximum power (P$_{max}$) generated by the steady illumination of the photovoltaic system (blue peak) is greatly enhanced by the pyroelectric-photovoltaic effect upon illumination (red-peak). The significant effect of spontaneous polarization on the J-V and P-V characteristics and their performance consistency, as summarized in Supplementary Figs. 5, 6, confirms the outstanding power harvesting ability of the pyroelectric absorber-embedded heterojunction device.

To evaluate the photon-to-electric conversion performance, it is necessary to summarize the key parameters of sixteen devices in the array, such as the short-circuit current density (J$_{SC}$), open-circuit voltage (V$_{OC}$), P$_{max}$, fill factor (FF), and PCE. Figure 3e shows the J$_{SC}$ values of all devices, which significantly improved from 113 to 262 μA cm$^{-2}$, a 2.32-fold increase. Similarly, Fig. 3f shows that the V$_{OC}$ values of the devices remarkably improved from 0.285 to 0.380 V. The P$_{max}$ values, as shown in Fig. 3g, were collected from the P-V plots of Supplementary Fig. 6, indicating a 4.06-fold increase from 12.45 to 50.6 μW cm$^{-2}$. Additionally, the maximum current (J$_m$) and voltage (V$_m$) are also provided in Supplementary Figs. 5, 6.

Then, the FF of the device can be calculated by FF = (J$_m$ × V$_m$)/(J$_{SC}$ × V$_{OC}$), as shown in Fig. 3h. This calculation resulted in a more equitable value from 47.8 to 56.34%. PCE can then be estimated by PCE = (J$_{SC}$ × V$_{OC}$ × FF)/$P_{h\upsilon}$ = P$_{max}$/$P_{h\upsilon}$, as summarized in Fig. 3i. The PCE values demonstrate a 3.848-fold increase, from 3.28 to 12.624%, in the energy conversion process by the PE-PV.

Spontaneous polarization significantly increases the photocurrent. This effect is analyzed beyond the thermodynamic limits using the IPCE relationship as follows:

$$IPCE = \frac{electrons/cm^2/s}{photons/cm^2/s} = \frac{J_{SC}(\mu A\,cm^{-2}) \times 1239.8(V\,nm)}{P_{h\upsilon}(\mu W\,cm^{-2}) \times \lambda(nm)} \tag{9}$$

According to the thermodynamic limit, the maximum IPCE of a standard photovoltaic device should not exceed 100%[54,55]. The IPCE depends on the charges generated by the light, how they recombine, and how they are transported through the device. However, the ZnO/NiO device shows a significant increase in J$_{SC}$ due to spontaneous polarization. This enhancement leads to a significant improvement in the IPCE when the device is pulsed with light. In Fig. 3j, the IPCE of the device ranges from 77% to 177%, indicating highly sensitive performance beyond the thermodynamic limit.

We conducted photon balance calculations to determine the IPCE values for the PV and PE-PV technologies. In these calculations, we assumed that the absorption of each photon results in an electron-hole pair, which generates a photocurrent when collected. Under perfect conditions where there is complete absorption, no reflection, perfect electron-hole pair generation, no parasitic electrical losses, and no recombination, the IPCE should reach 100%. Using this assumption and an estimated photon density, we can calculate the photocurrent density for the zero-bias condition, which is also known as the J$_{SC}$. We estimated the photon density using the formula given below:

$$Photon\,density = \frac{I}{E_P} = \frac{I \times \lambda \times 10^{-9}}{hc} = \frac{I(W\,m^{-2}) \times \lambda \times 10^{-9}(m\,s)}{1.988 \times 10^{-25}(J\,s\,m)} \tag{10}$$
$$= I \times \lambda \times 5.03 \times 10^{15}(m^{-2}\,s^{-1}).$$

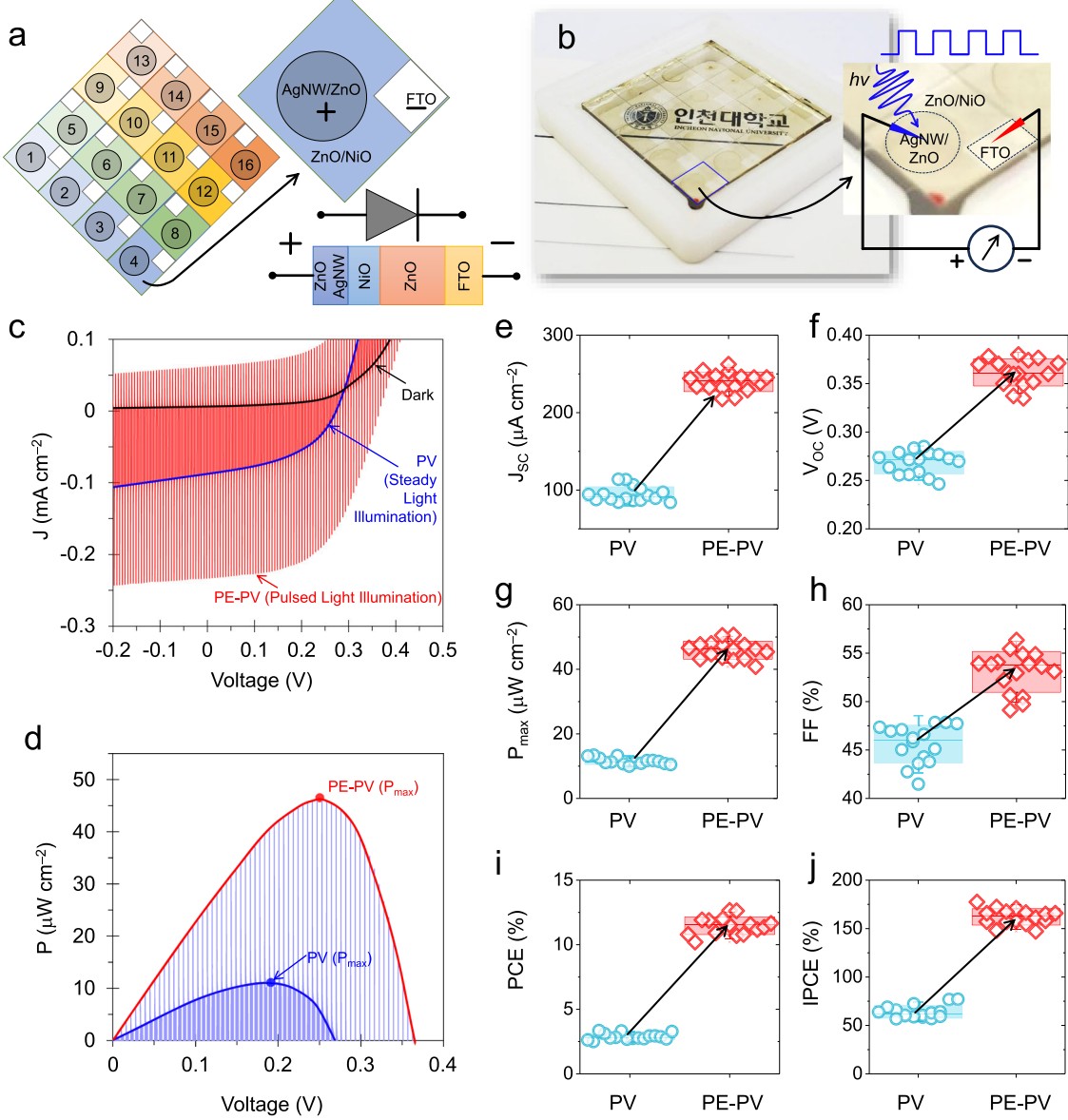

**Fig. 3 | Characteristic performance distribution under the illumination pulse.** **a** Schematic depiction of a TPHD unit cell and its array with a total of sixteen devices (4 × 4). **b** Original prototype TPHD array; the right panel shows the unit cell device and its electrical connection under pulsed light illumination. **c** Current-voltage characteristics and **d** power-voltage characteristics of the device under pulsed light illumination. Performance distribution of a total of sixteen devices showing the **e** short-circuit current density ($J_{SC}$) in µA cm⁻², **f** open-circuit voltage ($V_{OC}$) in V, **g** maximum power density ($P_{max}$) in µW cm⁻², **h** fill factor (FF) in %, **i** power conversion efficiency (PCE) in %, and **j** incident-photon-to-current conversion efficiency (IPCE) in %. PV and PE-PV represent photovoltaic and pyroelectric-photovoltaic operations, respectively.

where $\lambda$ and $I$ are the illumination wavelengths in m and intensity in W m⁻², respectively; $h$ is the Planck constant ($6.63 \times 10^{-34}$ J s); and $c$ is the speed of light ($2.998 \times 10^{8}$ m s⁻¹).

First, we confirmed the accuracy of the IPCE values estimated from the PV segment, which should match the calculated $J_{SC}$ value and balance the photons. The IPCE performances for the TPHD array devices are shown in Fig. 3j. We estimated these values for PV and PE-PV to be 77% and 177%, respectively, from the $J_{SC}$ values measured by the J-V characteristic plots under 365 nm wavelength and 500 µW cm⁻² illumination. This illumination provides a photon density of $9.17975 \times 10^{14}$ cm⁻² s⁻¹. Under ideal conditions, the PV operation provides a $J_{SC}$ value of 147.076 µA cm⁻², which serves as a check value for the measured PV performances. Table 2 summarizes the detailed calculations for the photon flux, calculated $J_{SC}$, measured $J_{SC}$ of the sixteen array devices, and their calculated IPCE values. Our analysis revealed that the measured $J_{SC}$ value

for PV performance complies with the theoretically calculated $J_{SC}$ value. The average value is 94.58 µA cm⁻², resulting in an IPCE of 64.25%. However, due to spontaneous polarization by the pyroelectric-photovoltaic device, the measured $J_{SC}$ value for the PE-PV performances exceeded the calculated $J_{SC}$ value, with an average value of 239.6 µA cm⁻², equivalent to an IPCE value of 162.77%. This analysis showed that the charge density that participated in the spontaneous polarization by the PE-PV phenomenon was $9.05 \times 10^{14}$ cm⁻² s⁻¹. This value is close to the photon flux and suggests that photoionized impurities participate in spontaneous polarization via the PE–PV phenomenon.

An experiment was conducted to evaluate the importance of the shape of the working region. Various shapes of the top electrodes, namely, circular, square, rectangular, and pentagon, were examined, and linear arrays of each shape's active area of 10 mm² with a standard

**Table 2 | Summary of the detailed photon balance calculations and measured short-circuit current density ($J_{SC}$) and incident photon-to-current conversion efficiency (IPCE) values for transparent pyroelectric heterojunction devices**

| Photon density | Calculated $J_{SC}$ | Device | Measured $J_{SC}$ ($\mu A\ cm^{-2}$) | | Measured IPCE (%) | | Additional charges by pyroelectric ($cm^{-2}\ s^{-1}$) |
|---|---|---|---|---|---|---|---|
| | | | PV | PE-PV | PV | PE-PV | |
| $9.17975 \times 10^{14}\ cm^{-2}\ s^{-1}$ | $147.076\ \mu A\ cm^{-2}$ | D1 | 86.55 | 233.33 | 58.80 | 158.51 | $9.16 \times 10^{14}$ |
| | | D2 | 89.18 | 252.76 | 60.58 | 171.71 | $1.02 \times 10^{15}$ |
| | | D3 | 87.08 | 231.94 | 59.16 | 157.57 | $9.04 \times 10^{14}$ |
| | | D4 | 107.00 | 229.09 | 72.69 | 155.63 | $7.62 \times 10^{14}$ |
| | | D5 | 84.32 | 218.88 | 57.28 | 148.69 | $8.40 \times 10^{14}$ |
| | | D6 | 93.15 | 217.69 | 63.28 | 147.89 | $7.77 \times 10^{14}$ |
| | | D7 | 88.65 | 232.39 | 60.22 | 157.87 | $8.97 \times 10^{14}$ |
| | | D8 | 89.49 | 247.58 | 60.79 | 168.19 | $9.87 \times 10^{14}$ |
| | | D9 | 94.94 | 246.25 | 64.50 | 167.29 | $9.44 \times 10^{14}$ |
| | | D10 | 97.35 | 254.72 | 66.13 | 173.04 | $9.82 \times 10^{14}$ |
| | | D11 | 87.77 | 238.28 | 59.63 | 161.87 | $9.39 \times 10^{14}$ |
| | | D12 | 113.57 | 245.23 | 77.15 | 166.60 | $8.22 \times 10^{14}$ |
| | | D13 | 101.78 | 233.61 | 69.14 | 158.70 | $8.23 \times 10^{14}$ |
| | | D14 | 84.00 | 244.14 | 57.06 | 165.85 | $1.00 \times 10^{15}$ |
| | | D15 | 113.97 | 262.45 | 77.42 | 178.29 | $9.27 \times 10^{14}$ |
| | | D16 | 94.45 | 245.29 | 64.16 | 166.64 | $9.41 \times 10^{14}$ |
| | | **Average** | **94.58** | **239.60** | **64.25** | **162.77** | **$9.05 \times 10^{14}$** |

deviation error of only 1% were used (Supplementary Fig. 7). The results showed consistent PE–PV power generation from devices with different electrode shapes. Moreover, the PCE and IPCE were improved by PE-PV, particularly when the IPCE > 100%, as summarized in Supplementary Table 2. This summary indicates that there is no requirement for the working region's shape to achieve PE-PV performance.

To further investigate the PE-PV and PV components of the device, a transparent device with ZnO/NiO and TiO$_2$/NiO was fabricated and measured under identical conditions, as shown in Supplementary Fig. 8. In this device, anatase-TiO$_2$ was used because of its centrosymmetric nature with n-type conduction, which is suitable for providing photovoltaic characteristics with a NiO layer[56]. Under identical conditions of 50% duty cycles and pulsed illumination at 60 Hz, their J-V and P-V characteristics demonstrate the distinguishing nature of centrosymmetric and noncentrosymmetric absorbers. This result clarifies the PE-PV power generation by the pyroelectric-absorber embedded heterojunction.

**Pyroelectric-photovoltaic heterojunction device mechanism**

Figure 4a shows the energy band diagram of the device, including the defect states of $V_{Zn}$ and $V_O$, to reveal the underlying operational mechanism. Photonic acceptor defects at 0.8 eV and 2.8 eV are represented by $V'_{Zn}/V^X_{Zn}$ and $V''_{Zn}/V'_{Zn}$, respectively, and donor defects at 1 eV and 3 eV are represented by $V_O/V'_O$ and $V'_O/V^X_O$, respectively[52]. As shown, heterojunctions have drift and diffusion fields for charge transport, where a built-in electric field ($E_i$) forms a built-in potential that is responsible for the PV performance under steady illumination. In the illuminated and dark states, pyroelectric-embedded heterojunctions include the addition of a pyroelectric field, which should be long-range, spontaneous, and switchable based on changes in the thermal state.

Photonic defects play an active role in spontaneous polarization to reconfigure junctions. Equivalent circuits can reveal the underlying charge carrier dynamics, as depicted in Fig. 4b. Analysis of the data shown in Fig. 4b revealed a total of four states, where the current flow in the device under the steady-dark state, largely represented by the diode current, presented by $J = J_D$, was approximately 0.8 $\mu A\ cm^{-2}$.

Upon light illumination, there are two additional currents in the opposite direction of $J_D$. Prior is the current due to spontaneous polarization by pyroelectric energy ($J_{PEL}$), and the latter is the $J_{SC}$ due to its photovoltaic nature; as a result, $J = J_D - J_{SC} - J_{PEL}$. Therefore, in response to light illumination, the device exhibited a large current surge along the $J_{SC}$.

By providing steady illumination, the thermal state diffuses and eliminates the pyroelectric current from the device. This operation enables the device to return to its conventional PV operation, where $J = J_D - J_{SC}$. Up to the dark state, the device undergoes a thermal process that causes spontaneous polarization opposite to the previous pyroelectric effect (polarization from hot to cold). This effect leads to a large current surge toward $J_D$, resulting in a net current $J = J_D + J_{PED}$, where $J_{PED}$ represents the pyroelectric polarization current in the dark state. Importantly, when exposed to light or in the dark, a pyroelectric field is generated due to the pyroelectric ZnO. This pyroelectric ZnO overtakes diffusion transport by drift transport during spontaneous polarization, as illustrated in Supplementary Fig. 9, which is crucial for the development of ballistic-like transport devices by pyroelectric-photovoltaics[57].

Due to the balance of these four states, pyroelectric-embedded heterojunctions offer highly durable energy cycles, high PCE, and reconfigurable electrostatic properties that help to achieve performances above the thermodynamic limit. Using the current and voltage of the PV device, enhancement of the PE-PV performance can be confirmed. The $V_{OC}$ of the PV device is expressed by:

$$V_{OC} = \frac{nkT}{q} \times \ln\left(\frac{J_{SC}}{J_D}\right) \tag{11}$$

where $n$, $kT/q$, $J_{SC}$, and $J_D$ denote the diode ideality factor, thermal voltage, short-circuit current, and diode current, respectively[58]. Considering the $n = 2$ of the ZnO/NiO device, this relation confirms a $V_{OC}$ of 0.275 V for a steady illumination of 0.5 mW cm$^{-2}$, a $J_D$ of 0.8 $\mu A$ cm$^{-2}$, and a $J_{SC}$ of 100 $\mu A$ cm$^{-2}$. Upon illumination, an additional polarization current ($J_{PL}$), resulting in a $J_{SC}$ of 250 $\mu A$ cm$^{-2}$, increased the $V_{OC}$ to 0.327 V.

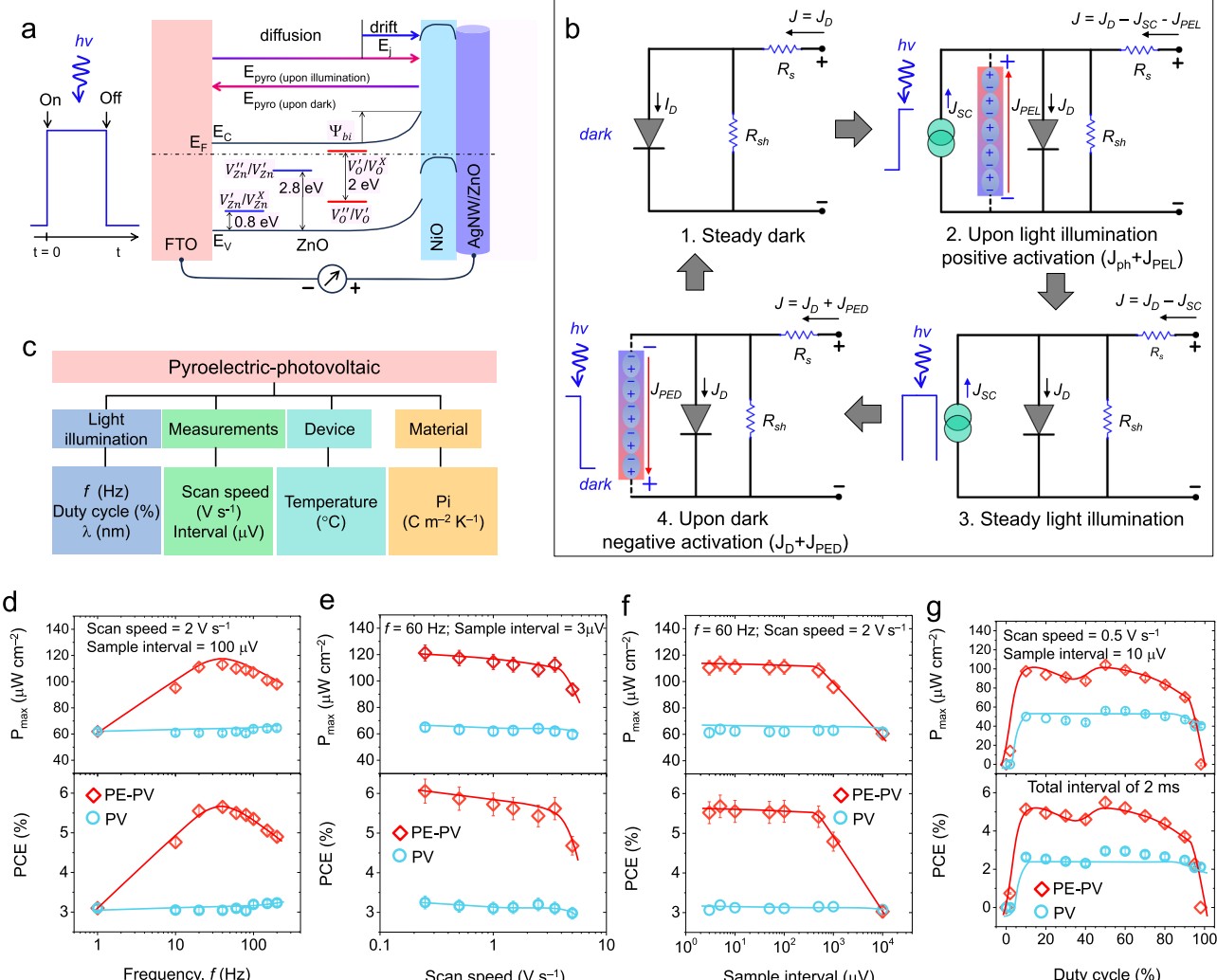

**Fig. 4 | Influential parameters required to increase the pyroelectric-photovoltaic power. a** Energy band diagram of the device depicting the energy states of Zn and O vacancies ($V'_{Zn}/V^X_{Zn}$, $V''_{Zn}/V'_{Zn}$, $V'_O/V^X_O$, $V''_O/V'_O$), drift-diffusion segments of the heterojunction, built-in electric field ($\Psi_{bi}$) and long-range pyro-electric field ($E_{pyro}$) under pulsed illumination. $E_C$, $E_V$, and $E_F$ represent the conduction band, valence band and Fermi energy levels, respectively. **b** Equivalent electrical circuits depicting transparent pyroelectric heterojunction device operation under steady darkness, upon illumination, under steady-light illumination, and in the dark state manifest photovoltaic and pyro-photovoltaic power reaping, where J, $J_D$, $J_{SC}$, $J_{PL}$, $J_{PD}$, $R_s$, and $R_{SH}$ are the total current, diode current, short-circuit current, pyroelectric current upon light illumination, pyroelectric current in the dark, series resistance, and shunt resistance of the device, respectively. **c** Influential

parameters of pulsed frequency (f) of light illumination, duty cycle (D), voltage scan speed (V s⁻¹), sample interval (μV), light wavelength (λ), device temperature (T), and pyro-coefficient (Pi). **d** $P_{max}$ and PCE versus pulsed illumination frequency, f in Hz when the scan speed is 2 V s⁻¹ and the sample interval is 100 μV (error bar is 2.5%). **e** $P_{max}$ and PCE versus scan speed in V s⁻¹ when the pulsed illumination rate 'f' is 60 Hz and the sample interval is 3 μV (error bar is 5%). **f** $P_{max}$ and PCE versus sample interval at μV for a pulsed illumination rate of 60 Hz and a scan speed of 2 V s⁻¹ (error bar is 5%). **g** $P_{max}$ and PCE versus duty cycle in % when the pulsed illumination f is 50 Hz, the scan speed is 0.5 V s⁻¹, and the sample interval is 10 μV (error bar is 2.5%). The illumination intensity was 2 mW cm⁻² during the frequency, scan speed, and sample interval studies.

## Influential parameters for harvesting pyroelectric-photovoltaic compounds from spontaneous polarization

When embedded in a pyroelectric material, a heterostructure device can convert pulsed illumination into spontaneous polarization and therefore improve the PCE with a fast response time. To better understand this phenomenon, we consider the influential parameters summarized in Fig. 4c, which include the pulse illumination frequency (f), voltage scan speed (ω), step interval (Δν) and duty cycle (D). Analyzing these parameters can help identify their impact on the PE-PV performance.

The pyroelectric current, which occurs due to spontaneous polarization upon illumination, can be defined as $J_{PL}(f, ω, Δν, D)$ and should correspond to the spontaneous polarization of the upon illumination state. To match the spontaneous polarization by pulse

illumination, the first step for investigating this current was to test the response of sweeping f from 1 to 200 Hz with D = 50% for ω at 2 V s⁻¹ and a Δν of 100 μV. Figure 4d illustrates the corresponding relationship between $P_{max}$ and the PCE as a function of f in Hz (obtained from the results summarized in Supplementary Figs. 10–12), indicating its crucial role. Both parameters reach their peak value at approximately 60 Hz, which is consistent with the conventional frequency used for energy harvesting via PE-PV-photovoltaic performance, suggesting a possible match.

The second step in this experiment involved testing ω at various values ranging from 0.2 to 5 V s⁻¹ to the optimal f at 60 Hz and a Δν of 3 μV. The results are shown in Fig. 4e, which summarizes the performances of $P_{max}$ and the PCE as a function of ω in V s⁻¹ (obtained from the results summarized in Supplementary Figs. 13–15). This result

indicates that $\omega$ is less effective and should be <1 V s$^{-1}$ for this setup. Both parameters reach their peak values at 0.2 V s$^{-1}$, which suggests that a good number of cycles of pyroelectric-photovoltaic performances were acquired.

The third step of this experiment involved testing various values of $\Delta v$ (ranging from 3 to 10$^4$ μV) using the optimal values of $f$ and $\omega$. The results of this test are shown in Fig. 4f (obtained from the results summarized in Supplementary Figs. 16–18) and indicate that a large $\Delta v$ value leads to a loss of performance in pyroelectric-photovoltaic enhanced photovoltaic PV. Therefore, it is recommended that $\Delta v$ be kept below 100 μV for this setup to capture the power cycles by pulsed illumination. Considering the three investigated parameters, finding the optimum pulse frequency for the maximum P$_{max}$ and PCE (approximately 60 Hz) when $\omega$ is less than 1 V s$^{-1}$ and $\Delta v$ is less than 10 μV is shown to be a favorable approach.

To generate the PE-PV effect, pulsed illumination is needed. To evaluate the associated energy conversion, we varied the duty cycles by changing the pulse width of the light-on interval within a fixed total signal period of 20 ms, as shown in Supplementary Fig. 19. At an intensity of 1.9 mW cm$^{-2}$ and a wavelength of 365 nm, we achieved a photon density of 3.488 × 10$^{15}$ cm$^{-2}$ s$^{-1}$. By controlling the duty cycle, we maintained the photon dose within a range of 1.395 × 10$^{12}$ to 6.977 × 10$^{13}$ cm$^{-2}$. For 100% IPCE, the expected J$_{SC}$ ranges from 55.88 μA cm$^{-2}$ for a 10% duty cycle to 503.89 μA cm$^{-2}$ for a 90% duty cycle.

The J-V characteristics of the TPHD device for different duty cycles are shown in Supplementary Fig. 20. A duty cycle range of 10–90% enables PE-PV energy conversion, which distinguishes the PV and PE-PV components. The P–V characteristic plots of the duty cycle series are summarized in Supplementary Fig. 21, where the output power is maximized at for a duty cycle 50% for the PE-PV segment and remains consistent for the PV segment for a duty cycle range of 20% to 98%.

The duty cycle is responsible for controlling the amount of photons that are delivered during a fixed interval of the pulse cycle (which is typically 2 ms). During the light-on pulse region, the absorbed photons activate the Frenkel reaction of the Zn interstitials, inducing ionized donor-type defects ($Zn_i^+$ and $Zn_i^{++}$) with activation energies of 50 meV and 150 meV[52]. This likely occurs to induce a temperature change and cause the polarization of these ionized defects, providing pyroelectric components to the photovoltaic system under pulsed illumination. The resulting photon dose triggers the polarization current, and the effect is more prominent for a duty cycle of 2%, which is equivalent to a photon dose of 1.395 × 10$^{12}$ cm$^{-2}$. Based on the obtained results, it can be concluded that the duty cycle can serve as an indicator for evaluating the PE-PV and PV performance parameters as a function of the duty cycle. The data show that a lower photon dose provides greater IPCE performance via the PE-PV. This dynamic is attributed to the inevitable Frenkel reaction of interstitial Zn, which is activated by pulsed photon illumination, and the ionized species cause a polarization current.

During the fourth step of the experiment, duty cycles ranging from 0 to 100% were tested using the optimal values of $f$, $\omega$, and $\Delta v$. The results shown in Fig. 4g (which are obtained from the results summarized in Supplementary Figs. 20–22) indicate that a $D$ value between 10–90% provides consistent performance for PE-PV operation. The power generation ability of the TPHD device is lost if $D$ is less than 5% for the given parameters. Therefore, it is recommended that $D$ be kept at approximately 50% for this setup to capture the power cycles by pulsed illumination. Considering the four parameters, determining the optimum pulse frequency for the maximum P$_{max}$ and PCE (approximately 60 Hz with $D$ of approximately 50%) when $\omega$ is less than 1 V s$^{-1}$ and $\Delta v$ is less than 10 μV is demonstrated to be a favorable approach.

Based on the optimal parameters of $\omega$ and $\Delta v$, the power produced by pyroelectric-photovoltaic devices is concluded to be dependent on the frequency of pulsed illumination. During pulsed illumination, the polarization of ZnO crystals changes from positive to negative by the flipping of noncentrosymmetric polar symmetry (P6$_3$/mc) through centrosymmetric symmetry (P6$_3$/mmc)[11,14,36]. This dynamic crystal symmetry change leads to an optimal ionic current along the photovoltaic system when the pulsed frequency is matched. For maximum power conversion, first-order pyroelectric polarization switching by relative atom movements of Zn and O should be matched to the relaxation frequency of Born-effective charges, which is approximately 60 Hz. At such a rate, rigid-ion displacements by pulse illumination may also have a dimensionality effect on the PE-PV performance[11].

We evaluated five device designs for their PV and PE-PV performances. To achieve this goal, we further clarified the design of the ZnO/NiO heterojunction and stacking of the AgNW/ZnO heterostructure using optimum evaluation indicators (Supplementary Fig. 23). The ZnO and NiO layers had thicknesses of 100 nm and 20 nm, respectively. The AgNW/ZnO composite had identical conditions, with uniform drop-casting of AgNWs and a low-power-sputtered ZnO layer thickness of 12.5 nm. We observed pyroelectric performance with energy harvesting by only a ZnO device with a PCE of 0.06%, which is solely governed by pyroelectric phenomena, while PV phenomena are extremely difficult to observe. A device with only a NiO layer lacks a photoactivated response and characteristic plot and has an ohmic response to the illumination. The ZnO/NiO heterojunction without a AgNW electrode shows a PV response with a V$_{OC}$ of 0.579 V, which increases to 0.6 V due to the PE-PV phenomenon. However, the J$_{SC}$ of 4.43-4.6 μA cm$^{-2}$ resulted in a PCE of 0.046%, as the collection of charges was significantly poor due to the lack of a suitable electrode. With a suitable AgNW electrode, the ZnO/NiO heterojunction offers a noticeable PCE of 6.97%. However, the AgNW network is susceptible to thermal stimuli and easily deforms under thermal stress, weakening its electrical conduction[59]. Overcoating a ZnO layer onto a AgNW network provides outstanding stability with improved optical and electrical properties. As a result, for the ZnO/NiO heterojunction device with the AgNW/ZnO heterostructure, PE-PV phenomena with greater PCE and IPCE performance persist.

## Spectral and thermal characteristics

To ensure the optimal performance of a PV device, it is important to consider both spectral and thermal attributes. This consideration is important because the short wavelength of illumination can affect the PCE of a PV device due to thermalization. We applied the optimal values of $f$, $\omega$, $\Delta v$, and $D$ to examine the spectral attributes of PE-PV performance. Additionally, understanding how illumination wavelength regulates PE-PV performance is crucial to comprehending the role of various optical excitations. As such, we varied the pulsed illumination wavelength from 340–850 nm using a monochromator to examine the PE-PV performance by the J-V and P-V characteristics. For more information, see Supplementary Fig. 24 for the device under test and Supplementary Figs. 25 and 26.

Figure 5a shows the PCE values versus illumination wavelength. The results confirmed that PE-PV power generation occurred at wavelengths ranging from 340–460 nm. However, power generation was solely from PV for wavelengths of 520–625 nm. The highest PCE value was 12% at an illumination wavelength of 365 nm and an intensity of 340 μW cm$^{-2}$. For the PV performance, the PCE value of 3% was consistent for wavelengths of 340–410 nm. This can be attributed to the excitons in ZnO. The results suggest that pulsed illumination at wavelengths of approximately 365 nm induced a prominent thermal change, which caused an extraordinary current via spontaneous polarization of ionized defects along the c-axis, such as V$_{Zn}$ and V$_O$. This result shows that the PE-PV-induced IPCE was greater than 100% for an illumination wavelength of 340–410 nm, with a peak value of 216.5% for 365 nm (Fig. 5b). However, the PV-induced IPCE performances ranged from 70–80%. The PE–PV performance parameters

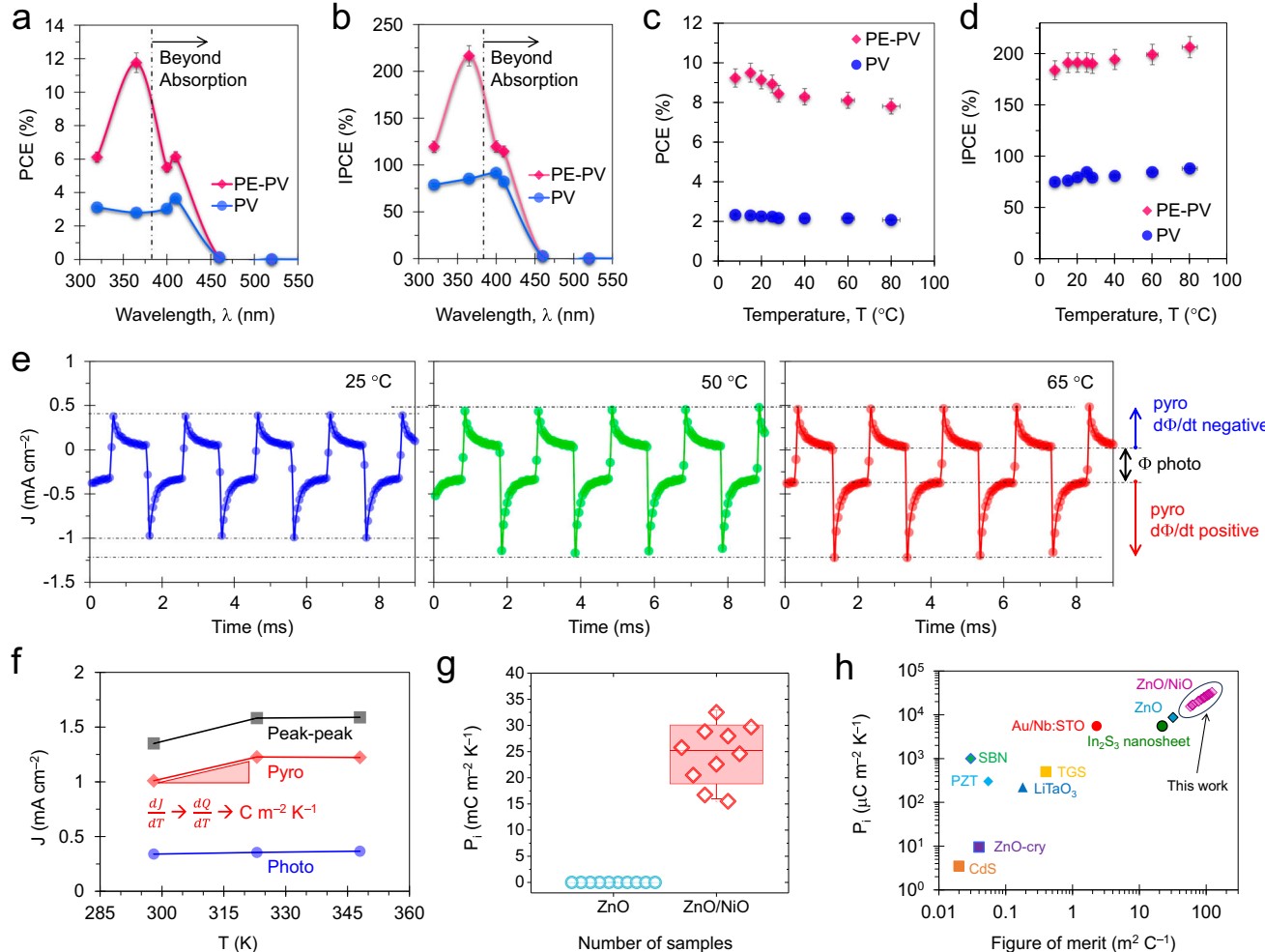

**Fig. 5 | Photovoltaic spectral characteristics and pyroelectric coefficient.**
Spectral characteristic performance, **a** power conversion efficiency (PCE) and
**b** incident-photon-to-current conversion efficiency (IPCE) versus illumination
wavelength (error bar is 5%). The device performance as a function of temperature,
**c** PCE, and **d** IPCE versus temperature T at °C for an illumination wavelength of
365 nm and an intensity of 100 μW cm⁻² (error bar is 5%). During the spectral and
thermal tests, the illumination f, scan speed, and sample interval were 60 Hz,
0.5 V s⁻¹, and 3μV, respectively. **e** Pyro-photoresponse of the device at 25, 50, and
65 °C for an illumination wavelength of 365 nm, intensity of 1.5 mW cm⁻², and pulse
frequency of 1000 Hz. **f** Polarization current for the peak − peak, photo, and pyro
intervals. **g** Pyroelectric coefficient, $P_i$ distribution of ZnO and ZnO/NiO.
**h** Comparison of the pyroelectric coefficients and figures of merit of the devices
with those of conventional and emerging pyroelectric materials. (The pyroelectric
coefficient and FOM data for conventional, bulk, and emerging materials are taken
from ref. [10],[11]).

versus wavelength are summarized in Supplementary Fig. 27. Due to
the large increase in pyroelectricity $P_{max}$ caused by pulse illumination,
the performances of the PCE and IPCE exceeded the
Shockley–Queisser limit[6,58,60–62].

To further elucidate the pyroelectric properties, the device per-
formance was profiled across a broad range of wavelengths of light.
The ability of the pyroelectric-photovoltaic devices to harvest energy
beyond the absorption region is related to the bandgap of ZnO, which
is 3.25 eV. According to the formula $E = \frac{h\upsilon}{\lambda}$, where $E$ is the photon
energy, $h$ is Planck's constant ($6.626 \times 10^{-34}$ J s), $c$ is the speed of light
($3.0 \times 10^8$ m s⁻¹), and $\lambda$ is the wavelength of light, ZnO has a photon
wavelength absorption cutoff value of 382.26 nm ($\lambda(nm) = \frac{1242.37}{E_g(eV)}$). Due
to the pyroelectric characteristics of the device, the ZnO/NiO hetero-
junction can function in a broad spectrum beyond absorption wave-
lengths, such as 400, 410, 460, 520, and 625 nm. The threshold line
corresponding to beyond-photon absorption utilization in Fig. 5a, b
highlights the crucial role of the pyroelectric absorber. This result
clearly demonstrated that pyroelectric energy harvesting is possible
for longer wavelengths beyond the photon absorption limit. Thus, this

approach has the potential to harvest energy via pyroelectric utiliza-
tion beyond the photon absorption limit.

Thermal stress further confirmed the existence and magnitude of
the PE-PV effect in the heterojunction. This was accomplished by
applying thermal stress to the parallel plane and measuring the direct
pyroelectric-photovoltaic power. Supplementary Figs. 28, 29 sum-
marize the J-V and P-V characteristics of the device at temperatures
ranging from 8–80 °C. The temperature and pulsed light intensity and
their distributions were accurately measured using a thermal camera,
thermocouple unit, and reference Si photodiode (see methods). In
various thermal states, the ZnO/NiO device showed a PCE of approxi-
mately 9% in the 8–28 °C temperature range. This value decreased to
7.79% at 80 °C under pulsed illumination wavelengths of 365 nm and
100 μW cm⁻², as shown in Fig. 5c. This decrease was due to the
decrease in the $V_{OC}$ from 324 to 273 mV. An increase in pyroelectric-$J_{SC}$
caused an increase in the IPCE, from 183.5% to 206%, as shown in
Fig. 5d. Supplementary Fig. 30 summarizes the PV and PE-PV and
photovoltaic performance parameters of the temperature series.

The AgNW/ZnO top electrode design offers excellent stability to
the device under thermal operation due to its hybrid structure[59]. The

absorption profiles of the AgNW and ZnO-coated AgNW electrodes in the wavelength range of 250–500 nm were identical, as confirmed by Supplementary Fig. 31. These devices exhibited similar PE-PV performance under pulsed illumination. This confirms the usefulness of the AgNW/ZnO electrode design, which offers enhanced performance with a $P_{max}$ of 108.8 μW cm$^{-2}$ and a PCE of 7.77% (Supplementary Table 3) for TPHD. We also conducted an adhesion test using the Kapton tape peeling method (Supplementary Fig. 32). Prior to the Kapton peeling test, we measured the $J_{SC}$ profiles of TPHD devices with AgNW/ZnO electrodes, which showed $J_{SC}$ values of 0.312 mA cm$^{-2}$ for the PV phenomenon and 1.1125 mA cm$^{-2}$ for the PE-PV phenomenon. It resulted in IPCE values of 88.47% and 314.95% for the PV and PE-PV phenomena, respectively. During the first cycle of Kapton tape peeling from the device electrode, we observed some exfoliation of the AgNW network from the electrode border region due to the agglomeration of the AgNW bundles near the mask edge area by drop casting. However, we noticed greater consistency in the $J_{SC}$ profiles throughout the Kapton peeling test. The TPHD device with the AgNW/ZnO electrode provided outstanding performance beyond the thermodynamic limit (IPCE > 100%).

### Pyroelectric coefficient and figure of merit

The pyroelectric coefficient ($P_i$) was carefully measured to study the high pyroelectric current responses of the ZnO/NiO devices. We used the pulsed illumination method of Chynoweth, which is a sensitive and nondestructive way to study the polarization state of a device[5,10,11,63]. By illuminating the device with pulsed light that enters from the top electrode to the junction, we observed a surge in the nanosecond-scale pyroelectric current response due to increased thermal change (d$T$/d$t$), resulting in spontaneous polarization. This test continued with normalization at the millisecond scale. Upon returning to the dark state, the device showed a current surge in the opposite direction due to altered d$T$/d$t$ and then returned to the dark current of the device, as shown in Fig. 5e. As the device temperature increased from 25–65 °C, the observed trend increased, leading to a prominent pyrocurrent response. The electrode area was defined using a laser-patterned mask to ensure the accuracy of the top AgNW/ZnO electrode. Furthermore, to improve the statistics, we fabricated a total of 21 devices. The vertical structure of the ZnO/NiO device and the illumination direction eliminated the possible contributions from the thermoelectric and lateral electrode geometries.

The $P_i$ constant can be calculated within a specific temperature range as follows[14,64]:

$$P_i = \frac{Jdt}{dT} = \frac{\sigma}{dT} \quad (12)$$

where J represents the current density measured for the pyroelectric response, $\sigma$ is the charge density achieved by integrating the pyroelectric current J over time ($t$), and d$T$ is the temperature interval. In Fig. 5f, the estimated pyrocurrent responses for various T are shown, where the photocurrent contribution should be excluded for accurate $P_i$ calculations. From this, we obtained an effective $P_i$ for a pyroelectric-photovoltaic heterojunction device. Figure 5g presents the $P_i$ values for all devices, demonstrating a remarkable consistency of $P_i$ in the range of 25–35 mC m$^{-2}$ K$^{-1}$. This result shows that the ZnO/NiO device can deliver 1000 times better $P_i$ performance than the ZnO device. These effects have two important features that arise from the heterojunction interface. First, the built-in electric field is polar and induces additional polarization along the $c$-axis. This polarization is proportional to the square of the electric field through electrostriction[10]. The electric field E is calculated as $V_{OC}/d$, where $d$ is the thickness of the heterostructure (approximately 100 nm). This value is determined to be $4 \times 10^6$ V m$^{-1}$. Second, the heterojunction partner provides a thermal reservoir due to the dissimilar thermal

expansion coefficient. This can be engineered for dT/dt with an appropriate p-type partner to the ZnO pyroelectric absorber[25].

The formula given below defines the figure of merit (FOM) for pyroelectric materials, which considers the specific heat capacity ($C_p$), the relative permittivity ($\varepsilon_r$) of the pyroelectric material, and the free space permittivity ($\varepsilon_o$):

$$FOM_{Pi} = \frac{P_i}{C_p \varepsilon_r \varepsilon_o} \quad (13)$$

For ZnO, $C_p$ is 2.9 J K$^{-1}$ cm$^{-3}$, and $\varepsilon_r$ is 10.4. We use these values in Fig. 5h to summarize $P_i$ versus $FOM_{Pi}$ for various types of devices, including crystalline[14], epitaxial[12], nanosheet[11], Schottky polar symmetry[10], pyroelectric material[12], and ZnO/NiO devices. The experimental pyroelectric coefficients for ZnO/NiO heterostructures with various devices are greater than those of most conventional pyroelectric materials. Moreover, the $FOM_{Pi}$ for ZnO/NiO heterostructure devices is significantly larger (ranging from 58 to 121 m$^2$ C$^{-1}$) than that of conventional and emerging pyroelectric devices.

## Discussion

This study demonstrated that a photovoltaic device embedded with a pyroelectric absorber has excellent pyroelectricity. The device can harvest power from periodic photon-induced thermal energy and can use this power to achieve sensing and energy conversion beyond the classical thermodynamic limits. The use of pyroelectric ZnO, which comprises vacancy defects of Zn and O, provides photonic charges that lead to spontaneous polarization along the $c$-axis under pulsed illumination. The pyroelectric-photovoltaic performance can be fine-tuned by optimizing parameters such as pulse illumination frequency, scan speed, and step interval, allowing for outstanding PCE and IPCE values that surpass the classic thermodynamic limit. A mechanistic sequence describing this power harvesting is described and supports a robust and efficient energy conversion process that occurs during each thermodynamic cycle. The combination of the pyrocoefficient, thermal stress, and illumination wavelength results in a remarkable PCE of 11% and an IPCE of 200%. The incorporation of a pyroelectric absorber with spontaneous polarization can be improved to achieve highly efficient, sensitive, and high-speed photoelectric devices. Additionally, the possible influence of optically ionized dynamic charges in heterojunctions allows diffusion-less ballistic-like transport and optically tunable electrostatics. This work provides promising approaches for pyroelectric-photovoltaic power generation and expands the scope of pyroelectric materials.

## Methods

### Device fabrication

First, fluorine-doped tin oxide (FTO)/glass with a size of 2.5×2.5 cm$^2$ (735159 Aldrich, sheet resistance of 7 Ω square$^{-1}$) was used as the substrate. These samples were cleaned using an ultrasonic bath with acetone, methanol, and deionized water for 10 min each at a bath temperature of 30 °C and then dried using nitrogen gas.

The second step involved the use of a 4-inch magnetron sputtering method at a temperature of 60 °C to form films of Zinc Oxide (ZnO)/Nickel Oxide (NiO) heterojunctions. A ZnO film was deposited from the ZnO target (iTASCO, purity 99.99%) using a radio frequency (RF) sputtering system (Solarlight Ltd., Korea) at a sputtering power of 150 W, an Ar gas flow of 50 sccm, and a working pressure of 5 mTorr, resulting in a ZnO film growth rate of 3.977 nm min$^{-1}$. NiO film was deposited from the Ni target (iTasco, purity 99.99%) by reactive sputtering using direct current (DC) sputtering at a sputtering power of 50 W, an Ar gas flow of 20 sccm, an O$_2$ gas flow of 5 sccm, and a working pressure of 3 mTorr, resulting in NiO film growth of 2 nm min$^{-1}$. The thickness and growth rate of the films were confirmed by ellipsometry. The ZnO and NiO film thicknesses were set to 100 and

12 nm, respectively, for the ZnO/NiO heterojunction. Before film deposition, a base vacuum pressure of $2 \times 10^{-6}$ Torr was applied. The substrate was rotated five times per minute to ensure uniform film preparation. FTO films were masked using Kapton tape for cathode connection.

In the third step, the top anode electrode of the silver nanowire (AgNW)/ZnO electrode was formed using drop-cast AgNW ink followed by ZnO sputtering. The diameter of the AgNWs was 20 nm, and the length was 20 μm. The AgNWs were dispersed in isopropanol and dropped as received from Nanopyrix (South Korea) over the heterojunction. Ink spreading was confined using a mask with a diameter of 3 mm. Casting was performed at 100 °C for 2 min under a vacuum chamber, and the pressure was $5 \times 10^{-2}$ Torr. To provide thermal and environmental protection, the AgNW network was coated with a 10 nm layer of ZnO to form reliable electric contacts.

Finally, array devices of $4 \times 4$ were fabricated using LASER patterning and a custom mask of polyethylene terephthalate (PET). First, FTO/glass was patterned using a Nd-YAG laser with a unit size of $6.25 \times 6.25$ mm$^2$ (Cat-UV 5 W, Marcs, South Korea). The laser power, scan speed, and number of cycles were optimized for FTO scribing, which provides electrical isolation. Then, a Kapton tape mask ($2 \times 2$ mm$^2$) was applied to each cell for cathode contact and processed for ZnO film deposition of 100 nm. Subsequently, the NiO heterojunction layer was deposited on the ZnO area with a thickness of 12 nm. Then, the masks were removed from each device. A PET mask array was fabricated for top AgNW/ZnO electrode preparation using LASER patterning. The mask was designed using the EZCAD laser marking tool and processes with optimum parameters. The resulting PET mask array was aligned on the ZnO/NiO electrode using an optical microscope. Subsequently, AgNW/ZnO anode electrodes were prepared by AgNW drop casting, thermal treatment, and ZnO deposition.

### Device characterization

A high-resolution PL/Raman microscope (JOBIN YVON LabRAM Hr800) was used to study the photoluminescence (PL) spectra. The measurements were taken at room temperature with the help of 355 nm laser excitation. The transmittance spectra of the device were measured using a UV–visible-NIR spectrophotometer (Shimadzu, UV-2600) with a diffuse reflectance integrated sphere. The interval between each step was 5 nm, and air was used as a baseline. The incident light direction was from the anode side. Ellipsometry (ALPHA-SE, ellipsometer) was used to obtain the film growth rate by measuring the thickness. A Si wafer with a transparent film model was used for fitting the measured ellipsometry data. A cross-sectional field-emission transmission electron microscope (FETEM, JEM-2100F; JEOL) was used to analyze the physical and elemental properties of each layer. A focused ion beam was applied for cross-sectional analysis. The FETEM results were analyzed using a GMS 3 digital micrograph. Diamond 3.1 f (Crystal Impact GbR) was used for crystal structure visualization and c-axis identification.

### Pyroelectric-photovoltaic effect test

For electrical measurements, gold-coated tungsten pogo pins connected by crocodile clips were used. The device was positioned underneath the light source, with the AgNW/ZnO electrode connected to the working electrode (anode) and the FTO electrode connected to the counter electrode (cathode). The counter and working sense electrodes were then connected to the counter and working electrode, respectively.

To investigate the current-voltage (J-V) and power-voltage (P-V) characteristics, a potentiostat/galvanostat (PGStat, WonATech, ZIVE SP2) was used. Before conducting the measurements, both the PGStat and electrodes were calibrated using standard LCR circuits. Linear sweep voltammetry was performed with a voltage scan range of $-0.2$ to 0.6 V, scan speed of 0.2 to 5 V s$^{-1}$, and sample interval of 3 μV to 100 mV, with appropriate current compliance.

Light-emitting diodes with various wavelengths were linked to a dual-adjustable power supply and a function generator (Mightex WheeLEDTM, P/N: WLS-22-A). The wavelengths of the LED light sources used were 340 nm (WLS-LED-0340-02), 365 nm (WLS-LED-0365-04), 400 nm (WLS-LED-0400-04), 410 nm (WLS-LED-0410-03), 460 nm (WLS-LED-0460-03), 520 nm (WLS-LED-0520-03), 625 nm (WLS-LED-0625-03), 730 nm (WLS-LED-0730-03) and 850 nm (WLS-LED-0850-03). The light intensity was calibrated using a standard Si photodiode and a UV light meter (Lutron, UV-340A). The pulse illumination frequency was adjusted from 1 to 250 Hz with a duty cycle of 50%.

The device arrays were situated on a custom-built thermal setup that included a Peltier module and heatsink. The temperature of the setup was calibrated in real time using an Arduino Uno module and a K-type thermocouple that was attached to the device with appropriate electrical insulation. A thermal imaging camera (Fluke, Fluke Ti95 Thermal Imager) was used to monitor the temperature during the measurements.

The Chynoweth technique was used to obtain the pyroelectric current at various temperatures. The chronoamperometry method was utilized to measure the transient current profiles under pulse illumination. The current range was custom fit to 2 mA, and the sample period was set to burst mode, allowing for a resolution of 40 μs. A pulse illumination wavelength of 365 nm, an intensity of 1.5 mW cm$^{-2}$, and a frequency of 500 Hz were applied for a data acquisition duration of 20 ms.

## Data availability

All relevant data are presented via this publication and Supplementary Information. The data that support this study are available from the corresponding author upon request. Source data are provided with this paper.

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

## Acknowledgements

The authors acknowledge the financial support of the Brain Pool Program funded by the Ministry of Science and ICT (RS-2023-00283263, NRF-2022H1D3A2A01089675, and NRF-2021H1D3A2A02096147) and Basic Science Research Program through the National Research Foundation (NRF-2022R1I1A1A01054397) of the Ministry of Education of Korea. This work was also supported by the Industrial Strategic Technology Development Program (RS-2023-00257784, Development of Process Technology for Nanomaterials and Components in Semiconductor and Display) funded by the Ministry of Trade, Industry & Energy (MOTIE, Korea).

## Author contributions

The manuscript was written through the contributions of all authors. M.P. and J.K. conceived the project. M.P. designed and performed most of the experiments and analyzed the data. H.P. performed HRTEM analysis, and M.P. analyzed the structural properties, P.B., N.K., and J.L. offered help in pyroelectric-coefficient experiments. M.P. and P.B. prepared array devices and setup for pyroelectric measurements. P.B. and N.K. confirmed the spectral and thermal performances. M.P. prepared the manuscript. M.P. and J.K. revised the manuscript. J.K. supervised the project.

## Competing interests

The authors declare no competing interests.
