## [Peer Review File · Nature Communications]

Transparent integrated pyroelectric-photovoltaic structure for photo-thermo hybrid power generationReviewers' comments:

Reviewer #1 (Remarks to the Author):

This manuscript reports a photovoltaic device consisting of a pyroelectric absorber that possesses excellent pyroelectricity. The device can harvest power through periodic photon-induced thermal energy for sensing and energy conversion. This work is interesting, which provides promising approaches for pyroelectric-photovoltaic power generation. However, the logic of this manuscript is in a mass and it is not well written that requires substantial modification.

1. The first part in the Results “Pyroelectric embedded heterojunction device” should be put into the introduction as the authors mainly focused on the previous research, which is related to this research.
2. It seems that the authors mainly focus on the characterization of the structure and materials instead of the whole device, which makes the novelty of this research unclear. Moreover, the mechanism of the results in Fig. 3 and 4 should be explained clearly.
3. The authors claimed that their power generation is beyond the thermodynamic limit. I am not sure it is new as this kind of device already possessed an efficiency above the thermodynamic limit as described in Line 40, 101 and 203 with clearly cited references.
4. A rigorous Fig. 1c should be composed.
5. It is hard to understand in Figure 1d that a built-in electric field and temporal electric field were caused by temperature rate change.
6. Why are the pyroelectric coefficients of ZnO/NiO heterostructure samples higher than those of most conventional pyroelectric samples?

I would recommend that the authors consider these remarks seriously, and I would be happy to see a revised version again.

Reviewer #2 (Remarks to the Author):

In this manuscript, Dr. J. Kim and collaborators have prepared and characterized the pyroelectric-photovoltaic devices equipped with the ZnO/NiO heterostructure geometry. Some interesting results are presented. After the careful reading and comparison with previous work, however, I find that the current results lack enough novelty and have some main issues that are very important and addressed throughout the manuscript. At this form, I do not think this manuscript is suitable for publication in Nat. Commun. Here are some concerning issues:

-> From the design point of view, the authors have designed a heterostructure detector structure by stacking the ZnO/AgNW heterostructure on the NiO/ZnO heterojunction. As far as we know, this structure is a very common means of constructing the heterojunction devices and the authors have published many similar reports, such as Nano Energy 2022, 100, 107504; Adv. Sci. 2023, 10, 2303895. In the submitted work, the authors simply combine the idiomatic methods of the field, and the device performances obtained are not satisfactory for in Nat. Commun.

-> From the perspective of mechanism, it is well known that the pyro-phototronic effect enhances photovoltaic energy harvesting, and there also have been many articles about this phenomenon (Nanomaterials 2023, 13, 1336). In addition, the ability of about 2.5 times photopyroelectric current to enhance the photovoltaic signal ($I_{\text{pyro+photo}}/I_{\text{photo}}$) is lower than many other studies (Nano Energy 2019, 62, 310–318). This result is not attractive for readers.

-> The authors should describe why ZnO/AgNW/NiO/ZnO was chosen for the photovoltaic devices. Besides, the not-sharp cross-sections in Figure 1f may cause serious interface problems, which is important to control the device quality.

-> The authors should add more characterization work on the device quality, such as adhesion, stability, etc

Overall, I think this manuscript is not suitable for the possible publication in Nat. Commun., and it should be submitted to a specialized journal.

Reviewer #3 (Remarks to the Author):

See Attachment

In this paper, the authors attempt to use a photovoltaic heterojunction incorporated with a pyroelectric effect to efficiently generate electricity beyond thermodynamic limit. Specifically, the pyroelectric-photovoltaic devices utilize spontaneous polarization by pulsed light-induced thermal change to produce output power, which is further beyond the traditional photovoltaics. Moreover, through the adjustments of pulse frequency, scan speed, sample interval, illumination wavelength, temperature, and pyroelectric coefficient, this type of pyroelectric-photovoltaic devices can optimize power harvesting, achieving a high power conversion efficiency of 11.9% and an incident-photon-to current conversion efficiency of 200%.

Despite the notable achievements presented by the authors, several critical issues have been identified, making it difficult for me to recommend this article for publication in this journal. Firstly, from my perspective, the authors may have overly exaggerated their results. Given that photovoltaic devices are typically evaluated under the AM1.5G spectrum, the reliance on pulsed light with specific wavelengths in this study raises concerns about the generalizability of the results. Additionally, the evaluation method requires further confirmation. Given the use of a pulsed light source, it is suggested that energy output calculations should be weighted based on the pulse on/off ratio rather than relying solely on the maximum value.

Technology comments:

1. In this study, the reliance on a specific pulse wavelength rather than operating under AM1.5g conditions should be explicitly highlighted. Consequently, this article should be viewed as a conceptual validation rather than a universal breakthrough of thermodynamic limits for traditional photovoltaic devices, preventing potential reader misinterpretation. The clarification also extends to the acknowledgment that the presented maximum efficiency of 12% is lower than the current highest efficiency of single-junction devices (>26%), reinforcing the conceptual nature of the study.
2. Further clarification on the evaluation indicators is required. Considering the thermoelectric effect necessitates a pulse light source for excitation, it is crucial to incorporate the pulse on/off ratio in the calculation of maximum efficiency to eliminate potential ambiguities in the 200% photoelectric conversion.

Other comments:

1. Role of silver nanowires
Here, silver nanowires doped with ZnO are used as the front-sided electrodes. The authors provide the function of silver nanowires, as it can generate significant parasitic absorption, thereby reducing the effective absorption and performance of the devices.
2. Element distributions
The issue of element ratios not totaling 100% within the specified range (i.e., 0-100 nm) in Figure 1g should be addressed, and an explanation for the absence of the Zn

signal within this range should be provided. Additionally, the roughness of the EDS signal tested by TEM should be acknowledged, highlighting its limitations in accurately confirming the distribution of elements.

3. Preparation Details of ZnO and NiO

How are zinc oxide and nickel oxide prepared? Magnetron sputtering? Because the authors demonstrate high crystallinity of these materials, the details of the preparation need to be emphasized.

4. Ionization Energy

A literature-supported explanation for obtaining activation energy values in Equations 3-6 should be provided.

5. Working area

As shown in Fig. 3a-3b, why the working region of the samples is cylindrical?

6. PE and PV current

In Fig. 3c, how to distinguish the currents of PE and PV components?

7. Error bars

A detailed explanation for the significant difference in error bars between P_{max} and power conversion efficiency in Fig. 4d-4f should be provided.

8. The authors should provide a detailed explanation for the phenomenon where incident-photon-to-current conversion efficiency (IPCE) exceeds 100%.

Response to Reviewer's Comments

Research Article no.: NCOMMS-23-56542

TITLE: Transparent Pyroelectric-Combined Photovoltaic Structure for Power Generation beyond Thermodynamic Limit

We appreciate the reviewers who took the time to review the study and provided constructive feedback. The detailed and thoughtful comments and suggestions are helpful for the authors to improve their work.

The manuscript has been revised following the reviewers' advice, and the changes are highlighted in yellow. The responses to the reviewer's comments are presented below in blue fonts.

Response to the reviewer	Page numbers
Reviewer#1	2-16
Reviewer#2	17-33
Reviewer#3	34-64

Reviewer #1 (Remarks to the Author):

This manuscript reports a photovoltaic device consisting of a pyroelectric absorber that possesses excellent pyroelectricity. The device can harvest power through periodic photon-induced thermal energy for sensing and energy conversion. This work is interesting, which provides promising approaches for pyroelectric-photovoltaic power generation. However, the logic of this manuscript is in a mass and it is not well written that requires substantial modification. I would recommend that the authors consider these remarks seriously, and I would be happy to see a revised version again.

Response

We sincerely thank the reviewer for taking the time to review our study and providing constructive feedback. We appreciate your positive comments, which encourage us so much. Your detailed and thoughtful comments and advice are very helpful to us. We have revised the manuscript by following the reviewer's suggestion in the following point-by-point responses.

1. The first part in the Results "Pyroelectric embedded heterojunction device" should be put into the introduction as the authors mainly focused on the previous research, which is related to this research.

Response-1

We greatly thank the reviewer for this thoughtful suggestion.

We put this discussion into the introduction, which mainly provides progress on the previous research.

Changes made to the manuscript

The first part of the Results, "Pyroelectric embedded heterojunction device" moved to the introduction. Page 3.

Fig. 1a illustrates the Carnot energy diagram, which shows the work done by a substance while transferring heat from a hot temperature source ($T_H = 5800$ K for the sun) to a cold sink ($T_C = 300$ K for the device). When the conversion process is reversible, the energy conversion efficiency of the Carnot engine is given by $\eta = 1 - \frac{T_C}{T_H}$, and it equals 94.82%.³⁷ Energy conversion devices can intake energy in steady or pulsed modes, as shown in Fig. 1b. The single junction photovoltaic cell, usually used in steady-state mode, has a PCE limit due to the bandgap of absorbing material and thermalization process. However, pyroelectric materials can go beyond this thermodynamic limit by producing work under pulsed illumination. Fig. 1c shows that the pyroelectric material does cyclic work under pulsed illumination, which exceeds the power conversion efficiency limit.^{2,6,12} The reversible polarization current, produced by the electric field modulation, is responsible for this. The cyclic process consists of four steps: $C \rightarrow D$ and $A \rightarrow B$ are isoelectric, with no difference in electric field, for heat addition and rejection. $B \rightarrow C$ and $D \rightarrow A$ are the adiabatic process, i.e., a thermodynamic process that occurs without heat transfer for charging and discharging.³⁸ Pyroelectric materials can convert energy cyclically through switchable

spontaneous polarization, producing work beyond the thermodynamic limit under pulsed illumination.³⁹

2. It seems that the authors mainly focus on the characterization of the structure and materials instead of the whole device, which makes the novelty of this research unclear. Moreover, the mechanism of the results in Fig. 3 and 4 should be explained clearly.

Response-2

We are grateful to the reviewer for this comment and thoughtful suggestion.

In order to provide a complete view of the device structure, Fig. 1f in the manuscript displays a lower magnification image that shows various layers such as glass, FTO, ZnO, NiO, and AgNW. To analyze each layer in detail, a thorough examination was conducted to determine the *c*-axis orientation of individual layers of FTO, ZnO, and NiO (Fig. 2b-2e). The atomic layer spacing of each layer was also measured over the 5-nm region, and the results are represented in Figure R1-1. The analysis indicates the presence of FTO, ZnO, ZnO/NiO, and NiO layers with measured *d*-spacing values. The FTO layer (region-A) has a *d*-spacing value of 0.165 nm, while the ZnO layer has a *d*-spacing value of 0.264 nm at the FTO interface (region-B) and in the center region (region-C). The NiO layer (region-B) has a *d*-spacing value of 0.2337 nm. The manuscript provides a detailed discussion of these results for the readers.

Changes made to the manuscript

On Page 4

“...which shows a device interface of FTO/ZnO ...”

“...(HRTEM) of the device is depicted...”

Figure R1-1. Top panel in red highlight: Device cross-section analysis revolving the c -axis orientation of each layer of FTO (A), ZnO at FTO interface (B), ZnO center region (C), and NiO at ZnO interface (D). Bottom panel in blue highlight: Inter-atomic distance over the c -axis resolved region in the yellow box is profiled, confirming the d -spacing of 0.165 nm for the FTO (region-A), 0.264 nm of ZnO at FTO interface (B) and center region (C), and 0.2337 nm of the NiO (region-B).

To explain the mechanism of the results in Fig. 3 and 4, we have carried out detailed photon balance calculations to explain the IPCE values for the PV and PE-PV performances. We assume that each photon absorption generates an electron-hole pair, and their collection generates photocurrent. Under ideal conditions (perfect absorption, zero reflection, unit electron-hole pair generation, zero parasitic electrical losses, and recombination-less transport), the IPCE value is 100%. We can also calculate the photocurrent density for the zero-bias condition, called the short-

circuit current density (J_{sc}). As this report discloses pyroelectric-photovoltaic power generation, we first confirm the accuracy of the IPCE values estimated from the PV segment. The IPCE values should match the calculated J_{sc} value and should not exceed it, providing photon balance.

The IPCE performances for the TPHD array devices are shown in Fig. 3j (main manuscript). We estimated these values for PV and PE-PV as 77% and 177%, respectively, from the J_{sc} values measured by the J-V characteristic plots under the illumination of 365 nm wavelength and intensity of $500 \mu\text{W cm}^{-2}$. This illumination provides a photon flux density of $9.17975 \times 10^{14} \text{ cm}^{-2} \text{ s}^{-1}$. Under ideal conditions, the photovoltaic device provides a J_{sc} value of $147.076 \mu\text{A cm}^{-2}$, which serves as a check value for the measured PV performances. Table R1-1 summarizes the detailed calculation for the photon-flux, calculated J_{sc} , measured J_{sc} of the sixteen array devices, and their calculated IPCE values. We found that the measured J_{sc} value for the PV performance remained below the theoretically calculated J_{sc} value, with an average value of $94.58 \mu\text{A cm}^{-2}$, resulting in an IPCE value of 64.25%. However, due to spontaneous polarization by the pyroelectric-photovoltaic device, the measured J_{sc} value for the PE-PV performances exceeded the calculated J_{sc} value with an average value of $239.6 \mu\text{A cm}^{-2}$, equivalent to an IPCE value of 162.77%. Our analysis shows that the charge density that participated in the spontaneous polarization by the pyroelectric-photovoltaic phenomenon is $9.05 \times 10^{14} \text{ cm}^{-2} \text{ s}^{-1}$. This value is close to the photon flux and suggests that photo-ionized impurities participate in spontaneous polarization by the pyroelectric-photovoltaic phenomena. We have included these results and analyses in the revised manuscript as suggested.

Table R1-1. Summary of the detailed photon balance calculation and measured J_{sc} and IPCE values for TPHD devices.

Photon flux	Calculated J_{sc}	Device	Measured J_{sc} $\mu\text{A cm}^{-2}$		Measured IPCE (%)		Additional charges by pyroelectric ($\text{cm}^{-2} \text{ s}^{-1}$)
			PV	PE-PV	PV	PE-PV	
$9.17975 \times 10^{14} \text{ cm}^{-2} \text{ s}^{-1}$	$147.076 \mu\text{A cm}^{-2}$	D1	86.55	233.33	58.80	158.51	9.16×10^{14}
		D2	89.18	252.76	60.58	171.71	1.02×10^{15}
		D3	87.08	231.94	59.16	157.57	9.04×10^{14}
		D4	107.00	229.09	72.69	155.63	7.62×10^{14}
		D5	84.32	218.88	57.28	148.69	8.40×10^{14}
		D6	93.15	217.69	63.28	147.89	7.77×10^{14}
		D7	88.65	232.39	60.22	157.87	8.97×10^{14}
		D8	89.49	247.58	60.79	168.19	9.87×10^{14}
		D9	94.94	246.25	64.50	167.29	9.44×10^{14}
		D10	97.35	254.72	66.13	173.04	9.82×10^{14}
		D11	87.77	238.28	59.63	161.87	9.39×10^{14}
		D12	113.57	245.23	77.15	166.60	8.22×10^{14}
		D13	101.78	233.61	69.14	158.70	8.23×10^{14}
		D14	84.00	244.14	57.06	165.85	1.00×10^{15}
		D15	113.97	262.45	77.42	178.29	9.27×10^{14}
		D16	94.45	245.29	64.16	166.64	9.41×10^{14}
Average			94.58	239.60	64.25	162.77	9.05×10^{14}

Changes made to the manuscript

On Page 7

We conducted photon balance calculations to determine the IPCE values for the PV and PE-PV technologies. We assumed that each photon absorption results in an electron-hole pair, which generates photocurrent when collected. In perfect conditions where there is complete absorption, no reflection, perfect electron-hole pair generation, no parasitic electrical losses, and no recombination, the IPCE value should be 100%. Using this assumption and an estimated photon density, we can calculate the photocurrent density for the zero-bias condition, which is also known as (J_{sc}). We estimated photon density using the formula given below:

$$\text{Photon density} = \frac{I}{E_p} = \frac{I \times \lambda \times 10^{-9}}{hc} = \frac{I (W m^{-2}) \times \lambda \times 10^{-9} (m s)}{1.988 \times 10^{-25} (J s m)} = I \times \lambda \times 5.03 \times 10^{15} (m^{-2} s^{-1}).$$

where, λ and I are the illumination wavelengths in m and intensity in $W m^{-2}$, respectively, h is the Planck constant ($6.63 \times 10^{-34} J s$) and c is the speed of light ($2.998 \times 10^8 m s^{-1}$).

First, we confirmed the accuracy of the IPCE values estimated from the PV segment, which should match the calculated J_{sc} value and should not exceed it, providing photon balance. The IPCE performances for the TPHD array devices are shown in Fig. 3j. We estimated these values for PV and PE-PV as 77% and 177%, respectively, from the J_{sc} values measured by the J-V characteristic plots under the illumination of 365 nm wavelength and intensity of $500 \mu W cm^{-2}$. This illumination provides a photon density of $9.17975 \times 10^{14} cm^{-2} s^{-1}$. Under ideal conditions, the PV operation provides a J_{sc} value of $147.076 \mu A cm^{-2}$, which serves as a check value for the measured PV performances. Table 2 summarizes the detailed calculation for the photon-flux, calculated J_{sc} , measured J_{sc} of the sixteen array devices, and their calculated IPCE values. Based on our analysis, we found that the measured J_{sc} value for the PV performance complies with the theoretically calculated J_{sc} value. The average value is $94.58 \mu A cm^{-2}$, resulting in an IPCE value of 64.25%. However, due to spontaneous polarization by the pyroelectric-photovoltaic device, the measured J_{sc} value for the PE-PV performances exceeded the calculated J_{sc} value with an average value of $239.6 \mu A cm^{-2}$, equivalent to an IPCE value of 162.77%. This analysis shows that the charge density that participated in the spontaneous polarization by the PE-PV phenomenon is $9.05 \times 10^{14} cm^{-2} s^{-1}$. This value is close to the photon flux and suggests that photo-ionized impurities participate in spontaneous polarization by the PE-PV phenomena.

Furthermore, as the pyroelectric-photovoltaic effect requires a pulsed illumination, we have designed a pulse on/off ratio to calculate the maximum efficiency, eliminate confusion in photoelectric conversion, and serve as an evaluation indicator.

Using LED illumination, we regulate the pulse on/off ratio by varying the pulse width (PW) for the light-on interval within a fixed total signal period of 20 ms. We have programmed duty cycles ranging from 0-100%, which are illustrated in Figure R1-2. At an intensity of $1.9 mW cm^{-2}$ and a wavelength of 365 nm, we obtain a photon density of $3.48831 \times 10^{15} cm^{-2} s^{-1}$. By controlling the duty cycle, we can regulate the photon dose within a range of 1.395×10^{12} to $6.977 \times 10^{13} cm^{-2}$,

as summarized in Figure R1-2c. Assuming 100% IPCE (incident photon to current efficiency) performance, the expected photocurrent density is summarized as 55.88 $\mu\text{A cm}^{-2}$ for a 10% duty cycle and 503.89 $\mu\text{A cm}^{-2}$ for a 90% duty cycle. We can calculate the photon density using the following formula:

$$\text{Photon density} = \frac{I}{E_p} = \frac{I \times \lambda \times 10^{-9}}{hc} = \frac{I (\text{W m}^{-2}) \times \lambda \times 10^{-9} (\text{m s})}{1.988 \times 10^{-25} (\text{J s m})} = I \times \lambda \times 5.03 \times 10^{15} (\text{m}^{-2} \text{s}^{-1}).$$

where, λ and I are the illumination wavelengths in m and intensity in W m^{-2} , respectively, h is the Planck constant (6.63×10^{-34} J s) and c is the speed of light (2.998×10^8 m s^{-1}).

The Figure R1-3 shows us the current density-voltage (J-V) characteristics of the TPHD device for different duty cycles. The experiment was conducted using an illumination wavelength of 365 nm, with an intensity of 1.9 mW cm^{-2} and a frequency of 50 Hz. The results demonstrate that the 10-90% duty cycle range enables pyroelectric-photovoltaic energy conversion, which helps distinguish the PV and PE-PV components. The power density-voltage (P-V) characteristic plots of the duty cycle series are summarized in Figure R1-4, where the output power is maximum for the duty cycle value of 50% for the PE-PV segment, while it is consistent for the PV segment for the duty cycle from 20% to 98%.

The duty cycle controls the photon dose for the fixed interval of the pulse cycle, which is 2 ms. During the light-on pulse region, photon absorption activates the Frenkel reaction of the Zn interstitials, leading to the generation of ionized donor-type defects (Zn_i^+ and Zn_i^{++}) with activation energy of 50 meV and 150 meV. (Materials Today 10, 40-48 (2007)) This is likely to induce temperature change and cause the polarization of these ionized defects, providing pyroelectric components to the photovoltaic under the pulsed illumination. The photon dose triggers the polarization current, and the effect is prominent for the duty cycle of 2%, which is equivalent to the photon dose of 1.395×10^{12} cm^{-2} .

The results obtained suggest that the duty cycle could serve as an evaluation indicator for pyroelectric-photovoltaic and photovoltaic performance parameters, including open-circuit voltage (V_{oc}), short-circuit current density (J_{sc}), maximum output power (P_{max}), fill factor (FF), incident-photon-to-current conversion efficiency (IPCE), and power conversion efficiency (PCE), as a function of the duty cycle. These parameters are shown in Figure R1-5. The data shows that a lower photon dose provides higher IPCE performances by PE-PV, attributed to the inevitable Frenkel reaction of the Zn interstitial, activated by the pulsed photon illumination, and their ionized species cause the polarization current. This result and discussion are added to the revised manuscript.

Figure R1-2. Experimental design on clarification on the pyroelectric power evaluation indicators. (a) The device under test incorporates the pulse on/off ratio employing a duty cycle. (b) Schematic showing the duty cycle calculation. (c) Summary of duty cycle, pulse width, photon density, and dose during complete pulse interval.

Figure R1-3. Current density-voltage (J - V) characteristics of the device under the pulsed light illumination of various duty cycles from 2% to 98%. Throughout these measurements, scan speed was 0.5 V s^{-1} . The light illumination of the wavelength, intensity, and pulsed frequency was 365 nm, 1.9 mW cm^{-2} , and 50 Hz, respectively.

Figure R1-4. Power density-voltage (P-V) characteristics of the device under the pulsed light illumination of various duty cycles from 2% to 98%. Throughout these measurements, the scan speed was 0.5 V s^{-1} . The light illumination of the wavelength, intensity, and pulsed frequency was 365 nm, 1.9 mW cm^{-2} , and 50 Hz, respectively. These plots show output power for load voltage. The peak value of the power plot shows the maximum output power and, hence, maximum voltage and current density.

Figure R1-5. Summary of (a) open-circuit voltage (V_{oc}), (b) short-circuit current density (J_{sc}), (c) maximum output power (P_{max}), (d) fill factor (FF), (e) incident-photon-to-current conversion efficiency (IPCE), and (f) power conversion efficiency (PCE) as a function duty cycle of light illumination.

Changes made to the manuscript

On Page 9-10

In order to generate the PE-PV effect, a pulsed illumination is required. To evaluate the energy conversion, we varied the duty cycles by changing the pulse width of the light-on interval within a fixed total signal period of 20 ms, as shown in Supplementary Fig. 19. At an intensity of 1.9 mW cm^{-2} and a wavelength of 365 nm, we achieved a photon density of $3.488 \times 10^{15} \text{ cm}^{-2} \text{ s}^{-1}$. By controlling the duty cycle, we were able to maintain the photon dose within a range of 1.395×10^{12} to $6.977 \times 10^{13} \text{ cm}^{-2}$. For 100% IPCE performance, the expected J_{sc} ranges from $55.88 \text{ } \mu\text{A cm}^{-2}$ for a 10% duty cycle to $503.89 \text{ } \mu\text{A cm}^{-2}$ for a 90% duty cycle.

The J-V characteristics of the TPHD device for different duty cycles are shown in Supplementary Fig. 20. The duty cycle range of 10-90% enables PE-PV energy conversion, which distinguishes the PV and PE-PV components. The P-V characteristic plots of the duty cycle series are summarized in Supplementary Fig. 21, where the output power is maximum for the duty cycle value of 50% for the PE-PV segment, while it remains consistent for the PV segment for the duty cycle range of 20% to 98%.

The duty cycle is responsible for controlling the amount of photons delivered during a fixed interval of the pulse cycle, which is typically 2 ms. During the light-on pulse region, the absorbed photons activate the Frenkel reaction of the Zn interstitials, leading to the creation of ionized donor-type defects (Zn_i^+ and Zn_i^{++}) with activation energy of 50 meV and 150 meV.⁵² This is likely to induce a temperature change and cause the polarization of these ionized defects, providing pyroelectric components to the photovoltaic under the pulsed illumination. The photon dose triggers the polarization current, and the effect is more prominent for a duty cycle of 2%, which is equivalent to a photon dose of $1.395 \times 10^{12} \text{ cm}^{-2}$. Based on the obtained results, it can be concluded that the duty cycle can serve as an evaluation indicator for PE-PV and PV performance parameters as a function of the duty cycle. The data shows that a lower photon dose provides higher IPCE performances by PE-PV. This is attributed to the inevitable Frenkel reaction of the Zn interstitial, which is activated by the pulsed photon illumination, and their ionized species cause the polarization current.

During the fourth step of the experiment, duty cycles ranging from 0 to 100% were tested using the optimal values of f , ω , and Δv . The results, shown in Fig. 4g (obtained from the results summarized in Supplementary Figs. 20-22), indicate that a D value between 10-90% provides consistent performance for PE-PV operation. Power generation ability of the TPHD device is lost if D values are less than 5% for the given parameters. Therefore, it is recommended that D be kept around 50% for this setup to capture the power cycles by pulsed illumination. Among the four parameters, it is suggested to find the optimum pulse frequency for maximum P_{\max} and PCE (around 60 Hz with D around 50%) when ω is less than 1 V s^{-1} and Δv is less than $10 \text{ } \mu\text{V}$.

Reference:

52. Schmidt-Mende, L. & MacManus-Driscoll, J. L. ZnO - nanostructures, defects, and devices. *Mater. Today* **10**, 40–48 (2007).

3. The authors claimed that their power generation is beyond the thermodynamic limit. I am not sure it is new as this kind of device already possessed an efficiency above the thermodynamic limit as described in Line 40, 101 and 203 with clearly cited references.

Response-3

We appreciate the thoughtful comment provided by the reviewer. In our response to Comment #2, we have provided greater detail on the performance beyond the thermodynamic limit, where IPCE performances exceed 100%. We have clarified our approaches described in Lines 40, 101, and 203 to facilitate the reader's understanding of the thermodynamic limit.

Line #40 “The photophysics of piezoelectricity, ferroelectricity, flexoelectricity, and pyroelectricity, offer the device design methodology that integrates low-cost materials with efficiencies and functionalities above the thermodynamic limit.¹⁻⁶”

Reference:

1. Pan, C., Zhai, J. & Wang, Z. L. Piezotronics and piezo-phototronics of third generation semiconductor nanowires. *Chem. Rev.* **119**, 9303–9359 (2019).
2. Velarde, G., Pandya, S., Karthik, J., Pesquera, D. & Martin, L. W. Pyroelectric thin films—Past, present, and future. *APL Mater.* **9**, 010702 (2021).
3. Pandya, S. *et al.* Pyroelectric energy conversion with large energy and power density in relaxor ferroelectric thin films. *Nat. Mater.* **17**, 432–438 (2018).
4. Lheritier, P. *et al.* Large harvested energy with non-linear pyroelectric modules. *Nature* **609**, 718–721 (2022).
5. Ok, K. M., Chi, E. O. & Halasyamani, P. S. Bulk characterization methods for non-centrosymmetric materials: Second-harmonic generation, piezoelectricity, pyroelectricity, and ferroelectricity. *Chem. Soc. Rev.* **35**, 710–717 (2006).
6. Spanier, J. E. *et al.* Power conversion efficiency exceeding the Shockley–Queisser limit in a ferroelectric insulator. *Nat. Photonics* **10**, 611–616 (2016).

References 1-6 report the progress and perspectives of third-generation semiconductors, which provide spontaneous polarization under perturbed input energy. These reports suggest the superior potential of pyroelectric absorbers for energy harvesting, with performance beyond thermodynamic limits. However, none of these reports show the photoelectric energy conversion with IPCE and PCE beyond the thermodynamic limit from pulsed illumination light sources.

Line # 101 “Pyroelectric materials can convert energy cyclically through switchable spontaneous polarization, producing work beyond the thermodynamic limit under pulsed illumination.”³⁷

Reference:

37. Pandya, S. *et al.* New approach to waste-heat energy harvesting: pyroelectric energy conversion. *NPG Asia Mater.* **11**, 26 (2019).

Reference 37 by Pandya et al. presents a new approach to waste-heat energy conversion. The authors provide power generation potential by non-centrosymmetric polar pyroelectrics of crystals by electrical polarization at the rate of temperature and suggest expanding our ability to accurately measure materials and design criteria at the nanoscale device structure. This marks the beginning of a great development for pyroelectric materials and their application in thermal energy conversion.

Line # 203 “According to the thermodynamic limit, the maximum IPCE of a standard photovoltaic device should not exceed 100%.”^{43,44}

Reference:

43. Almora, O. *et al.* Device performance of emerging photovoltaic materials (version 3). *Adv. Energy Mater.* **13**, 2203313 (2023).
44. Green, M. A. *et al.* Solar cell efficiency tables (version 62). *Prog. Photovoltaics Res. Appl.* **31**, 651–663 (2023).

References 43, by Almora et al., provide the progress of the device performance of emerging photovoltaic materials, and reference 44, by Green et al., provides a solar cell efficiency table of the recent year's progress of solar cells and modules. We have cited both references in the context of IPCE performances, explaining the response to Comment #2.

4. A rigorous Fig. 1c should be composed.

Response-4

We appreciate the careful suggestion provided by the reviewer. Figure 1c is revised as shown in Figure R1-6.

Figure R1-6. Polarization versus electric field for transient illustrates Brayton pyroelectric cycle. (a) Before and (b) After correction.

5. It is hard to understand in Figure 1d that a built-in electric field and temporal electric field were caused by temperature rate change.

Response-5

We would like to apologize to the reviewer for any inconvenience caused. We have revised Fig. 1d to illustrate better the built-in electric field (at the ZnO/NiO interface) and temporal electric field (across the device) caused by a temperature rate change. The revised figure, now called Figure R1-7, includes two device schematics. The first schematic shows case-1, which is steady-state illumination in which the built-in electric field is responsible for photovoltaic (PV) activity. The second schematic shows case 2, which is pulsed illumination that clarifies the temporal electric field created by temperature rate change (responsible for pyroelectric-photovoltaic (PE-PV) activity). We have included these revised schematics in the manuscript, as suggested by the reviewer.

Furthermore, the device schematic, along with the modulation of the electric field (built-in and pyroelectric) for the steady-dark, upon light illumination, steady-light illumination, and upon dark, are shown in Figure R1-8, which is also provided in Supplementary Fig. 9 of the revised manuscript.

Figure R1-7. Revised Figure 1d shows the built-in and temporal electric fields by the temperature rate change by the pulsed illumination. This schematic explicitly separates the steady state and pulsed illumination to aid the understanding of the electric field responsible for the PV and PE-PV phenomena.

Figure R1-8. A schematic of the device and its built-in electric field (E_i) and the pyroelectric field (E_{PL} in the light-on state and E_{PD} in the dark state). The schematics demonstrate how drift and diffusion transport are modulated under pulsed light illuminations.

6. Why are the pyroelectric coefficients of ZnO/NiO heterostructure samples higher than those of most conventional pyroelectric samples?

Response-6

We are grateful for the reviewer's important comment.

In the main manuscript, Fig. 5g presents the pyroelectric coefficient (P_i) values for all devices, demonstrating a remarkable consistency of P_i in the range of 25-35 mC m⁻² K⁻¹. This result shows that the ZnO/NiO device can deliver 1000 times better P_i performance than the ZnO device.

These effects have two important features that arise from the heterojunction interface. Firstly, the built-in electric field is polar and induces additional polarization along the *c*-axis. This polarization is proportional to the square of the electric field through electrostriction.¹⁰ The electric-field *E* is calculated as V_{oc}/d , where *d* is the thickness of the heterostructure (approximately 100 nm). This value is determined to be 4×10^6 V m⁻¹. Secondly, the heterojunction partner provides a thermal reservoir due to the dissimilar thermal expansion coefficient. This can be engineered for dT/dt with an appropriate p-type partner to the ZnO pyroelectric absorber.³¹

The main manuscript discusses the pyroelectric coefficient of samples that contain a ZnO/NiO heterostructure. The results show that the pyroelectric coefficient of these samples is higher than that of most conventional pyroelectric samples. According to the authors, embedding a pyroelectric absorber into a photovoltaic heterojunction could lead to significant breakthroughs in conventional pyroelectric samples. This could help improve the pyroelectric energy harvesting methodology presented in this report.

Reference:

10. Yang, M. M. *et al.* Piezoelectric and pyroelectric effects induced by interface polar symmetry. *Nature* **584**, 377–381 (2020).

31. Nguyen, T. T., Kim, J., Yi, J. & Wong, C.-P. P. High-performing UV photodetectors by thermal-coupling transparent photovoltaics. *Nano Energy* **100**, 107504 (2022).

We are grateful to the reviewer for taking the time to provide us with insightful comments and suggestions. We have carefully considered all the comments and incorporated the suggestions into our revised manuscript, which has significantly improved the quality of our work. We hope the reviewer is satisfied with our responses and will consider our manuscript for possible publication in Nature Communications. Thank you for your support and feedback.

Reviewer #2 (Remarks to the Author):

In this manuscript, Dr. J. Kim and collaborators have prepared and characterized the pyroelectric-photovoltaic devices equipped with the ZnO/NiO heterostructure geometry. Some interesting results are presented. After the careful reading and comparison with previous work, however, I find that the current results lack enough novelty and have some main issues that are very important and addressed throughout the manuscript. At this form, I do not think this manuscript is suitable for publication in Nat. Commun. Here are some concerning issues:

Response

We sincerely thank the reviewer for taking the time to review our paper and providing constructive feedback to improve our manuscript. We have revised the manuscript by following the reviewer's suggestion in the following point-by-point responses.

In response to the general comment concerning the novelty, we would like to convey the significance of our research compared to the previous reports as follows:

The efficiency of energy conversion devices is limited by thermalization losses, especially for short wavelength photons in single junction photovoltaics. Our study proposes a single harvester pyroelectric absorber-embedded photovoltaic device array with a transparent framework to exceed this limit. By utilizing cyclic input energy and pyroelectricity, thermalization losses are converted into work done by the system, resulting in a power conversion efficiency (PCE) of 11.9% and incident-photon-to-current conversion efficiency (IPCE) of 200% of the illumination wavelength of 365 nm and intensity of $100 \mu\text{W cm}^{-2}$.

The proposed device array consists of a ZnO/NiO heterostructure, which can be grown at room temperature using large-scale sputtering methods. The reliability and consistency of the proposed performance have been demonstrated by 4×4 array configurations of a total of sixteen devices with spectral characteristics and thermal variations. Our investigation using atomic-level resolution of the device cross-section and photonic defects combined with device performances showed the boost of the pyroelectric coefficient, as measured using the Chynoweth method, of heterostructure first-order pyroelectric polarization by the rigid-ion displacement by the pulsed illumination.

We have included a polarization *c*-axis resolved high-resolution transmission electronic microscopy image of a pyroelectric absorber at various interfaces in the devices, which shows the consequences of the optical excitations upon the pulsed illumination during the power-generation cycles, referred to as pyroelectric-photovoltaic operation. Figure R2-1 summarizes the pyroelectric-photovoltaic device design and its performance. Our study contributes significantly to the literature because our proposed system can be a promising entity for the pyroelectric embedded photovoltaic device, providing performances beyond thermodynamic limits with high-speed and transparent features.

Figure R2-1. (a) Input energies for energy conversion device. (b) Schematic of the pyroelectric absorber-combined photovoltaic device and the original device array prototype. (c) Crystal structure of hexagonal ZnO, and polarization along the c -axis (pink and blue spheres present Zn and O atoms, and the orange stick is the covalent bond). High-resolution image showing the FTO/ZnO, ZnO film, and ZnO/NiO interface. (d) Schematic depiction of the device, the built-in electric field at ZnO/NiO heterojunction (black arrows), and the long-range electric field function of dT/dt . Current-voltage characteristic and power-voltage characteristic of the device under steady and pulsed light illumination. (e) Spectral characteristic and temperature effect on the PCE and IPCE performances.

1> From the design point of view, the authors have designed a heterostructure detector structure by stacking the ZnO/AgNW heterostructure on the NiO/ZnO heterojunction. As far as we know, this structure is a very common means of constructing the heterojunction devices and the authors have published many similar reports, such as *Nano Energy* 2022, 100, 107504; *Adv. Sci.* 2023, 10, 2303895. In the submitted work, the authors simply combine the idiomatic methods of the field, and the device performances obtained are not satisfactory for in *Nat. Commun.*

Response-1

We sincerely thank the reviewer for this comment on the unique design presented in our study. As mentioned in our research, the ZnO absorber layer embedded heterostructure design yields exceptional power generation through pyroelectric-photovoltaic phenomena. This design is an

example of how to develop a pyroelectric-absorber embedded energy device that can harvest energy from the pulsed illumination of ultra-low-intensity photon flux.

We clarify that this study establishes the design procedure, methodology, evaluation indicators, and understanding of the operation of such pyroelectric-photovoltaic phenomena for the first time. We also elaborate on the novelty of our work compared to our previous reports, as suggested by the reviewer. In our first report in *Nano Energy* 2022, 100, 107504, we disclosed high-performance UV photodetectors thermal-coupling transparent photovoltaics. We reported a responsivity of 0.98 A W^{-1} by regulating the heat flow through a suitable p-type Cu_2O layer with a pyroelectric-ZnO layer. In the second report in *Adv. Sci.* 10, 2303895 (2023), we disclosed the design of a field-embedded transparent electrode for high-performance transparent photovoltaics and heater devices. We proposed a broadband energy harvester by embedding an a-Si-based transparent photovoltaic device for onsite power generation and thermal regulation. Although both reports provide thermal applications of the TPV device, they do not address how the pyroelectric-absorber embedded heterostructure aids in energy harvesting from pulsed illumination for ultrahigh performances beyond the thermodynamic limit. In this context, our study rigorously designed and addressed this issue to present it to society.

Furthermore, we present a thoughtful experiment to clarify the design of the presented heterostructure of ZnO/NiO and stacking of AgNW/ZnO electrode. We resolved five designs of the structure in Figure R2-2, including the device schematic, current density-voltage (J-V) and power density-voltage (P-V) characteristic plots, and their performance parameters for the photovoltaic (PV) and pyroelectric-photovoltaic (PE-PV) phenomena. The heterojunction involving ZnO and NiO layer thickness was 100 and 20 nm, respectively (grown by magnetron sputtering as described in methods), while the heterostructure involving AgNW/ZnO had identical conditions with uniform drop-casting of AgNW and low power sputtered ZnO layer thickness of 12.5 nm (as disclosed in *J. Power Sources* 491, 229578 (2021)). The active area of the device was 16 mm^2 with a 3×3 array configurations. These devices were measured under the pulsed illumination wavelength of 365 nm, an intensity of 1.4 mW cm^{-2} , a pulse frequency of 60 Hz with a duty cycle of 50%, a scan speed of 0.5 V s^{-1} , and a sample interval of $10 \text{ }\mu\text{V}$. The illumination direction was from the FTO side, while Au pogo pin was applied from the top electrode to avoid possible parasitic absorption and artifacts from non-uniform illumination.

In Figure R2-2a, we show pyroelectric performance with energy harvesting by only a ZnO device with a power conversion efficiency (PCE) of 0.06%, which is solely governed by the pyroelectric phenomena. It is worth noting that photovoltaic phenomena are extremely difficult to observe. While the device with only the NiO layer, in Figure R2-2b, lacks a photoactivated response and characteristic plot, it seems to have an Ohmic nature. Figure R2-2c shows the heterojunction of ZnO/NiO without AgNW electrode, demonstrating the photovoltaic response with an open-circuit voltage (V_{oc}) of 0.579 V, which increased to 0.599 V by pyroelectric phenomena. However, due to the significantly poor collection of charges by a lack of a suitable electrode, a short-circuit current density (J_{sc}) value of $4.43\text{-}4.6 \text{ }\mu\text{A cm}^{-2}$ resulted in a power conversion efficiency of 0.046%. With a suitable AgNW electrode, the ZnO/NiO heterojunction offers a noticeable PCE of 6.97%, as shown in Figure R2-2d. However, the AgNW network is susceptible to thermal stimuli and easily deforms under thermal stress, weakening its electrical conduction (as demonstrated in *Adv. Sci.* 10, 2303895 (2023)). Over-coating the ZnO layer to the AgNW network provides outstanding stability with improved optical and electrical properties. Figure R2-2e shows the

performance of the ZnO/NiO heterojunction device with AgNW/ZnO heterostructure. The results clearly show the pyroelectric-photovoltaic phenomena with a PCE of 7.77%. We have also added the result and discussion of the structural evaluation as per the reviewer's comment to the revised manuscript.

Figure R2-2. Assessment of the device structure to enable pyroelectric-photovoltaic phenomena. (a) Glass/FTO/ZnO/AgNW/ZnO. (b) Glass/FTO/NiO/AgNW/ZnO. (c) Glass/FTO/ZnO/NiO. (d) Glass/FTO/ZnO/NiO/AgNW. (e) Glass/FTO/ZnO/NiO/AgNW/ZnO.

Changes made to the manuscript

We have conducted an evaluation of five device designs for their PV and PE-PV performances. To achieve this, we have further clarified the design of ZnO/NiO heterojunction and stacking of AgNW/ZnO heterostructure using optimum evaluation indicators (Supplementary Fig. 23). The ZnO and NiO layers had a thickness of 100 nm and 20 nm, respectively. The AgNW/ZnO had identical conditions with uniform drop-casting of AgNW and low power sputtered ZnO layer thickness of 12.5 nm. We observed pyroelectric performance with energy harvesting by only a ZnO device with a PCE of 0.06%, which is solely governed by the pyroelectric phenomena, while PV phenomena are extremely difficult to observe. A device with only the NiO layer lacks a photoactivated response and characteristic plot, and seems to have an Ohmic nature. The heterojunction of ZnO/NiO without an AgNW electrode shows the PV response with a V_{OC} of 0.579 V, which increased to 0.6 V by PE-PV phenomena. However, the J_{SC} value of 4.43-4.6 $\mu A\ cm^{-2}$ resulted in a PCE of 0.046%, as the collection of charges was significantly poor due to a lack of a suitable electrode. With a suitable AgNW electrode, the ZnO/NiO heterojunction offers a noticeable PCE of 6.97%. However, the AgNW network is susceptible to thermal stimuli and easily deforms under thermal stress, weakening its electrical conduction.⁵⁹ Over-coating the ZnO layer to the AgNW network provides outstanding stability with improved optical and electrical properties. As a result, the ZnO/NiO heterojunction device with AgNW/ZnO heterostructure, PE-PV phenomena with greater PCE, and IPCE performance persists.

Reference:

59. Patel, M., Kim, S. & Kim, J. Field-induced transparent electrode-integrated transparent solar cells and heater for active energy windows: Broadband energy harvester. *Adv. Sci.* **10**, 2303895 (2023).

2> From the perspective of mechanism, it is well known that the pyro-phototronic effect enhances photovoltaic energy harvesting, and there also have been many articles about this phenomenon (Nanomaterials 2023, 13, 1336). In addition, the ability of about 2.5 times photopyroelectric current to enhance the photovoltaic signal ($I_{pyro+photo}/I_{photo}$) is lower than many other studies (Nano Energy 2019, 62, 310–318). This result is not attractive for readers.

Response-2

We sincerely thank the reviewer for their valuable comments regarding the mechanism of the pyro-phototronic effect and its potential applications in signal enhancement. They suggested referring to the progress made in Nanomaterials 2023, 13, 1336 and Nano Energy 2019, 62, 310-318, which explore the use of pyroelectric material in the pyro-phototronic effect to harvest light energy for advanced photodetection, leading to enhanced photoresponses. However, it should be noted that none of these reports provide any information on the quantum efficiency of the pyro-phototronic effect, leaving the extent of the possible improvements in photodetection unexplored. The pyro-phototronic device that has been reported does exhibit decent photovoltaic phenomena, but the use of pyroelectricity significantly enhances the pyro-phototronic current by ($I_{pyro+photo}/I_{photo}$), as

the reviewer pointed out. Our current study presents the usefulness of a pyroelectric heterojunction device that can convert incidental photons to current with an efficiency greater than 100% and a PCE of above 12% under pulsed illumination with low intensities.

We have carried out detailed photon balance calculations to clarify the perspective mechanism and quantum efficiency to explain the IPCE values for the PV and PE-PV performances. We assume that each photon absorption generates an electron-hole pair, and their collection generates photocurrent. Under ideal conditions (perfect absorption, zero reflection, unit electron-hole pair generation, zero parasitic electrical losses, and recombination-less transport), the IPCE value is 100%. We can also calculate the photocurrent density for the zero-bias condition, called the short-circuit current density (J_{sc}). As this report discloses pyroelectric-photovoltaic power generation, we first confirm the accuracy of the IPCE values estimated from the PV segment. The IPCE values should match the calculated J_{sc} value and should not exceed it, providing photon balance.

The IPCE performances for the TPHD array devices are shown in Fig. 3j (main manuscript). We estimated these values for PV and PE-PV as 77% and 177%, respectively, from the J_{sc} values measured by the J-V characteristic plots under the illumination of 365 nm wavelength and intensity of $500 \mu\text{W cm}^{-2}$. This illumination provides a photon flux density of $9.17975 \times 10^{14} \text{ cm}^{-2} \text{ s}^{-1}$. Under ideal conditions, the photovoltaic device provides a J_{sc} value of $147.076 \mu\text{A cm}^{-2}$, which serves as a check value for the measured PV performances. Table R2-1 summarizes the detailed calculation for the photon-flux, calculated J_{sc} , measured J_{sc} of the sixteen array devices, and their calculated IPCE values. We found that the measured J_{sc} value for the PV performance remained below the theoretically calculated J_{sc} value, with an average value of $94.58 \mu\text{A cm}^{-2}$, resulting in an IPCE value of 64.25%. However, due to spontaneous polarization by the pyroelectric-photovoltaic device, the measured J_{sc} value for the PE-PV performances exceeded the calculated J_{sc} value with an average value of $239.6 \mu\text{A cm}^{-2}$, equivalent to an IPCE value of 162.77%. Our analysis shows that the charge density that participated in the spontaneous polarization by the pyroelectric-photovoltaic phenomenon is $9.05 \times 10^{14} \text{ cm}^{-2} \text{ s}^{-1}$. This value is close to the photon flux and suggests that photo-ionized impurities participate in spontaneous polarization by the pyroelectric-photovoltaic phenomena. We have included these results and analyses in the revised manuscript as suggested.

Furthermore, we have summarized the progress of pyroelectric phenomena for various applications in Table R2-2, which is included in the manuscript (Table 1). The table chronologically compares input energy, pyroelectric-absorber used, pyroelectric coefficient, figure of merit, PCE, and IPCE performances. This summary shows that the breakthrough finding demonstrates an IPCE value of 200% with a PCE of 11% using a pyroelectric-heterostructure of ZnO/NiO.

Table R2-1. Summary of the detailed photon balance calculation and measured J_{SC} and IPCE values for TPHD devices.

Photon flux	Calculated J_{SC}	Device	Measured J_{SC} $\mu A\ cm^{-2}$		Measured IPCE		Additional charges by pyroelectric ($cm^{-2}\ s^{-1}$)
			PV	PE-PV	PV	PE-PV	
$9.17975 \times 10^{14}\ cm^{-2}\ s^{-1}$	$147.076\ \mu A\ cm^{-2}$	D1	86.55	233.33	58.80	158.51	9.16×10^{14}
		D2	89.18	252.76	60.58	171.71	1.02×10^{15}
		D3	87.08	231.94	59.16	157.57	9.04×10^{14}
		D4	107.00	229.09	72.69	155.63	7.62×10^{14}
		D5	84.32	218.88	57.28	148.69	8.40×10^{14}
		D6	93.15	217.69	63.28	147.89	7.77×10^{14}
		D7	88.65	232.39	60.22	157.87	8.97×10^{14}
		D8	89.49	247.58	60.79	168.19	9.87×10^{14}
		D9	94.94	246.25	64.50	167.29	9.44×10^{14}
		D10	97.35	254.72	66.13	173.04	9.82×10^{14}
		D11	87.77	238.28	59.63	161.87	9.39×10^{14}
		D12	113.57	245.23	77.15	166.60	8.22×10^{14}
		D13	101.78	233.61	69.14	158.70	8.23×10^{14}
		D14	84.00	244.14	57.06	165.85	1.00×10^{15}
		D15	113.97	262.45	77.42	178.29	9.27×10^{14}
		D16	94.45	245.29	64.16	166.64	9.41×10^{14}
		Average	94.58	239.60	64.25	162.77	9.05×10^{14}

Table R2-2. Summary of the application evaluation based on the various pyroelectric phenomena for energy conversion, where P_i is the pyro-coefficient, FOM is the figure of merit, τ is the response speed, PCE is the power conversion efficiency, IPCE is the incident-photon-to-current conversion efficiency.

Application	Device	Pyroelectric phenomena	Input energy	P_i $\mu\text{C m}^{-2}$ K^{-1}	FOM $\frac{\text{cm}^2}{\mu\text{C}^{-1}}$	τ μs	PCE %	IPCE %	Year ref.
Thermal sensors	Triglycine sulfate	Pyroelectric	IR radiation	270	-	-	-	-	2005 16
Nuclear fusion	LiTaO ₃ crystal	Pyroelectric	Neutron flux	-	-	-	-	-	2005 21
UV Photodetector	FTO/ZnO/MAPbI ₃ /HTM/Cu	Light-induced pyroelectric	Pulsed light 325 nm	-	-	53	-	-	2015 17
Hybrid energy cell	PEDOTs	Photo-thermal-pyroelectric	NIR sunlight	-	-	-	11.7	-	2015 20
Bolometer Mid-infrared photodetector	LiNbO ₃ /graphene	Pyroelectric	Polarized quantum cascade LASER (6.2-10 μm)	780	-	-	-	-	2017 23
Energy harvesters	Pb(Mg _{1/3} Nb _{2/3})O ₃ -PbTiO ₃	Pyroelectric	Joule heating (25-115 °C)	550	-	-	-	-	2018 3
Hydrogen generation Water purification	Few layer black phosphorene	Pyro-catalytic	IR lamp thermal cycles (15-65 °C)	5,287	-	-	-	-	2018 19
Nanogenerator	PVDF film PZT ceramic	Pyroelectric	Thermal cycle using Peltier module (26-55 °C)	27,000	-	-	-	-	2019 18
Water purification	Pb(Zr _{0.52} Ti _{0.48})O ₃	Pyro-catalytic	Water-bath (15-70 °C)	605	-	-	-	-	2019 27 22
CO ₂ to methanol reduction	Bi ₂ WO ₆	Pyro-catalytic	Water-bath (15-70 °C)	-	-	-	-	-	2021 28
Hydrogen production	BaTiO ₃ -Au	Pyro-catalytic	Nanosecond LASER 532 nm, 786 mW/cm ²	300	-	-	-	-	2022 24
Tooth whitening	BaTiO ₃ nanowire	Pyro-catalysis	Oral temperature fluctuation	210	-	-	-	-	2022 29
Wide spectrum photodetector	Au/Molecular N-IBATFA/Cu	Photo-pyroelectric	Photon 266-1950 nm	69,000	188×10 ⁻²	-	-	-	2023 25
Slippery surfaces microrobots	SiO ₂ /LiNbO ₃	Photo-pyroelectric	LASER 808 nm, 100-1000 mW	83	-	-	-	-	2023 26
Pyro-photovoltaic	FTO/ZnO/NiO/AgNW/ZnO	Pyro-photovoltaic	Pulsed illumination 340-520 nm	25000-35000	58-121	<10	11	200	This work

Changes made to the manuscript

On Page 2

From the mechanism perspective, it is well known that the pyro-phototronic effect enhances photovoltaic energy harvesting by enhancing the pyro-photocurrent signal; its quantum efficiency and power conversion efficiency have been unknown to date.^{17,23–25}

Reference:

17. Wang, Z. L. Z. *et al.* Light-induced pyroelectric effect as an effective approach for ultrafast ultraviolet nanosensing. *Nat. Commun.* **6**, 8401 (2015).
23. You, D. *et al.* Photovoltaic-pyroelectric effect coupled broadband photodetector in self-powered ZnO/ZnTe core/shell nanorod arrays. *Nano Energy* **62**, 310–318 (2019).
24. Li, F., Peng, W., Wang, Y., Xue, M. & He, Y. Pyro-phototronic effect for advanced photodetectors and novel light energy harvesting. *Nanomaterials* **13**, 1336 (2023).
25. Nguyen, T. T., Kim, J., Yi, J. & Wong, C.-P. P. High-performing UV photodetectors by thermal-coupling transparent photovoltaics. *Nano Energy* **100**, 107504 (2022).

On Page 7

We conducted photon balance calculations to determine the IPCE values for the PV and PE-PV technologies. We assumed that each photon absorption results in an electron-hole pair, which generates photocurrent when collected. In perfect conditions where there is complete absorption, no reflection, perfect electron-hole pair generation, no parasitic electrical losses, and no recombination, the IPCE value should be 100%. Using this assumption and an estimated photon density, we can calculate the photocurrent density for the zero-bias condition, which is also known as (J_{sc}). We estimated photon density using the formula given below:

$$\text{Photon density} = \frac{I}{E_p} = \frac{I \times \lambda \times 10^{-9}}{hc} = \frac{I (W m^{-2}) \times \lambda \times 10^{-9} (m.s)}{1.988 \times 10^{-25} (J s m)} = I \times \lambda \times 5.03 \times 10^{15} (m^{-2} s^{-1}).$$

where, λ and I are the illumination wavelengths in m and intensity in $W m^{-2}$, respectively, h is the Planck constant ($6.63 \times 10^{-34} J s$) and c is the speed of light ($2.998 \times 10^8 m s^{-1}$).

First, we confirmed the accuracy of the IPCE values estimated from the PV segment, which should match the calculated J_{sc} value and should not exceed it, providing photon balance. The IPCE performances for the TPHD array devices are shown in Fig. 3j. We estimated these values for PV and PE-PV as 77% and 177%, respectively, from the J_{sc} values measured by the J-V characteristic plots under the illumination of 365 nm wavelength and intensity of $500 \mu W cm^{-2}$. This illumination provides a photon density of $9.17975 \times 10^{14} cm^{-2} s^{-1}$. Under ideal conditions, the PV operation provides a J_{sc} value of $147.076 \mu A cm^{-2}$, which serves as a check value for the measured PV performances. Table 2 summarizes the detailed calculation for the photon-flux, calculated J_{sc} ,

measured J_{SC} of the sixteen array devices, and their calculated IPCE values. Based on our analysis, we found that the measured J_{SC} value for the PV performance complies with the theoretically calculated J_{SC} value. The average value is $94.58 \mu\text{A cm}^{-2}$, resulting in an IPCE value of 64.25%. However, due to spontaneous polarization by the pyroelectric-photovoltaic device, the measured J_{SC} value for the PE-PV performances exceeded the calculated J_{SC} value with an average value of $239.6 \mu\text{A cm}^{-2}$, equivalent to an IPCE value of 162.77%. This analysis shows that the charge density that participated in the spontaneous polarization by the PE-PV phenomenon is $9.05 \times 10^{14} \text{cm}^{-2} \text{s}^{-1}$. This value is close to the photon flux and suggests that photo-ionized impurities participate in spontaneous polarization by the PE-PV phenomena.

3> The authors should describe why ZnO/AgNW/NiO/ZnO was chosen for the photovoltaic devices. Besides, the not-sharp cross-sections in Figure 1f may cause serious interface problems, which is important to control the device quality.

Response-3

We sincerely thank the reviewer for their valuable comment on ZnO/AgNW/NiO/ZnO structure choice. In response to the reviewer's comment, we would like to clarify that this structure is crucial and performs well in pyroelectric-photovoltaic phenomena. Based on this structural evaluation, we suggest that an n-type pyroelectric absorber, in combination with suitable heterojunction partners like p-type NiO, can provide ultra-high IPCE and PCE performance. We also propose that pyroelectric polar absorbers like GaN, CdS, In_2S_3 , BiFeO_3 , PbTiO_3 , and SrTiO_3 , when combined with a suitable heterojunction partner layer, can result in IPCE > 100% from ultra-low intense pulsed illumination with onsite power generation. This information is mentioned in the introduction of our manuscript with references cited. For ease of reference, we have summarized these references below.

- Yang, M. M. et al. Piezoelectric and pyroelectric effects induced by interface polar symmetry. *Nature* 584, 377–381 (2020).
- Jiang, J. et al. Giant pyroelectricity in nanomembranes. *Nature* 607, 480–485 (2022).
- Mondal, R., Hasan, M. A. M., Baik, J. M. & Yang, Y. Advanced pyroelectric materials for energy harvesting and sensing applications. *Mater. Today* 66, 273–301 (2023).
- Tian, S., Li, B., Dai, Y. & Wang, Z. L. Piezo-phototronic and pyro-phototronic effects enhanced broadband photosensing. *Mater. Today* 68, 254–274 (2023).
- Liu, J., Fernández-Serra, M. V. & Allen, P. B. First-principles study of pyroelectricity in GaN and ZnO. *Phys. Rev. B* 93, 081205 (2016).
- Yang, Y. et al. Pyroelectric nanogenerators for harvesting thermoelectric energy. *Nano Lett.* 12, 2833–2838 (2012).

In order to provide a complete view of the device structure, Fig. 1f in the manuscript displays a lower magnification image that shows various layers such as glass, FTO, ZnO, NiO, and AgNW. To analyze each layer in detail, a thorough examination was conducted to determine the *c*-axis orientation of individual layers of FTO, ZnO, and NiO (Fig. 2b-2e). The atomic layer spacing of

each layer was also measured over the 5-nm region, and the results are represented in Figure R2-3. The analysis indicates the presence of FTO, ZnO, ZnO/NiO, and NiO layers with different d -spacing values. The FTO layer (region-A) has a d -spacing value of 0.165 nm, while the ZnO layer has a d -spacing value of 0.264 nm at the FTO interface (region-B) and in the center region (region-C). The NiO layer (region-B) has a d -spacing value of 0.2337 nm. The manuscript provides a detailed discussion of these results for the readers.

Figure R2-3. Device cross-section analysis revolving the c -axis orientation of each layer of FTO (A), ZnO at FTO interface (B), ZnO center region (C), and NiO at ZnO interface (D) (Top panel in red highlight). The inter-atomic distance over the c -axis resolved region in the yellow box is profiled, confirming the d -spacing of 0.165 nm for the FTO (region-A), 0.264 nm of ZnO at FTO interface (B) and center region (C), and 0.2337 nm of the NiO (region-B).

Changes made to the manuscript

On Page 4

“...which shows a **device** interface of FTO/ZnO ...”

“...(HRTEM) **of the device** is depicted...”

4> The authors should add more characterization work on the device quality, such as adhesion, stability, etc

Response-4

We sincerely thank the reviewer for this valuable comment.

--Clarity on AgNW versus AgNW/ZnO electrode--

The hybrid design of the AgNW/ZnO top electrode offers outstanding stability of the device under thermal operation. We resolved the role of the silver nanowire (AgNW) and parasitic absorption by the ZnO-coated AgNW electrode. The fabricated devices with AgNW and AgNW/ZnO electrodes are shown in Figure R2-4a. Their absorbance profiles, shown in Figure R2-4b, confirm identical absorbance in the wavelength range of 250-500 nm, which is also evidenced by the photo-image of the sample in Figure R2-4a. We measured the performance of these devices under identical conditions of pyroelectric-photovoltaic, which are pulsed frequency of 60 Hz, scan speed of 0.5 V s^{-1} , and sample interval of $10 \text{ } \mu\text{V}$. The illumination wavelength was 365 nm, and the intensity was 1.4 mW cm^{-2} . The current density-voltage (J-V) characteristic plots in Figures R2-4a and R2-4b show that pyroelectric-photovoltaic performances clarified the usefulness of the AgNW-ZnO electrode design and offered reasonably enhanced performance. The power density-voltage (P-V) characteristic plots shown in Figure R2-3e and 2-3f show that TPHD with AgNW/ZnO electrode offers an output power density of $108.8 \text{ } \mu\text{W cm}^{-2}$ with a power conversion efficiency of 7.77%. The photovoltaic and pyroelectric-photovoltaic performance parameters of the devices with AgNW and AgNW/ZnO electrodes are summarized in Table R2-3.

--Stability issue of AgNW versus AgNW/ZnO electrode--

Furthermore, we would like to summarize the issue related to the operational stability of the AgNW network. When bare AgNWs are heated up to 250°C , the network resistance increases dramatically, as shown in Figure R2-5a.(Cell Rep. Phys. Sci. 2021, 2, 100591) After heating the sample for about 2 minutes, the resistance increased from $\sim 10 \text{ } \Omega$ to $\sim 10,000 \text{ } \Omega$. The morphology change of the bare AgNW is clearly shown in scanning electron microscope (SEM) images (Figure R2-5b), which show the loss of the AgNW network because of the Rayleigh instability and agglomeration of Ag nanospheres (spheroidization).

Figure R2-6 displays the impact of 250°C heating and subsequent cooling cycles on the morphology and electric current flow of ZnO-coated AgNWs under a constant electric bias. The SEM images of the samples indicate that the ZnO thin film coating can prevent the AgNW networks from changing their structure during repeated thermal cycles. The stable performance of TPHD is attributed to the hybrid structure of AgNW/ZnO, which can withstand temperature-dependent pyroelectric-photovoltaic phenomena (as shown in Figures 5c and 5d in the main manuscript).

--Adhesion test of the device by Kapton tape peeling--

We experimented to investigate the adhesion test using the Kapton tape peeling method. The results are summarized in Figure R2-7. We measured the J_{sc} profiles of TPHD devices with AgNW/ZnO electrodes before the Kapton peeling test. The results showed a J_{sc} value of 0.312 mA cm⁻² for the PV and 1.1125 mA cm⁻² for the PE-PV phenomenon. We measured performance under the pulsed illumination wavelength of 365 nm and intensity of 1.2 mW cm⁻², resulting in IPCE values of 88.47% and 314.95% for the PV and PE-PV phenomena.

During the first cycle of Kapton tape peeling from the device electrode, we observed some exfoliation of the AgNW network from the electrode border region due to the agglomeration of the AgNW bundles near the mask edge area by the drop casting, as shown in Figure R2-7a. However, we noticed greater consistency in the J_{sc} profiles throughout the Kapton peeling test. The TPHD device with AgNW/ZnO electrode provided outstanding performance beyond the thermodynamic limit (IPCE > 100%). The IPCE values as a function of Kapton tape peeling cycles are shown in Figure R2-7.

We have added this result and discussion to the revised manuscript as per the reviewer's suggestion.

Figure R2-4. (a) Transparent pyroelectric heterojunction device array with AgNW and AgNW/ZnO electrode and (b) absorbance spectra. Current density-voltage (J - V) characteristic plot of the TPHD with (c) AgNW and (d) AgNW/ZnO electrode. Power density-voltage (P - V) characteristic plot of the TPHD with (e) AgNW and (f) AgNW/ZnO electrode.

Table R2-3. Summary of the performance parameters of the device with AgNW and AgNW/ZnO electrode.

Parameters	AgNW		AgNW/ZnO	
	PV	PE-PV	PV	PE-PV
J_{SC} (mA cm^{-2})	0.347	0.47776	0.314	0.4754
V_{OC} (V)	0.404	0.4847	0.401	0.4982
P_{max} ($\mu\text{W cm}^{-2}$)	54.87	97.54	52.00	108.8
IPCE (%)	84.28	115.93	76.19	115.35
FF (%)	39.16	42.12	41.43	45.93
PCE (%)	3.92	6.97	3.73	7.77

Figure R2-5. Evolution of (a) the electrical resistance and (b) the morphology of bare AgNWs during thermal annealing of the samples up to 250°C. These data are from our earlier work (Cell Rep. Phys. Sci. 2021, 2, 100591).

Figure R2-6. Electric current and morphology variation of the ZnO-coated AgNWs during repeated heating and cooling cycles.

Figure R2-7. Adhesion test of the device with AgNW/ZnO electrode. (a) The original photo image of the device after the adhesion test by the Kapton tape. (b) The short-circuit current density (J_{sc}) profile under the pulse illumination wavelength of 365 nm and intensity of 1.2 mW cm^{-2} before the adhesion test. (b) The J_{sc} profiles of the device after Kapton tape peeling cycles from 1 to 8. (d) incident photon to current conversion efficiency (IPCE) of the device for the PV and PE-PV phenomena are presented as a function of peeling cycles. It is worth noting that the thermodynamic limit was set at 100% of the IPCE value.

Changes made to the manuscript

On Page 12

The AgNW/ZnO top electrode design offers excellent stability to the device under thermal operation due to its hybrid structure.⁵⁹ The absorption profiles of AgNW and ZnO-coated AgNW electrode in the wavelength range of 250-500 nm were identical, as confirmed by Supplementary Fig. 31. These devices exhibited similar performances of PE-PV under pulsed illumination. This confirms the usefulness of the AgNW/ZnO electrode design, which offers enhanced performance with a P_{\max} of $108.8 \mu\text{W cm}^{-2}$ and a PCE of 7.77% (Supplementary Table 3) for TPHD. We also conducted an adhesion test using the Kapton tape peeling method (Supplementary Fig. 32). Prior to the Kapton peeling test, we measured the J_{sc} profiles of TPHD devices with AgNW/ZnO electrodes, which showed a J_{sc} value of 0.312 mA cm^{-2} for the PV and $1.1125 \text{ mA cm}^{-2}$ for the PE-PV phenomenon. It resulted in IPCE values of 88.47% and 314.95% for the PV and PE-PV phenomena, respectively. During the first cycle of Kapton tape peeling from the device electrode, we observed some exfoliation of the AgNW network from the electrode border region due to the agglomeration of the AgNW bundles near the mask edge area by the drop-casting. However, we noticed greater consistency in the J_{sc} profiles throughout the Kapton peeling test. The TPHD device with AgNW/ZnO electrode provided outstanding performance beyond the thermodynamic limit (IPCE > 100%).

Reference:

59. Patel, M., Kim, S. & Kim, J. Field-induced transparent electrode-integrated transparent solar cells and heater for active energy windows: Broadband energy harvester. *Adv. Sci.* **10**, 2303895 (2023).

Overall, I think this manuscript is not suitable for the possible publication in Nat. Commun., and it should be submitted to a specialized journal.

Response

We sincerely thank the reviewer for taking the time to provide us with valuable feedback and suggestions. We have carefully considered all the comments and incorporated the suggestions into our revised manuscript, which has significantly improved the quality of our work. We hope the reviewer is satisfied with our responses and will consider our manuscript for possible publication in Nature Communications. Thank you for your support and feedback.

Reviewer #3 (Remarks to the Author):

In this paper, the authors attempt to use a photovoltaic heterojunction incorporated with a pyroelectric effect to efficiently generate electricity beyond thermodynamic limit. Specifically, the pyroelectric-photovoltaic devices utilize spontaneous polarization by pulsed light-induced thermal change to produce output power, which is further beyond the traditional photovoltaics. Moreover, through the adjustments of pulse frequency, scan speed, sample interval, illumination wavelength, temperature, and pyroelectric coefficient, this type of pyroelectric-photovoltaic devices can optimize power harvesting, achieving a high power conversion efficiency of 11.9% and an incident photon-to current conversion efficiency of 200%.

Despite the notable achievements presented by the authors, several critical issues have been identified, making it different for me to recommend this article for publication in this journal. Firstly, from my perspective, the authors may have overly exaggerated their results. Given that photovoltaic devices are typically evaluated under the AM1.5G spectrum, the reliance on pulsed light with specific wavelengths in this study raises concerns about the generalizability of the results. Additionally, the evaluation method requires further confirmation. Given the use of a pulsed light source, it is suggested that energy output calculations should be weighted based on the pulse on/off ratio rather than relying solely on the maximum value.

Responses

We sincerely thank the reviewer for taking the time to review our study and providing constructive feedback. We appreciate your positive comments which encourage us so much. Your detailed and thoughtful comments and advice are very helpful to us. We have revised the manuscript accordingly by following the reviewer's suggestion in the following point-by-point responses.

Technology comments:

1. In this study, the reliance on a specific pulse wavelength rather than operating under AM1.5g conditions should be explicitly highlighted. Consequently, this article should be viewed as a conceptual validation rather than a universal breakthrough of thermodynamic limits for traditional photovoltaic devices, preventing potential reader misinterpretation. The clarification also extends to the acknowledgment that the presented maximum efficiency of 12% is lower than the current highest efficiency of single-junction devices (>26%), reinforcing the conceptual nature of the study.

Response to technology comment-1

We are very grateful to the reviewers for their valuable suggestions. We explicitly highlight that the operation of the device is dependent on specific pulse wavelength and measurement parameters in order to harvest pyroelectric-photovoltaic power. We appreciate the reviewer's perspective that our work represents a conceptual breakthrough in pyroelectric-photovoltaic performance, and we have made it clear in both the abstract and conclusion to avoid any misinterpretation. It is important to note that our work does not aim to compare or claim the highest efficiency of the single junction device under standard illumination conditions of AM1.5G.

The discovery of light-emitting diodes (LEDs) based on group III/nitride semiconductors by Nakamura et al. has revolutionized energy-efficient lighting (Nature Photonics 3, 180-182 (2009)).

These LEDs emit light within a narrow spectral range of 400-750 nm and operate in pulsed mode, typically between 50-120 Hz (Infomat 3, 445-459 (2021) and <https://www.richtek.com/Design%20Support/Technical%20Document/AN022>). It is important to note that a pyroelectric-photovoltaic device is needed to capture the energy from indoor LED illumination, which operates under pulsed illumination with frequency. Such a device would be essential for creating a ubiquitous energy system for the Internet of Things, providing both energy harvesting and exceptional photosensing capabilities. Additionally, underwater exploration requires the device to operate within the illumination wavelength spectrum of 300-500 nm, as longer visible and infrared wavelengths are significantly attenuated by water. This device must be able to produce power and ultra-sensitive photodetection to facilitate underwater needs of onsite power, ultrasensitive UV and short visible wavelengths photodetection, and photo-communication (Nature Photonics 17, 747-754 (2023)). These requirements have been added to the introduction section of the revised manuscript to ensure the development of a high-performance pyroelectric-photovoltaic device.

Changes made to the manuscript

On Page 2

From the mechanism perspective, it is well known that the pyro-phototronic effect enhances photovoltaic energy harvesting by enhancing the pyro-photocurrent signal; its quantum efficiency and power conversion efficiency have been unknown to date.^{17,23-25}

Reference:

17. Wang, Z. L. Z. *et al.* Light-induced pyroelectric effect as an effective approach for ultrafast ultraviolet nanosensing. *Nat. Commun.* **6**, 8401 (2015).
23. You, D. *et al.* Photovoltaic-pyroelectric effect coupled broadband photodetector in self-powered ZnO/ZnTe core/shell nanorod arrays. *Nano Energy* **62**, 310–318 (2019).
24. Li, F., Peng, W., Wang, Y., Xue, M. & He, Y. Pyro-phototronic effect for advanced photodetectors and novel light energy harvesting. *Nanomaterials* **13**, 1336 (2023).
25. Nguyen, T. T., Kim, J., Yi, J. & Wong, C.-P. P. High-performing UV photodetectors by thermal-coupling transparent photovoltaics. *Nano Energy* **100**, 107504 (2022).

On Page 3

The discovery of light-emitting diodes (LEDs) based on group III/nitride semiconductors has transformed energy-efficient lighting.⁴⁰ These LEDs emit light in a narrow spectral range of 400-750 nm and operate in pulsed mode, typically between 50-200 Hz.⁴¹⁻⁴³ It is important to note that a pyroelectric-photovoltaic device is required to capture the energy from LED illumination, which operates under pulsed illumination with frequency. Such a device would be necessary for creating

a widespread energy system for the Internet of Things, providing both energy harvesting and exceptional photosensing capabilities. Additionally, for underwater exploration, the device must operate within the illumination wavelength spectrum of 300-500 nm, as longer visible and infrared wavelengths are significantly weakened by water. This device must be able to produce power and ultra-sensitive photodetection to meet underwater needs of onsite power, UV and short visible wavelengths photodetection, and photo-communication.⁴⁴

Reference:

17. Wang, Z. L. Z. *et al.* Light-induced pyroelectric effect as an effective approach for ultrafast ultraviolet nanosensing. *Nat. Commun.* **6**, 8401 (2015).
23. You, D. *et al.* Photovoltaic-pyroelectric effect coupled broadband photodetector in self-powered ZnO/ZnTe core/shell nanorod arrays. *Nano Energy* **62**, 310–318 (2019).
24. Li, F., Peng, W., Wang, Y., Xue, M. & He, Y. Pyro-phototronic effect for advanced photodetectors and novel light energy harvesting. *Nanomaterials* **13**, 1336 (2023).
25. Nguyen, T. T., Kim, J., Yi, J. & Wong, C.-P. P. High-performing UV photodetectors by thermal-coupling transparent photovoltaics. *Nano Energy* **100**, 107504 (2022).

2. Further clarification on the evaluation indicators is required. Considering the thermoelectric effect necessitates a pulse light source for excitation, it is crucial to incorporate the pulse on/off ratio in the calculation of maximum efficiency to eliminate potential ambiguities in the 200% photoelectric conversion.

Response to technology comment-2

We are grateful to the reviewers for their valuable suggestion, which has helped us understand an evaluation indicator much better. As the thermoelectric effect requires a pulsed illumination, we have designed a pulse on/off ratio to calculate the maximum efficiency and eliminate confusion in photoelectric conversion.

We regulate the pulse on/off ratio by varying the pulse width (PW) for the light-on interval within a fixed total signal period of 20 ms, as we use LED illumination. We have programmed duty cycles ranging from 0-100%, which are illustrated in Figure R3-1. At an intensity of 1.9 mW cm⁻² and a wavelength of 365 nm, we obtain a photon density of 3.48831 × 10¹⁵ cm⁻² s⁻¹. By controlling the duty cycle, we can regulate the photon dose within a range of 1.395 × 10¹² to 6.977 × 10¹³ cm⁻², as summarized in Figure R3-1c. Assuming 100% IPCE (incident photon to current efficiency) performance, the expected photocurrent density is summarized as 55.88 μA cm⁻² for a 10% duty cycle and 503.89 μA cm⁻² for a 90% duty cycle. We can calculate the photon density using the following formula:

$$\text{Photon density} = \frac{I}{E_p} = \frac{I \times \lambda \times 10^{-9}}{hc} = \frac{I (W m^{-2}) \times \lambda \times 10^{-9} (m s)}{1.988 \times 10^{-25} (J s m)} = I \times \lambda \times 5.03 \times 10^{15} (m^{-2} s^{-1}) .$$

where, λ and I are the illumination wavelengths in m and intensity in W m^{-2} , respectively, h is the Planck constant (6.63×10^{-34} J s) and c is the speed of light (2.998×10^8 m s⁻¹).

Figure R3-1. Experimental design on clarification on the pyroelectric power evaluation indicators. (a) The device under test incorporates the pulse on/off ratio employing a duty cycle. (b) Schematic showing the duty cycle calculation. (c) Summary of duty cycle, pulse width, photon density, and dose during complete pulse interval.

Figure R3-2 shows us the current density-voltage (J-V) characteristics of the TPHD device for different duty cycles. The experiment was conducted using an illumination wavelength of 365 nm, with an intensity of 1.9 mW cm^{-2} and a frequency of 50 Hz. The results demonstrate that the duty cycle range of 10-90% enables pyroelectric-photovoltaic energy conversion, which helps to distinguish the PV and PE-PV components. The power density-voltage characteristic plots of the duty cycle series are summarized in Figure R3-3, where the output power is maximum for the duty cycle value of 50% for the PE-PV segment, while it is consistent for the PV segment for the duty cycle from 20% to 98%.

The duty cycle controls the photon dose for the fixed interval of the pulse cycle, which is 2 ms. During the light-on pulse region, photon absorption activates the Frenkel reaction of the Zn interstitials, leading to the generation of ionized donor-type defects (Zn_i^+ and Zn_i^{++}) with activation energy of 50 meV and 150 meV. (Materials Today 10, 40-48 (2007)) This is likely to induce temperature change and cause the polarization of these ionized defects, providing pyroelectric components to the photovoltaic under the pulsed illumination. The photon dose triggers the polarization current, and the effect is prominent for the duty cycle of 2%, which is equivalent to the photon dose of $1.395 \times 10^{12} \text{ cm}^{-2}$.

The results obtained suggest that the duty cycle could serve as an evaluation indicator for pyroelectric-photovoltaic and photovoltaic performance parameters, including open-circuit voltage (V_{oc}), short-circuit current density (J_{sc}), maximum output power (P_{max}), fill factor (FF), incident-photon-to-current conversion efficiency (IPCE), and power conversion efficiency (PCE), as a function of the duty cycle. These parameters are shown in Figure R3-4. The data shows that a lower photon dose provides higher IPCE performances by PE-PV, attributed to the inevitable Frenkel reaction of the Zn interstitial, activated by the pulsed photon illumination, and their ionized species cause the polarization current. This result and discussion are added to the revised manuscript.

Figure R3-2. Current density-voltage (J-V) characteristics of the device under the pulsed light illumination of various duty cycles from 2% to 98%.

Figure R3-3. Power density-voltage (P-V) characteristics of the device under the pulsed light illumination of various duty cycles from 2% to 98%.

Figure R3-4. Summary of (a) open-circuit voltage (V_{oc}), (b) short-circuit current density (J_{sc}), (c) maximum output power (P_{max}), (d) fill factor (FF), (e) incident-photon-to-current conversion efficiency (IPCE), and (f) power conversion efficiency (PCE) as a function duty cycle of light illumination.

Changes made to the manuscript

On Page 9-10

“...and duty cycle (D). Analyzing these parameters can help identify their impact on the PE-PV performance.

The pyroelectric current, by spontaneous polarization upon light, can be defined as $J_{PL}(f, \omega, \Delta V, D)$ and should correspond to the spontaneous polarization of the up-on illumination state. To match the spontaneous polarization by pulse illumination, the first step was to test f from 1 to 200 Hz with $D = 50\%$,...”

On Page 9-10

In order to generate the PE-PV effect, a pulsed illumination is required. To evaluate the energy conversion, we varied the duty cycles by changing the pulse width of the light-on interval within a fixed total signal period of 20 ms, as shown in Supplementary Fig. 19. At an intensity of 1.9 mW cm⁻² and a wavelength of 365 nm, we achieved a photon density of 3.488×10^{15} cm⁻² s⁻¹. By controlling the duty cycle, we were able to maintain the photon dose within a range of 1.395×10^{12} to 6.977×10^{13} cm⁻². For 100% IPCE performance, the expected J_{SC} ranges from 55.88 μA cm⁻² for a 10% duty cycle to 503.89 μA cm⁻² for a 90% duty cycle.

The J-V characteristics of the TPHD device for different duty cycles are shown in Supplementary Fig. 20. The duty cycle range of 10-90% enables PE-PV energy conversion, which distinguishes the PV and PE-PV components. The P-V characteristic plots of the duty cycle series are summarized in Supplementary Fig. 21, where the output power is maximum for the duty cycle value of 50% for the PE-PV segment, while it remains consistent for the PV segment for the duty cycle range of 20% to 98%.

The duty cycle is responsible for controlling the amount of photons delivered during a fixed interval of the pulse cycle, which is typically 2 ms. During the light-on pulse region, the absorbed photons activate the Frenkel reaction of the Zn interstitials, leading to the creation of ionized donor-type defects (Zn_i^+ and Zn_i^{++}) with activation energy of 50 meV and 150 meV.⁵² This is likely to induce a temperature change and cause the polarization of these ionized defects, providing pyroelectric components to the photovoltaic under the pulsed illumination. The photon dose triggers the polarization current, and the effect is more prominent for a duty cycle of 2%, which is equivalent to a photon dose of 1.395×10^{12} cm⁻². Based on the obtained results, it can be concluded that the duty cycle can serve as an evaluation indicator for PE-PV and PV performance parameters as a function of the duty cycle. The data shows that a lower photon dose provides higher IPCE performances by PE-PV. This is attributed to the inevitable Frenkel reaction of the Zn interstitial, which is activated by the pulsed photon illumination, and their ionized species cause the polarization current.

During the fourth step of the experiment, duty cycles ranging from 0 to 100% were tested using the optimal values of f , ω , and Δv . The results, shown in Fig. 4g (obtained from the results summarized in Supplementary Figs. 20-22), indicate that a D value between 10-90% provides consistent performance for PE-PV operation. Power generation ability of the TPHD device is lost if D values are less than 5% for the given parameters. Therefore, it is recommended that D be kept around 50% for this setup to capture the power cycles by pulsed illumination. Among the four parameters, it is suggested to find the optimum pulse frequency for maximum P_{\max} and PCE (around 60 Hz with D around 50%) when ω is less than 1 V s⁻¹ and Δv is less than 10 μV.

Reference:

52. Schmidt-Mende, L. & MacManus-Driscoll, J. L. ZnO - nanostructures, defects, and devices. *Mater. Today* **10**, 40–48 (2007).

Other comments:

1. Role of silver nanowires: Here, silver nanowires doped with ZnO are used as the front-sided electrodes. The authors provide the function of silver nanowires, as it can generate significant parasitic absorption, thereby reducing the effective absorption and performance of the devices.

Response-1

We are grateful to the reviewer for this valuable comment.

We have clarified the role of the silver nanowire (AgNW) and parasitic absorption by the ZnO-coated AgNW electrode. We fabricated devices with AgNW and AgNW/ZnO electrodes, as shown in Figure R3-5a. Their absorbance profiles, shown in Figure R3-5b, confirm identical absorbance in the wavelength range of 250-500 nm, which is also evidenced from the photo-image of the sample in Figure R3-5a. We measured the performance of these devices under identical conditions of pyroelectric-photovoltaic, which are pulsed frequency of 60 Hz, scan speed of 0.5 V s^{-1} , and sample interval of $10 \mu\text{V}$. The illumination wavelength was 365 nm, and the intensity was 1.4 mW cm^{-2} . The J-V characteristic plots in Figures R3-5a and 3-5b show that pyroelectric-photovoltaic performances clarified the usefulness of AgNW-ZnO electrode design and offer reasonably enhanced performance. The power-density voltage characteristic plots shown in Figure R3-5e-f show that TPHD with AgNW/ZnO electrode offers an output power density of $108.8 \mu\text{W cm}^{-2}$ with a power conversion efficiency of 7.77%. The photovoltaic and pyroelectric-photovoltaic performance parameters of the devices with AgNW and AgNW/ZnO electrodes are summarized in Table R3-1.

Furthermore, we would like to summarize the issue related to the operational stability of the AgNW network. When bare AgNWs are heated up to 250°C , the network resistance increases dramatically, as shown in Figure R3-6a. (Cell Rep. Phys. Sci. 2021, 2, 100591) After heating the sample for about 2 minutes, the resistance increased from $\sim 10 \Omega$ to $\sim 10,000 \Omega$. The morphology change of the bare AgNW is clearly shown in scanning electron microscope (SEM) images (Figure R3-6b), which show the loss of the AgNW network because of the Rayleigh instability and agglomeration of Ag nanospheres (spheroidization).

Figure R3-7 shows the influence of 250°C heating and following cooling cycles on the electric current flow and morphology of the ZnO-coated AgNWs while maintaining a constant electric bias. The SEM images of the samples suggest that the ZnO thin film coating can prevent the AgNW networks from changing in morphology during repeated thermal cycles.

Figure R3-5. (a) Transparent pyroelectric heterojunction device array with AgNW and AgNW/ZnO electrode and (b) absorbance spectra. Current density-voltage (J-V) characteristic plot of the TPHD with (c) AgNW and (d) AgNW/ZnO electrode. Power density-voltage (P-V) characteristic plot of the TPHD with (e) AgNW and (f) AgNW/ZnO electrode.

Table R3-1. Summary of the performance parameters of the device with AgNW and AgNW/ZnO electrode

Parameters	AgNW		AgNW/ZnO	
	PV	PE-PV	PV	PE-PV
J_{SC} (mA cm ⁻²)	0.347	0.47776	0.314	0.4754
V_{OC} (V)	0.404	0.4847	0.401	0.4982
P_{max} (μW cm ⁻²)	54.87	97.54	52.00	108.8
IPCE (%)	84.28	115.93	76.19	115.35
FF (%)	39.16	42.12	41.43	45.93
PCE (%)	3.92	6.97	3.73	7.77

Figure R3-6. Evolution of (a) the electrical resistance and (b) the morphology of bare AgNWs during thermal annealing of the samples up to 250°C. This data are from our earlier work (Cell Rep. Phys. Sci. 2021, 2, 100591).

Figure R3-7. Electric current and morphology variation of the ZnO-coated AgNWs during repeated heating and cooling cycles.

Changes made to the manuscript

On Page 12

The AgNW/ZnO top electrode design offers excellent stability to the device under thermal operation due to its hybrid structure.⁵⁹ The absorption profiles of AgNW and ZnO-coated AgNW electrode in the wavelength range of 250-500 nm were identical, as confirmed by Supplementary Fig. 31. These devices exhibited similar performances of PE-PV under pulsed illumination. This confirms the usefulness of the AgNW/ZnO electrode design, which offers enhanced performance with a P_{\max} of $108.8 \mu\text{W cm}^{-2}$ and a PCE of 7.77% (Supplementary Table 3) for TPHD. We also conducted an adhesion test using the Kapton tape peeling method (Supplementary Fig. 32). Prior to the Kapton peeling test, we measured the J_{sc} profiles of TPHD devices with AgNW/ZnO electrodes, which showed a J_{sc} value of 0.312 mA cm^{-2} for the PV and $1.1125 \text{ mA cm}^{-2}$ for the PE-PV phenomenon. It resulted in IPCE values of 88.47% and 314.95% for the PV and PE-PV phenomena, respectively. During the first cycle of Kapton tape peeling from the device electrode, we observed some exfoliation of the AgNW network from the electrode border region due to the agglomeration of the AgNW bundles near the mask edge area by the drop-casting. However, we noticed greater consistency in the J_{sc} profiles throughout the Kapton peeling test. The TPHD device with AgNW/ZnO electrode provided outstanding performance beyond the thermodynamic limit (IPCE > 100%).

Reference:

59. Patel, M., Kim, S. & Kim, J. Field-induced transparent electrode-integrated transparent solar cells and heater for active energy windows: Broadband energy harvester. *Adv. Sci.* **10**, 2303895 (2023).

2. Element distributions: The issue of element ratios not totaling 100% within the specified range (i.e., 0-100 nm) in Figure 1g should be addressed, and an explanation for the absence of the Zn signal within this range should be provided. Additionally, the roughness of the EDS signal tested by TEM should be acknowledged, highlighting its limitations in accurately confirming the distribution of elements.

Response-2

We highly appreciate the reviewer for providing us with a valuable suggestion. We have noticed that the elemental ratios do not add up to 100% within 100 nm of the thickness range in Figure 1g. This is because C and Si elements are absent from the table. However, Table R3-2 shows that these elements have been considered, and their percentages add up to 100%. We have also confirmed some atomic percentages of Zn atoms within 100 nm, although they are not very clear.

Since the thickness of the top ZnO layer is only about 12 nm, the EDS signal step resolution of >6 nm does not provide a clear picture of the Zn concentration of such a thin layer. We have acknowledged this fact in the revised manuscript. Additionally, we have provided evidence that confirms the ZnO top-layer thickness of 12 nm, as summarized in Figure R3-8, which accurately depicts the distribution of Zn, O, and Ag elements.

We have included this information in the revised manuscript to make it more comprehensive.

Table R3-2. Summary of the elemental distribution of the TPHD with AgNW/ZnO top electrode. This distribution is shown up to 305 nm in depth.

Distance (nm)	C	O	F	Si	Ag	Sn	Ni	Zn	Total
1.4081	89.8	7	0	2.99	0	0.21	0	0	100
7.0406	92.34	5	0	1.86	0.31	0.5	0	0	100
12.673	88.42	10.28	0	1.13	0.02	0.15	0	0	100
18.306	94.2	4.36	0	0.39	1.02	0.03	0	0	100
23.938	98.42	0	0.52	0.06	0.8	0.18	0	0.03	100
29.571	92.24	3.03	0.28	0	3.55	0.9	0	0	100
35.203	71.65	5	0.1	0	22.68	0	0	0.57	100
40.836	61.27	0	0	0.25	36.95	1.53	0	0	100
46.468	49.14	10.4	0.72	0.78	38.16	0.79	0	0	100
52.101	45.99	7.08	0	0.13	46.06	0.74	0	0	100
57.733	57.59	0	0	0.64	41.77	0	0	0	100
63.366	72.27	4.16	0.77	0	22.43	0.32	0	0.06	100
68.998	86.1	0.71	0.73	0.62	11.72	0.12	0	0	100
74.631	93.67	4.97	0	0	1.12	0.14	0	0.1	100
80.263	98.53	0.61	0.49	0.05	0	0.32	0	0	100
85.896	92.94	5.16	0.29	0.48	0.78	0.36	0	0	100
91.528	91.5	7	0	0	0.12	1.27	0	0.1	100

97.161	92.15	6.39	0	0.63	0.02	0.14	0	0.67	100
102.79	91.86	7.02	0	0	0.79	0.33	0	0	100
108.43	44.25	30.79	2.85	0	9.48	0.47	12.15	0	100
114.06	11.81	43.11	4.5	0.24	9.77	1.03	29.48	0.06	100
119.69	19.87	47.16	0	0	11.4	0.92	17.12	3.54	100
125.32	8.76	54	0.56	0.23	7.92	0.62	4.44	23.46	100
130.96	0.63	52.92	0	0	2.64	1.32	0	42.49	100
136.59	0	54.78	0	0	1.11	0	0	44.11	100
142.22	5.43	56.42	0	0	0.25	0	0	37.9	100
147.85	10.41	55.56	0.89	0.57	0.7	0.45	0	31.42	100
153.49	0	52.08	4.76	0.84	1.12	0.33	0	40.88	100
159.12	5.04	45.6	1.55	0	0.48	1.73	0	45.6	100
164.75	4.45	58.81	0	0	0.21	0	0	36.53	100
170.38	9.12	54.34	0	0.22	0.37	0.43	0	35.53	100
176.02	8.87	55.12	0	0.14	0.41	0.52	0	34.93	100
181.65	7.33	56.33	0	0.22	0.26	0.61	0	35.25	100
187.28	9.96	50.97	1.76	0	0.16	0.59	0	36.56	100
192.91	19.55	49.18	0	0	0.6	0	0	30.67	100
198.55	0.46	60.97	0	1.55	1.02	0.31	0	35.69	100
204.18	0.15	62.51	0	0.04	1.01	0.1	0	36.19	100
209.81	3.81	52.66	0	0.86	0.48	0	0	42.18	100
215.44	2.55	54.1	0	0	0	0.59	0	42.75	100
221.08	9.22	49.52	1.85	0.79	0	11.73	0	26.88	100
226.71	0	66.68	0	0.09	0.03	29.56	0	3.64	100
232.34	4.21	63.86	5.09	1.95	0.1	24.78	0	0	100
237.97	0	65.66	0	0	1.62	32.71	0	0	100
243.61	0	61.7	0	0	0.13	38.17	0	0	100
249.24	0	67.35	0	0	0	32.65	0	0	100
254.87	0	66.1	3.78	0.07	0	30.05	0	0	100
260.5	0	63.25	0	0	0.1	36.65	0	0	100
266.14	0	64.23	0	0.22	2.4	31.46	0	1.69	100
271.77	1.11	65.68	0	1.61	0.5	31.1	0	0	100
277.4	0	66.08	0	1.39	0	32.53	0	0	100
283.03	0	66.38	0	0	0.74	32.65	0	0.23	100
288.67	0	59.14	0	0	2.28	37.43	0	1.16	100
294.3	0	67.7	0	0	1.02	30.23	0	1.05	100
299.93	0	60.14	0	2.73	0.27	36.86	0	0	100
305.56	0	57.93	2.68	1.05	0	38.34	0	0	100

Figure R3-8. (a) Specimen schematic of AgNW-ZnO electrode. (b) FESEM image of AgNW/ZnO. (c) High-angle annular dark-field imaging (HAADF) and elemental mapping of AgNW/ZnO (scale bar, 30 nm). These data were taken from our earlier work (Cell Rep. Phys. Sci. 2021, 2, 100591).

Changes made to the manuscript

On Page 4

We observed that the inclusion of C and Si elements required for the elemental ratios adds up to 100% (Supplementary Table 1). Since the thickness of the top ZnO layer is only about 12 nm, the EDS signal step resolution of >6 nm does not provide a clear picture of the Zn concentration for the AgNW/ZnO region. Instead, evidence confirms the ZnO top-layer thickness of 12 nm in Supplementary Fig. 1 accurately depicts the distribution of Zn, O, and Ag elements.

3. Preparation Details of ZnO and NiO: How are zinc oxide and nickel oxide prepared? Magnetron sputtering? Because the authors demonstrate high crystallinity of these materials, the details of the preparation need to be emphasized.

Response-3

We are grateful to the reviewer for this advice. We utilized magnetron sputtering to create the pyroelectric ZnO layer and its heterojunction using NiO. The ZnO film was produced through RF sputtering of the ZnO target, while NiO films were created by reactive DC sputtering of the Ni target. By controlling the O₂ gas flow rate to 5 sccm, we achieved a highly-doped p-type and nanocrystalline NiO film suitable for forming a heterostructure with the intrinsically n-doped ZnO.

(Sol. Energy Mater Sol. Cells, 170, 246-253 (2017)) Sputtering power played a crucial role in the process, with lower power leading to the formation of an amorphous film and higher power inducing lattice strain and atomic defect due to the higher kinetic energy of sputtering materials (Matter 4, 3549-3584 (2021); Tominaga et al., Chapter 6 Handbook of Sputtering Deposition Technology, Fundamentals and Applications for Functional Thin Films, Nanomaterials, and MEMS). A suitable sputtering power allowed for the formation of a dense film, and in this study, we opted for 150 W of RF power, which resulted in a better pyroelectric ZnO film. We have included this information in the revised manuscript.

Changes made to the manuscript

On Page 4

We used magnetron sputtering to create a pyroelectric ZnO layer and its heterojunction with NiO. We produced the ZnO film by RF sputtering of the ZnO target and the NiO films by reactive DC sputtering of the Ni target. By controlling the O₂ gas flow rate to 5 sccm, we achieved a highly-doped p-type and nanocrystalline NiO film suitable for forming a heterostructure with the intrinsically n-doped ZnO.⁴⁵ The sputtering power played a crucial role in the process, with lower power leading to the formation of an amorphous film and higher power inducing lattice strain and atomic defect due to the higher kinetic energy of sputtering materials.^{46,47} We opted for 150 W of RF power to enable the formation of a dense film, which resulted in a better pyroelectric ZnO film.

Reference:

45. Patel, M. et al. Excitonic metal oxide heterojunction (NiO/ZnO) solar cells for all-transparent module integration. Sol. Energy Mater. Sol. Cells **170**, 246–253 (2017).
46. Aydin, E. et al. Sputtered transparent electrodes for optoelectronic devices: Induced damage and mitigation strategies. Matter **4**, 3549–3584 (2021).
47. Wasa, K. Kanno, I. & Kotera, H. Handbook of sputter deposition technology: fundamentals and applications for functional thin films, nano-materials and MEMS. vol. 2 (William Andrew, Elsevier, 2012).

4. Ionization Energy A literature-supported explanation for obtaining activation energy values in Equations 3-6 should be provided.

Response-4

We are so grateful for your suggestion. A literature-supported explanation for obtaining activation energy values in equations 3-6 is provided and added in the revised manuscript.

“ Zn_i^{++} , Zn_i^+ , Zn_i^X , V_o^{++} , V_o^+ and V_o are the donor-type defects; while $V_{Zn}^{\circ\circ}$, and V_{Zn}° are the acceptor-type defects. The ionization energy of Zn interstitials and oxygen vacancies varies

between 0.05 and 2.8 eV. (Nanoscale, 2014, 6, 10224–10234; Mater. Today 10, 40–48 (2007); J. Appl. Phys. 98, 542 041301 (2005);)”

- Kayaci, F. et al. Role of zinc interstitials and oxygen vacancies of ZnO in photocatalysis: a bottom-up approach to control defect density. *Nanoscale* 6, 10224-10234 (2014).
- Özgür, Ü., et al. A comprehensive review of ZnO materials and devices. *J. Appl. Phys.* 98, 542 041301 (2005).
- Schmidt-Mende, L. & MacManus-Driscoll, J. L. ZnO - nanostructures, defects, and devices. *Mater. Today* 10, 40–48 (2007).

Changes made to the manuscript

On Page 4

The ionization energy of Zn interstitials and oxygen vacancies varies between 0.05 and 2.8 eV.⁵¹⁻

Reference:

51. Özgür, Ü. *et al.* A comprehensive review of ZnO materials and devices. *J. Appl. Phys.* **98**, 041301 (2005).
52. Schmidt-Mende, L. & MacManus-Driscoll, J. L. ZnO - nanostructures, defects, and devices. *Mater. Today* **10**, 40–48 (2007).
53. Kayaci, F., Vempati, S., Donmez, I., Biyikli, N. & Uyar, T. Role of zinc interstitials and oxygen vacancies of ZnO in photocatalysis: a bottom-up approach to control defect density. *Nanoscale* **6**, 10224–10234 (2014).

5. Working area As shown in Fig. 3a-3b, why the working region of the samples is cylindrical?

Response-5

We greatly appreciate the reviewer’s comment. To clarify the importance of the working region, we conducted an experiment to evaluate the optimal shape of the top electrode. We tested circular, square, rectangular, and pentagon shapes and designed a top electrode mask for each shape, as shown in Figure R3-9a. We created 1 × 4 arrays of each shape, providing an accurate assessment on a device that measures 25 × 25 mm² and has an active area of 10 mm². The design parameters resulted in areas of 10.005, 9.985, 10.01, and 9.99 mm² for the circular, square, rectangular, and pentagon shapes, respectively, with a standard deviation error of only 1%. The original mask and device array we fabricated are shown in Figure R3-9b.

We obtained the current density-voltage (J-V) characteristics of the unit cell under identical conditions (pulsed frequency of 60 Hz, scan speed of 0.5 V s⁻¹, sample interval of 10 μV,

illumination wavelength of 365 nm, and intensity of $400 \mu\text{W cm}^{-2}$), as shown in Figure R3-9c. We also summarized the power density-voltage (P-V) characteristic plots in Figure R3-9d. The results showed consistent pyroelectric-photovoltaic power generation from the device with various electrode shapes. Moreover, the power conversion efficiency (PCE) and incidental photon-to-current conversion efficiency (IPCE) were both enhanced by pyroelectric-photovoltaic, specifically $\text{IPCE} > 100\%$. We summarized the performance parameters as a function of specific shapes in Table R3-3. We concluded that there is no special requirement for the working region's shape to achieve pyroelectric-photovoltaic performance. As suggested, we have included these results and discussions in the revised manuscript.

Figure R3-9. (a) Schematic showing the mask design of top electrode array of circular, square, rectangular, and pentagon shapes with a target area of 10 mm^2 . (b) Original photo-image of the mask fabricated using the laser patterning and THPD array with various shapes of the top electrode. (c) Current density-voltage and (d) power density-voltage characteristic plot of the devices with various shapes of the top electrode.

Table R3-3. Summary of the performance parameters of the TPHD with AgNW/ZnO top electrode of circular, square, rectangular, and pentagon shapes with identical area.

Parameters	Top electrode shape of area 10 mm ²							
	Circular		Square		Rectangular		Pentagon	
	PV	PE-PV	PV	PE-PV	PV	PE-PV	PV	PE-PV
J _{SC} (mA cm ⁻²)	0.0977	0.1661	0.103	0.1614	0.0994	0.1576	0.1085	0.1656
V _{OC} (V)	0.366	0.481	0.372	0.468	0.374	0.4795	0.3799	0.486
P _{max} (μW cm ⁻²)	16.22	37.05	17.4	33.91	16.45	35.06	18.34	35.97
IPCE (%)	82.98	141.07	87.48	137.08	84.42	133.85	92.15	140.65
FF (%)	45.36	46.38	45.41	44.89	44.26	46.39	44.49	44.69
PCE (%)	4.06	9.26	4.35	8.48	4.11	8.77	4.58	8.99

Changes made to the manuscript

On Page 7

An experiment was conducted to evaluate the importance of the shape of the working region. Various shapes of top electrodes, namely circular, square, rectangular, and pentagon, were examined, and linear arrays of each shape's active area of 10 mm² with a standard deviation error of only 1% were used (Supplementary Fig. 7). The results showed consistent PE-PV power generation from the device with different electrode shapes. Moreover, the PCE and IPCE were improved by PE-PV, particularly IPCE > 100%, as summarized in Supplementary Table 2. This summary implies that there is no requirement for the working region's shape to achieve PE-PV performance.

6. PE and PV current In Fig. 3c, how to distinguish the currents of PE and PV components?

Response-6

We are very grateful to the reviewer for this comment. In order to distinguish the current of the PE and PV components in Fig. 3c, the duty cycle of the pulsed illumination needs to be varied. As stated in our response to technology comment-2, the evaluation indicator is the duty cycle of the pulse illumination. By using pulse illumination of 50 Hz frequency with selective duty cycles, such as 2%, 50%, and 98%, we can classify the currents, voltage, and output power of PE and PV components, as shown in Figure R3-10.

To further clarify the PE and PV components identification for the pyroelectric heterojunction device, we designed a transparent device with ZnO/NiO and TiO₂/NiO and measured it under identical conditions. The device schematic and its performance characteristics are displayed in Figure R3-11. We chose anatase-TiO₂ because of its centrosymmetric nature with n-type conduction, which is suitable for providing photovoltaic characteristics with NiO layer. This has been disclosed in our previous study (J. Mater. Chem. C. 11, 14559-14570 (2023)). Under identical conditions of 50% duty cycles and pulsed illumination of 60 Hz, J-V and P-V characteristics demonstrate distinguishing natures of centrosymmetric and non-centrosymmetric absorbers. This result clarifies the pyroelectric-photovoltaic power generation by the pyroelectric-absorber embedded heterojunction. As per the reviewer's comment, this result has been provided in the revised manuscript.

Figure R3-10. Distinguish the current, voltage, and power of PE and PV components using the duty cycle control, in which the duty cycle >98% clarifies for PV, 5-90% for PE-PV, and <2% for the steady-dark conditions. The PE-PV components enhance the power output for the duty cycle of 50%. This result summarized the pulsed illumination frequency of 50 Hz and light intensity of 1.9 mW cm⁻².

Figure R3-11. Distinguish the current, voltage, and output power of PE and PV components using the centrosymmetric and non-centrosymmetric absorber. Device schematic, J-V and P-V characteristic plots of the heterojunction device of (a) TiO₂-anatase and (b) ZnO absorbers. The anatase-TiO₂ layer was grown using the conditions given in our previous study (*J. Mater. Chem. C*, 11, 14559-14570 (2023)). Pulsed illumination has a duty cycle of 50, frequency of 60 Hz, and intensity of 570 μW cm⁻². The performance parameters of both devices are summarized in the inset in the P-V characteristic plot.

Changes made to the manuscript

On Page 8

To clarify the PE-PV and PV components for the device, a transparent device with ZnO/NiO and TiO₂/NiO was designed and measured under identical conditions, as shown in Supplementary Fig. 8. The anatase-TiO₂ was used because of its centrosymmetric nature with n-type conduction, which is suitable to provide photovoltaic characteristics with NiO layer.⁵⁶ Under identical conditions of 50% duty cycles and pulsed illumination of 60 Hz, their J-V and P-V characteristics demonstrate distinguishing natures of centrosymmetric and non-centrosymmetric absorbers. This result clarifies the PE-PV power generation by the pyroelectric-absorber embedded heterojunction.

Reference:

56. Patel, M. *et al.* A study of the optical properties of wide bandgap oxides for a transparent photovoltaics platform. *J. Mater. Chem. C* **11**, 14559–14570 (2023).

7. Error bars A detailed explanation for the significant difference in error bars between P_{max} and power conversion efficiency in Fig. 4d-4f should be provided.

Response-7

We are very grateful for the valuable comments provided by the reviewer. In our study, we estimated the maximum output power (P_{max}) and power conversion efficiency (PCE) values in Fig. 4d-4f from the J-V and P-V characteristic plots of the device. In order to explain the error bars, we have presented the characteristic plots that study the evaluation parameters of illumination frequency, scan speed, and sample interval in Figure R3-12 to R3-20. These plots show the PE-PV and PV components as a function of these evaluation parameters. We assigned the error bar scale of 2.5% to P_{max} and PCE values for the frequency function in Fig. 4d, while it is 5% for the scan-speed and sample interval functions. This result is intended to support the inclusion of error bars in P_{max} and PCE values presented in Fig. 4d-4f. We have added more details to the revised manuscript to supplement this result.

Figure R3-12. Current density-voltage characteristics of the TPHD PV device for various pulsed frequencies of light illumination with wavelengths of 365 nm and intensity of 2 mW cm^{-2} . (a) 0 Hz, (b) 0.5 Hz, (c) 10 Hz, (d) 20 Hz, (e) 40 Hz, (f) 60 Hz, (g) 80 Hz, (h) 100 Hz, (i) 150 Hz, (j) 200 Hz. Throughout the measurements, scan speed and sample interval were 2 V s^{-1} and $100 \mu\text{V}$, respectively.

Figure R3-13. Power density-voltage characteristics of the TPHD device for various pulsed frequencies of light illumination with wavelengths of 365 nm and intensity of 2 mW cm^{-2} . (a) 0 Hz, (b) 0.5 Hz, (c) 10 Hz, (d) 20 Hz, (e) 40 Hz, (f) 60 Hz, (g) 80 Hz, (h) 100 Hz, (i) 150 Hz, (j) 200 Hz. Throughout the measurements, scan speed and sample interval were 2 V s^{-1} and $100 \mu\text{V}$, respectively. These plots show output power for load voltage. The peak value of the power plot shows the maximum output power and, hence, maximum voltage and current density.

Figure R3-14. Summary of the performance parameters of the TPHD device as a function of pulse illumination frequency (f). (a) V_{oc} vs. f . (b) J_{sc} vs. f . (c) FF vs. f . These parameters were obtained from the J-V and P-V characteristic plots in Figures R3-12 and R3-13.

Figure R3-15. Current density-voltage characteristics of the TPHD device for various scan rates. (a) 0.25 V s^{-1} , (b) 0.5 V s^{-1} , (c) 1 V s^{-1} , (d) 1.5 V s^{-1} , (e) 2.5 V s^{-1} , (f) 3.5 V s^{-1} , and (g) 5 V s^{-1} . The sample interval was $3 \mu\text{V}$. The light illumination of the wavelength, intensity, and pulsed frequency was 365 nm , 2 mW cm^{-2} , and 60 Hz , respectively.

Figure R3-16. Power density-voltage characteristics of the TPHD device for various scan rates. (a) 0.25 V s^{-1} , (b) 0.5 V s^{-1} , (c) 1 V s^{-1} , (d) 1.5 V s^{-1} , (e) 2.5 V s^{-1} , (f) 3.5 V s^{-1} , and (g) 5 V s^{-1} . The sample interval was $3 \mu\text{V}$. The light illumination of the wavelength, intensity, and pulsed frequency was 365 nm , 2 mW cm^{-2} , and 60 Hz , respectively. These plots show output power for load voltage. The peak value of the power plot shows the maximum output power and, hence, maximum voltage and current density.

Figure R3-17. Summary of the performance parameters of the TPHD device as a function of the scan speed of current-voltage characteristics. (a) V_{oc} , (b) J_{sc} , and (c) FF versus scan speed in $V s^{-1}$. These parameters were obtained from the J-V and P-V characteristic plots in Figures R3-15 and R3-16.

Figure R3-18. Current density-voltage characteristics of the TPHD device for various sample intervals. (a) 3 μV , (b) 5 μV , (c) 10 μV , (d) 50 μV , (e) 100 μV , (f) 500 μV , (g) 1 mV, and (h) 10 mV. Throughout these measurements, scan speed was 2 V s^{-1} . The light illumination of the wavelength, intensity, and pulsed frequency was 365 nm, 2 mW cm^{-2} , and 60 Hz, respectively.

Figure R3-19. Power-voltage characteristics of the TPHD device for various sample intervals. (a) 3 μV , (b) 5 μV , (c) 10 μV , (d) 50 μV , (e) 100 μV , (f) 500 μV , (g) 1 mV, and (h) 10 mV. Throughout these measurements, scan speed was 2 V s^{-1} . The light illumination of the wavelength, intensity, and pulsed frequency was 365 nm, 2 mW cm^{-2} , and 60 Hz, respectively. These plots show output power for load voltage. The peak value of the power plot shows the maximum output power and, hence, maximum voltage and current density.

Figure R3-20. Summary of the performance parameters of the TPHD device as a function of the sample interval of current density-voltage characteristics. (a) V_{OC} , (b) J_{SC} , and (c) FF versus sample interval in $\times 10^{-5}$ V. These parameters were obtained from the J-V and P-V characteristic plots shown in Figures R3-18 and R3-19.

Changes made to the manuscript

On Page 9

“...Fig. 4d illustrates the relationship between P_{max} and PCE as a function of f in Hz (obtained from the results summarized in Supplementary Figs. 10-12), indicating its crucial role...”

“... P_{max} and PCE as a function of ω in $V s^{-1}$ (obtained from the results summarized in Supplementary Figs. 13-15)...”

“...The results of this test, shown in Fig. 4f (obtained from the results summarized in Supplementary Figs. 16-18), indicate that a large Δv value leads to a loss of ...”

8. The authors should provide a detailed explanation for the phenomenon where incident-photon-to-current conversion efficiency (IPCE) exceeds 100%.

Response-8

We are grateful for the reviewer's important comment and interest. We have carried out detailed photon balance calculations to explain the IPCE values for the PV and PE-PV performances. We assume that each photon absorption generates an electron-hole pair, and their collection generates photocurrent. Under ideal conditions (perfect absorption, zero reflection, unit electron-hole pair generation, zero parasitic electrical losses, and recombination-less transport), the IPCE value is 100%. We can also calculate the photocurrent density for the zero-bias condition, which is called the short-circuit current density (J_{SC}). As this report discloses pyroelectric-photovoltaic power generation, we first confirm the accuracy of the IPCE values estimated from the PV segment. The

IPCE values should match the calculated J_{sc} value and should not exceed it, providing photon balance.

The IPCE performances for the TPHD array devices are shown in Fig. 3j (main manuscript). We estimated these values for PV and PE-PV as 77% and 177%, respectively, from the J_{sc} values measured by the J-V characteristic plots under the illumination of 365 nm wavelength and intensity of $500 \mu\text{W cm}^{-2}$. This illumination provides a photon flux density of $9.17975 \times 10^{14} \text{ cm}^{-2}$. Under ideal conditions, the photovoltaic device provides a J_{sc} value of $147.076 \mu\text{A cm}^{-2}$, which serves as a check value for the measured PV performances. Table R3-4 summarizes the detailed calculation for the photon-flux, calculated J_{sc} , measured J_{sc} of the sixteen array devices, and their calculated IPCE values. We found that the measured J_{sc} value for the PV performance remained below the theoretically calculated J_{sc} value, with an average value of $94.58 \mu\text{A cm}^{-2}$, resulting in an IPCE value of 64.25%. However, due to spontaneous polarization by the pyroelectric-photovoltaic device, the measured J_{sc} value for the PE-PV performances exceeded the calculated J_{sc} value with an average value of $239.6 \mu\text{A cm}^{-2}$, equivalent to an IPCE value of 162.77%. Our analysis shows that the charge density that participated in the spontaneous polarization by the pyroelectric-photovoltaic phenomenon is $9.05 \times 10^{14} \text{ cm}^{-2} \text{ s}^{-1}$. This value is close to the photon flux and suggests that photo-ionized impurities participate in spontaneous polarization by the pyroelectric-photovoltaic phenomena. We have included these results and analyses in the revised manuscript as suggested.

Table R3-4. Summary of the detailed photon balance calculation and measured J_{sc} and IPCE values for TPHD devices.

Photon flux	Calculated J_{sc}	Device	Measured J_{sc} $\mu\text{A cm}^{-2}$		Measured IPCE		Additional charges by pyroelectric ($\text{cm}^{-2} \text{ s}^{-1}$)
			PV	PE-PV	PV	PE-PV	
$9.17975 \times 10^{14} \text{ cm}^{-2} \text{ s}^{-1}$	$147.076 \mu\text{A cm}^{-2}$	D1	86.55	233.33	58.80	158.51	9.16×10^{14}
		D2	89.18	252.76	60.58	171.71	1.02×10^{15}
		D3	87.08	231.94	59.16	157.57	9.04×10^{14}
		D4	107.00	229.09	72.69	155.63	7.62×10^{14}
		D5	84.32	218.88	57.28	148.69	8.40×10^{14}
		D6	93.15	217.69	63.28	147.89	7.77×10^{14}
		D7	88.65	232.39	60.22	157.87	8.97×10^{14}
		D8	89.49	247.58	60.79	168.19	9.87×10^{14}
		D9	94.94	246.25	64.50	167.29	9.44×10^{14}
		D10	97.35	254.72	66.13	173.04	9.82×10^{14}
		D11	87.77	238.28	59.63	161.87	9.39×10^{14}
		D12	113.57	245.23	77.15	166.60	8.22×10^{14}
		D13	101.78	233.61	69.14	158.70	8.23×10^{14}
		D14	84.00	244.14	57.06	165.85	1.00×10^{15}
		D15	113.97	262.45	77.42	178.29	9.27×10^{14}
		D16	94.45	245.29	64.16	166.64	9.41×10^{14}
Average			94.58	239.60	64.25	162.77	9.05×10^{14}

Changes made to the manuscript

On Page 7

We conducted photon balance calculations to determine the IPCE values for the PV and PE-PV technologies. We assumed that each photon absorption results in an electron-hole pair, which generates photocurrent when collected. In perfect conditions where there is complete absorption, no reflection, perfect electron-hole pair generation, no parasitic electrical losses, and no recombination, the IPCE value should be 100%. Using this assumption and an estimated photon density, we can calculate the photocurrent density for the zero-bias condition, which is also known as (J_{sc}). We estimated photon density using the formula given below:

$$\text{Photon density} = \frac{I}{E_p} = \frac{I \times \lambda \times 10^{-9}}{hc} = \frac{I (\text{W m}^{-2}) \times \lambda \times 10^{-9} (\text{m s})}{1.988 \times 10^{-25} (\text{J s m})} = I \times \lambda \times 5.03 \times 10^{15} (\text{m}^{-2} \text{s}^{-1}).$$

where, λ and I are the illumination wavelengths in m and intensity in W m^{-2} , respectively, h is the Planck constant (6.63×10^{-34} J s) and c is the speed of light (2.998×10^8 m s^{-1}).

First, we confirmed the accuracy of the IPCE values estimated from the PV segment, which should match the calculated J_{sc} value and should not exceed it, providing photon balance. The IPCE performances for the TPHD array devices are shown in Fig. 3j. We estimated these values for PV and PE-PV as 77% and 177%, respectively, from the J_{sc} values measured by the J-V characteristic plots under the illumination of 365 nm wavelength and intensity of $500 \mu\text{W cm}^{-2}$. This illumination provides a photon density of $9.17975 \times 10^{14} \text{ cm}^{-2} \text{ s}^{-1}$. Under ideal conditions, the PV operation provides a J_{sc} value of $147.076 \mu\text{A cm}^{-2}$, which serves as a check value for the measured PV performances. Table 2 summarizes the detailed calculation for the photon-flux, calculated J_{sc} , measured J_{sc} of the sixteen array devices, and their calculated IPCE values. Based on our analysis, we found that the measured J_{sc} value for the PV performance complies with the theoretically calculated J_{sc} value. The average value is $94.58 \mu\text{A cm}^{-2}$, resulting in an IPCE value of 64.25%. However, due to spontaneous polarization by the pyroelectric-photovoltaic device, the measured J_{sc} value for the PE-PV performances exceeded the calculated J_{sc} value with an average value of $239.6 \mu\text{A cm}^{-2}$, equivalent to an IPCE value of 162.77%. This analysis shows that the charge density that participated in the spontaneous polarization by the PE-PV phenomenon is $9.05 \times 10^{14} \text{ cm}^{-2} \text{ s}^{-1}$. This value is close to the photon flux and suggests that photo-ionized impurities participate in spontaneous polarization by the PE-PV phenomena.

We are grateful to the reviewer for taking the time to provide us with insightful comments and valuable suggestions. We have carefully considered all the comments and incorporated the suggestions into our revised manuscript, which has significantly improved the quality of our work. We hope the reviewer is satisfied with our responses and will consider our manuscript for possible publication in Nature Communications. Thank you for your support and feedback.

REVIEWERS' COMMENTS

Reviewer #1 (Remarks to the Author):

I am the first reviewer, and still I believe this research, to some extent, provides a potential approach for pyroelectric-photovoltaic power generation. However, its novelty is still not clearly presented as indicated by all the reviewers. In addition, in-depth polishing of the language is required. I will leave it to the editor Whether or not it meets the threshold expected for Nature Communications.

Reviewer #2 (Remarks to the Author):

In the revision, the authors have rebutted or replied to the reviewer's comments of heterostructure detector structure, pyro-phototronic mechanism. The similar concerns have been reported by the same team or other research groups, as demonstrated in many literatures. Although there are some improvements in this work, I don't agree with the statement "this study establishes the design procedure, methodology, evaluation indicators, and understanding of the operation of such pyroelectric-photovoltaic phenomena for the first time".

In addition, the spectral measurements were performed under the 365 nm illumination (100uW/cm²), almost falling in the absorption range of heterostructure device. If pyroelectricity plays an important role, the device might work in a wide spectrum region beyond absorption. such as near-IR or even far. Is it possible?

Reviewer #3 (Remarks to the Author):

I have gone through the revised manuscript and the response letter from the authors. In my opinion, the authors have satisfactorily addressed the comments from the reviewers. The new version of the manuscript can be recommended for publication in Nature Communications.

Response to Reviewer's Comments

Research Article no.: NCOMMS-23-56542A-Z

TITLE: Transparent Pyroelectric-Combined Photovoltaic Structure for Power Generation approaching Thermodynamic Limit

We appreciate the reviewers who took the time to review the study and provided constructive feedback. The detailed and thoughtful comments and suggestions are helpful for the authors to improve their work.

The manuscript has been revised following the reviewers' advice, and the changes are highlighted in yellow. The responses to the reviewer's comments are presented below in blue fonts.

Response to the reviewer	Page numbers
Reviewer#1	2
Reviewer#2	3-5
Reviewer#3	6

Reviewer #1 (Remarks to the Author):

I am the first reviewer, and still I believe this research, to some extent, provides a potential approach for pyroelectric-photovoltaic power generation. However, its novelty is still not clearly presented as indicated by all the reviewers. In addition, in-depth polishing of the language is required. I will leave it to the editor Whether or not it meets the threshold expected for Nature Communications.

Response

We sincerely thank the reviewer for reviewing the revised manuscript along with the 1st response letter. Based on the reviewer’s valuable comments, the language issue has been reviewed in the Nature Research Editing Service (editing certificate is shown in Figure R1-1).

Figure R1-1. Language editing certificate.

We revised some parts of the manuscript to satisfy the threshold (hopefully beyond) of criteria of Nature communications.

Furthermore, the title was revised as

“Transparent integrated pyroelectric-photovoltaic structure for photo-thermo hybrid power generation”

We hope that the reviewer can find the possibility of the publication in Nature Communication through the revised manuscript and response letter.

Thank you for your support and feedback.

Reviewer #2 (Remarks to the Author):

In the revision, the authors have rebutted or replied to the reviewer's comments of heterostructure detector structure, pyro-phototronic mechanism. The similar concerns have been reported by the same team or other research groups, as demonstrated in many literatures. Although there are some improvements in this work, I don't agree with the statement "this study establishes the design procedure, methodology, evaluation indicators, and understanding of the operation of such pyroelectric-photovoltaic phenomena for the first time".

Response

We sincerely thank the reviewer for reviewing the manuscript and response letter to provide valuable comments.

For the 1st round of revision, we mentioned that “this study establishes the design procedure, methodology, evaluation indicators, and understanding of the operation of such pyroelectric-photovoltaic phenomena for the first time “ in the response letter.

As the reviewer mentioned, it may cause concern on the agreement. We authors discussed to conclude that it is better to find more fundamentals on pyroelectric-photovoltaic assembled technology. We have studied literatures more intensively and it is quite reasonable to follow the reviewer's comments.

Through the revised manuscript, we have removed any kind of aggressive words and meanings. As researchers, we use the word ‘the possibility’ of the enhanced energy production with combining of photovoltaic and pyroelectric elements.

Thank you for the great comments on the scientific philosophy and advices.

In addition, the spectral measurements were performed under the 365 nm illumination (100uW/cm²), almost falling in the absorption range of heterostructure device. If pyroelectricity plays an important role, the device might work in a wide spectrum region beyond absorption. such as near-IR or even far. Is it possible?

Response

We sincerely thank the reviewer for this valuable comment on the spectral performances and role of pyroelectricity in harvesting the light spectrum beyond absorption. We addressed these comments, which led us to show the promise of pyroelectric-photovoltaic devices in harvesting infrared beyond absorption limits.

We performed spectral measurements under various light illumination wavelengths ranging from 340 to 850 nm (UV-visible-NIR), as shown in Supplementary Figure 25. The corresponding current-voltage characteristic plots of the device are provided in Supplementary Figure 26. We aimed to demonstrate the pyroelectric-photovoltaic device's ability to harvest energy beyond the absorption region, necessitating considering the bandgap value of ZnO, which is 3.25 eV. According to the formula $E = \frac{hv}{\lambda}$, where E is the photon energy, h is Planck's constant ($6.626 \times$

10^{-34} J s), c is the speed of light (3.0×10^8 m s⁻¹), and λ is the wavelength of light, the ZnO has a photon wavelength absorption cut-off value of 382.26 nm (λ (nm) = $\frac{1242.37}{E_g(eV)}$). Thanks to the pyroelectric characteristic of the device, the ZnO/NiO heterojunction can function in a broad spectrum beyond absorption, such as 400, 410, 460, 520, and 625 nm. We have included Figure R2-1 in the revised manuscript in response to the reviewer's feedback. This correction indicates the threshold line corresponding to the beyond absorption utilization to highlight the pyroelectric absorber's critical role.

Figure R2-1. Spectral characteristic of the ZnO embedded transparent pyroelectric-heterojunction device with beyond spectrum absorption utilization.

We want to emphasize the crucial role of pyroelectricity in harvesting near-IR or even far. While the current device doesn't respond to the IR spectrum, we experimented with some device structures by replacing the p-type NiO heterojunction partner with the suitable p-type SnS. It is essential to note that SnS is an emerging pyroelectric material among the 2D semiconductor family (Nanoscale, 2017, 9, 19201; Adv. Funct. Mater. 2020, 30, 2001450; npj Computational Materials 2023, 9, 45).

The device structure consists of a ZnO/SnS heterojunction with a ZnO thickness of 200 nm and the SnS layer thickness varying from 20 to 60 nm. The growth condition of SnS films was as per our previous study (Nanoscale, 2017, 9, 15804-15812). The device structure and original photo of the ZnO/SnS samples are shown in Figure R2-2a. The AgNW top electrode is a positive terminal, and the bottom FTO electrode is a negative terminal.

Under the infrared illumination of 850 nm wavelength with a pulse frequency of 30 Hz, both devices showed pyroelectric photoresponse under the self-bias operation, as shown in Figure R2-2b and 2-2c. According to the band-gap limit of 2D-SnS, which has a bandgap value of 1.6 eV (Nanoscale, 2017, 9, 15804-15812), the device has an absorption limit of 776.5 nm. This result indicates that a heterostructure device with suitable p-type heterojunction material, such as SnS, can operate in a wide spectrum region beyond absorption, such as near-IR.

Based on this strong motivation, we plan to report this development based on pyroelectric SnS to society as we gain a deeper understanding.

Figure R2-2. Pyroelectric-embedded ZnO/SnS heterostructure device for beyond spectrum absorption utilization for infrared application.

Changes made to the manuscript

On page 12.

“To further elucidate the pyroelectric properties, the device performance was profiled across a broad range of wavelengths of light. The ability of the pyroelectric-photovoltaic devices to harvest energy beyond the absorption region is related to the bandgap of ZnO, which is 3.25 eV. According to the formula $E = \frac{hc}{\lambda}$, where E is the photon energy, h is Planck’s constant (6.626×10^{-34} J s), c is the speed of light (3.0×10^8 m s⁻¹), and λ is the wavelength of light, ZnO has a photon wavelength absorption cutoff value of 382.26 nm ($\lambda (nm) = \frac{1242.37}{E_g(eV)}$). Due to the pyroelectric characteristics of the device, the ZnO/NiO heterojunction can function in a broad spectrum beyond absorption wavelengths, such as 400, 410, 460, 520, and 625 nm. The threshold line corresponding to beyond-photon absorption utilization in Fig. 5a and 5b highlights the crucial role of the pyroelectric absorber. This result clearly demonstrated that pyroelectric energy harvesting is possible for longer wavelengths beyond the photon absorption limit. Thus, this approach has the potential to harvest energy via pyroelectric utilization beyond the photon absorption limit.”

We hope the reviewer is satisfied with our responses and will consider our manuscript for possible publication in Nature Communications.

Thank you for your support and feedback.

Reviewer #3 (Remarks to the Author):

I have gone through the revised manuscript and the response letter from the authors. In my opinion, the authors have satisfactorily addressed the comments from the reviewers. The new version of the manuscript can be recommended for publication in Nature Communications.

Response

We sincerely thank the reviewer for reviewing our revised manuscript and response letter.

We are grateful to the reviewer for recommending this finding for its publication in Nature Communications.

Thank you for your support and feedback.